# Fast Kernel Methods for Generic Lipschitz Losses via $p$-Sparsified Sketches

**Tamim El Ahmad**                                    *tamim.elahmad@telecom-paris.fr*
*LTCI, Télécom Paris,*
*IP Paris, France*

**Pierre Laforgue**                                    *pierre.laforgue@unimi.it*
*Department of Computer Science,*
*University of Milan, Italy*

**Florence d'Alché-Buc**                                    *florence.dalche@telecom-paris.fr*
*LTCI, Télécom Paris,*
*IP Paris, France*

**Reviewed on OpenReview:** *https://openreview.net/forum?id=ry2qgRqTOw*

## Abstract

Kernel methods are learning algorithms that enjoy solid theoretical foundations while suffering from important computational limitations. Sketching, which consists in looking for solutions among a subspace of reduced dimension, is a well-studied approach to alleviate these computational burdens. However, statistically-accurate sketches, such as the Gaussian one, usually contain few null entries, such that their application to kernel methods and their non-sparse Gram matrices remains slow in practice. In this paper, we show that sparsified Gaussian (and Rademacher) sketches still produce theoretically-valid approximations while allowing for important time and space savings thanks to an efficient *decomposition trick*. To support our method, we derive excess risk bounds for both single and multiple output kernel problems, with generic Lipschitz losses, hereby providing new guarantees for a wide range of applications, from robust regression to multiple quantile regression. Our theoretical results are complemented with experiments showing the empirical superiority of our approach over state-of-the-art sketching methods.

## 1 Introduction

Kernel methods hold a privileged position in machine learning, as they allow to tackle a large variety of learning tasks in a unique and generic framework, that of Reproducing Kernel Hilbert Spaces (RKHSs), while enjoying solid theoretical foundations (Steinwart & Christmann, 2008b; Scholkopf & Smola, 2018). From scalar-valued to multiple output regression (Micchelli & Pontil, 2005; Carmeli et al., 2006; 2010), these approaches play a central role in nonparametric learning, showing a great flexibility. However, when implemented naively, kernel methods raise major issues in terms of time and memory complexity, and are often thought of as limited to "fat data", i.e., datasets of reduced size but with a large number of input features. One way to scale up kernel methods are the Random Fourier Features (Rahimi & Recht, 2007; Rudi & Rosasco, 2017; Sriperumbudur & Szabó, 2015; Li et al., 2021), but they mainly apply to shift-invariant kernels. Another popular approach is to use sketching methods, first exemplified with Nyström approximations (Williams & Seeger, 2001; Drineas et al., 2005; Bach, 2013; Rudi et al., 2015). Indeed, sketching has recently gained a lot of interest in the kernel community due to its wide applicability (Yang et al., 2017; Lacotte et al., 2019; Kpotufe & Sriperumbudur, 2020; Lacotte & Pilanci, 2022; Gazagnadou et al., 2022) and its spectacular successes when combined to preconditioners and GPUs (Meanti et al., 2020).

Sketching as a random projection method (Mahoney et al., 2011; Woodruff, 2014) is rooted in the Johnson-Lindenstrauss lemma (Johnson & Lindenstrauss, 1984), and consists in working in reduced dimension subspaces while benefiting from theoretical guarantees. Learning with sketched kernels has mostly been studied in the case of scalar-valued regression, in particular in the emblematic case of Kernel Ridge Regression (Alaoui & Mahoney, 2015; Avron et al., 2017; Yang et al., 2017; Chen & Yang, 2021a). For several identified sketching types (e.g., Gaussian, Randomized Orthogonal Systems, adaptive sub-sampling), the resulting estimators come with theoretical guarantees under the form of the minimax optimality of the empirical approximation error. However, an important blind spot of the above works is their limitation to the square loss. Few papers go beyond Ridge Regression, and usually exclusively with sub-sampling schemes (Zhang et al., 2012; Li et al., 2016; Della Vecchia et al., 2021). In this work, we derive excess risk bounds for sketched kernel machines with generic Lipschitz-continuous losses, under standard assumption on the sketch matrix, solving an open problem from Yang et al. (2017). Doing so, we provide theoretical guarantees for a wide range of applications, from robust regression, based either on the Huber loss (Huber, 1964) or $\epsilon$-insensitive losses (Steinwart & Christmann, 2008a), to quantile regression, tackled through the minimization of the pinball loss (Koenker, 2005). Further, we address this question in the general context of single and multiple output regression. Learning vector-valued functions using matrix-valued kernels (Micchelli & Pontil, 2005) has been primarily motivated by multi-task learning. Although equivalent in functional terms to scalar-valued kernels on pairs of input and tasks (Hein & Bousquet, 2004, Proposition 5), matrix-valued kernels (Álvarez et al., 2012) provide a way to define a larger variety of statistical learning problems by distinguishing the role of the inputs from that of the tasks. The computational and memory burden is naturally heavier in multi-task/multi-output regression, as the dimension of the output space plays an inevitable role, making approximation methods for matrix-valued kernel machines a crucial issue. To our knowledge, this work is the first to address this problem under the angle of sketching. It is however worth mentioning Baldassarre et al. (2012), who explored spectral filtering approaches for multiple output regression, and the generalization of Random Fourier Features to operator-valued kernels by Brault et al. (2016).

An important challenge when sketching kernel machines is that the sketched items, e.g., the Gram matrix, are usually dense. Plain sketching matrices, such as the Gaussian one, then induce significantly more calculations than sub-sampling methods, which can be computed by applying a mask over the Gram matrix. Sparse sketching matrices (Clarkson & Woodruff, 2017; Nelson & Nguyên, 2013; Cohen, 2016; Derezinski et al., 2021) constitute an important line of research to reduce complexity while keeping good statistical properties when applied to sparse matrices (e.g., matrices induced by graphs), which is not the case of a Gram matrix. Motivated by these considerations, we analyze a family of sketches, unified under the name of $p$-sparsified sketches, that achieve interesting trade-offs between statistical accuracy (Gaussian sketches can be recovered as a particular case of $p$-sparsified sketches) and computational efficiency. The $p$-sparsified sketches are also memory-efficient, as they do not require computing and storing the full Gram matrix upfront. Besides theoretical analysis, we provide extensive experiments showing the superiority of $p$-sparsified sketches over SOTA approaches such as accumulation sketches (Chen & Yang, 2021a).

**Contributions.** Our goal is to provide a framework to speed-up both scalar and matrix-valued kernel methods which is as general as possible while maintaining good theoretical guarantees. For that purpose, we present three contributions, which may be of independent interest.

- We derive excess risk bounds for sketched kernel machines with generic Lipschitz-continuous losses, both in the scalar and multiple output cases. We hereby solve an open problem from Yang et al. (2017), and provide a first analysis to the sketching of vector-valued kernel methods.

- We show that sparsified Gaussian and Rademacher sketches provide valid approximations when applied to kernel methods. They maintain theoretical guarantees while inducing important space and computation savings, as opposed to plain sketches.

- We discuss how to learn these new sketched kernel machines, by means of an approximated feature map. We finally present experiments using Lipschitz-continuous losses, such as robust and quantile regression, on both synthetic and real-world datasets, supporting the relevance of our approach.

**Notation.** For any matrix $A \in \mathbb{R}^{m \times p}$, $A^\dagger$ is its pseudo-inverse, $\|A\|_{\mathrm{op}}$ its operator norm, $A_{i:} \in \mathbb{R}^p$ its $i$-th row, and $A_{\cdot j} \in \mathbb{R}^m$ its $j$-th column. The identity matrix of dimension $d$ is $I_d$. For a couple of random variables $(X, Y) \in \mathcal{X} \times \mathcal{Y}$ with distribution $P$, $P_X$ is the marginal distribution of $X$. For $f : \mathcal{X} \longrightarrow \mathcal{Y}$, we use $\mathbb{E}[f] = \mathbb{E}_{P_X}[f(X)]$, $\mathbb{E}[\ell_f] = \mathbb{E}_P[\ell(f(X), Y)]$ and $\mathbb{E}_n[\ell_f] = \frac{1}{n} \sum_{i=1}^n \ell(f(x_i), y_i)$ for any function $\ell : \mathcal{Y} \times \mathcal{Y} \longrightarrow \mathbb{R}$.

## 2 Sketching Kernels Machines with Lipschitz-Continuous Losses

In this section, we derive excess risk bounds for sketched kernel machines with generic Lipschitz losses, for both scalar and multiple output regression.

### 2.1 Scalar Kernel Machines

We consider a general regression framework, from an input space $\mathcal{X}$ to some scalar output space $\mathcal{Y} \subseteq \mathbb{R}$. Given a loss function $\ell : \mathcal{Y} \times \mathcal{Y} \to \mathbb{R}$ such that $z \mapsto \ell(z, y)$ is proper, lower semi-continuous and convex for every $y$, our goal is to estimate $f^* = \arg\inf_{f \in \mathcal{H}} \mathbb{E}_{(X,Y) \sim P}[\ell(f(X), Y)]$, where $\mathcal{H} \subset \mathcal{Y}^{\mathcal{X}}$ is a hypothesis set, and $P$ is a joint distribution over $\mathcal{X} \times \mathcal{Y}$. Since $P$ is usually unknown, we assume that we have access to a training dataset $\{(x_i, y_i)\}_{i=1}^n$ composed of i.i.d. realisations drawn from $P$. We recall the definitions of a scalar-valued kernel and its RKHS (Aronszajn, 1950).

**Definition 1** (Scalar-valued kernel). *A scalar-valued kernel is a symmetric function $k : \mathcal{X} \times \mathcal{X} \to \mathbb{R}$ such that for all $n \in \mathbb{N}$, and any $(x_i)_{i=1}^n \in \mathcal{X}^n$, $(\alpha_i)_{i=1}^n \in \mathbb{R}^n$, we have $\sum_{i,j=1}^n \alpha_i\, k(x_i, x_j)\, \alpha_j \geq 0$.*

**Theorem 1** (RKHS). *Let $k$ be a kernel on $\mathcal{X}$. Then, there exists a unique Hilbert space of functions $\mathcal{H}_k \subset \mathbb{R}^{\mathcal{X}}$ such that $k(\cdot, x) \in \mathcal{H}_k$ for all $x \in \mathcal{X}$, and such that we have $h(x) = \langle h, k(\cdot, x) \rangle_{\mathcal{H}_k}$ for any $(h, x) \in \mathcal{H}_k \times \mathcal{X}$.*

A kernel machine computes a proxy for $f^*$ by solving

$$\min_{f \in \mathcal{H}_k} \frac{1}{n} \sum_{i=1}^n \ell(f(x_i), y_i) + \frac{\lambda_n}{2} \|f\|_{\mathcal{H}_k}^2, \tag{1}$$

where $\lambda_n > 0$ is a regularization parameter. By the representer theorem (Kimeldorf & Wahba, 1971; Schölkopf et al., 2001), the solution to Problem (1) is given by $\hat{f}_n = \sum_{i=1}^n \hat{\alpha}_i\, k(\cdot, x_i)$, with $\hat{\alpha} \in \mathbb{R}^n$ the solution to

$$\min_{\alpha \in \mathbb{R}^n} \frac{1}{n} \sum_{i=1}^n \ell([K\alpha]_i, y_i) + \frac{\lambda_n}{2} \alpha^\top K \alpha, \tag{2}$$

where $K \in \mathbb{R}^{n \times n}$ is the kernel Gram matrix such that $K_{ij} = k(x_i, x_j)$.

**Definition 2** (Regularized Kernel-based Sketched Estimator). *Given a matrix $S \in \mathbb{R}^{s \times n}$, with $s \ll n$, sketching consists in imposing the substitution $\alpha = S^\top \gamma$ in the empirical risk minimization problem stated in Equation (2). We then obtain an optimisation problem of reduced size on $\gamma$, that yields the sketched estimator $\tilde{f}_s = \sum_{i=1}^n [S^\top \tilde{\gamma}]_i\, k(\cdot, x_i)$, where $\tilde{\gamma} \in \mathbb{R}^s$ is a solution to*

$$\min_{\gamma \in \mathbb{R}^s} \frac{1}{n} \sum_{i=1}^n \ell([KS^\top \gamma]_i, y_i) + \frac{\lambda_n}{2} \gamma^\top SKS^\top \gamma. \tag{3}$$

In practice, one usually obtains the matrix $S$ by sampling it from a random distribution. The literature is rich in examples of distributions that can be used to generate the sketching matrix $S$. For instance, the sub-sampling matrices, where each line of $S$ is sampled from $I_n$, have been widely studied in the context of kernel methods. They are computationally efficient from both time and space perspectives, and yield the so-called Nyström approach (Williams & Seeger, 2001; Rudi et al., 2015). More complex distributions, such as Randomized Orthogonal System (ROS) sketching or Gaussian sketch matrices, have also been considered (Yang et al., 2017). In this work, we first give a general theoretical analysis of regularized kernel-based

sketched estimators for any $K$-satisfiable sketch matrix (Definition 3). Then, we introduce the $p$-sparsified sketches and prove their $K$-satisfiablity, as well as their relevance for kernel methods in terms of statistical and computational trade-off.

Works about sketched kernel machines usually assess the performance of $\tilde{f}_s$ by upper bounding its squared $L^2(\mathbb{P}_N)$ error, i.e., $(1/n)\sum_{i=1}^n(\tilde{f}_s(x_i)-f_{\mathcal{H}_k}(x_i))^2$, where $f_{\mathcal{H}_k}$ is the minimizer of the true risk over $\mathcal{H}_k$, supposed to be attained (Yang et al., 2017, Equation 2), or through its (relative) recovery error $\|\tilde{f}_s-\hat{f}_n\|_{\mathcal{H}_k}/\|\hat{f}_n\|_{\mathcal{H}_k}$, see Lacotte & Pilanci (2022, Theorem 3). In contrast, we focus on the excess risk of $\tilde{f}_s$, the original quantity of interest. As revealed by the proof of Theorem 2, the approximation error of the excess risk can be controlled in terms of the $L^2(\mathbb{P}_N)$ error, and we actually recover the results from Yang et al. (2017) when we particularize to the square loss with bounded outputs (second bound in Theorem 2). Furthermore, studying the excess risk allows to better position the performances of $\tilde{f}_s$ among the known off-the-shelf kernel-based estimators available for the targeted problem. To achieve this study, we rely on the key notion of $K$-satisfiability for a sketch matrix (Yang et al., 2017; Liu et al., 2019; Chen & Yang, 2021a).

Let $K/n = UDU^\top$ be the eigendecomposition of the Gram matrix, where $D = \mathrm{diag}\,(\mu_1,\ldots,\mu_n)$ stores the eigenvalues of $K/n$ in decreasing order. Let $\delta_n^2$ be the critical radius of $K/n$, i.e., the lowest value such that $\psi(\delta_n) = (\frac{1}{n}\sum_{i=1}^n \min(\delta_n^2,\mu_i))^{1/2} \leq \delta_n^2$. The existence and uniqueness of $\delta_n^2$ is guaranteed for any RKHS associated with a positive definite kernel (Bartlett et al., 2006; Yang et al., 2017). Note that $\delta_n^2$ is similar to the parameter $\tilde{\varepsilon}^2$ used in Yang et al. (2012) to analyze Nyström approximation for kernel methods. We define the statistical dimension of $K$ as $d_n = \min\,\{j \in \{1,\ldots,n\}\colon \mu_j \leq \delta_n^2\}$, with $d_n = n$ if no such index $j$ exists.

**Definition 3** ($K$-satisfiability, Yang et al. 2017)**.** *Let $c > 0$ be independent of $n$, $U_1 \in \mathbb{R}^{n\times d_n}$ and $U_2 \in \mathbb{R}^{n\times(n-d_n)}$ be the left and right blocks of the matrix $U$ previously defined, and $D_2 = \mathrm{diag}\,(\mu_{d_n+1},\ldots,\mu_n)$. A matrix $S$ is said to be $K$-satisfiable for $c$ if we have*

$$\left\|(SU_1)^\top SU_1 - I_{d_n}\right\|_{\mathrm{op}} \leq 1/2\,, \quad and \quad \left\|SU_2 D_2^{1/2}\right\|_{\mathrm{op}} \leq c\delta_n\,. \tag{4}$$

Roughly speaking, a matrix is $K$-satisfiable if it defines an isometry on the largest eigenvectors of $K$, and has a small operator norm on the smallest eigenvectors. For random sketching matrices, it is common to show $K$-satisfiability with high probability under some condition on the sketch size $s$, see e.g., Yang et al. (2017, Lemma 5) for Gaussian sketches, Chen & Yang (2021a, Theorem 8) for Accumulation sketches. In Section 3, we show similar results for $p$-sparsified sketches.

To derive our excess risk bounds, we place ourselves in the framework of Li et al. (2021), see Sections 2.1 and 3 therein. Namely, we assume that the true risk is minimized over $\mathcal{H}_k$ at $f_{\mathcal{H}_k} \coloneqq \arg\min_{f\in\mathcal{H}_k}\,\mathbb{E}\,[\ell\,(f\,(X)\,,Y)]$. The existence of $f_{\mathcal{H}_k}$ is standard in the literature (Caponnetto & De Vito, 2007; Rudi & Rosasco, 2017; Yang et al., 2017), and implies that $f_{\mathcal{H}_k}$ has bounded norm, see e.g., Rudi & Rosasco (2017, Remark 2). Similarly to Li et al. (2021), we also assume that estimators returned by Empirical Risk Minimization have bounded norms. Hence, all estimators considered in the present paper belong to some ball of finite radius $R$. However, we highlight that our results do not require prior knowledge on $R$, and hold uniformly for all finite $R$. As a consequence, we consider without loss of generality as hypothesis set the unit ball $\mathcal{B}\,(\mathcal{H}_k)$ in $\mathcal{H}_k$, up to an *a posteriori* rescaling of the bounds by $R$ to recover the general case.

**Assumption 1.** *The true risk is minimized at $f_{\mathcal{H}_k}$.*

**Assumption 2.** *The hypothesis set considered is $\mathcal{B}\,(\mathcal{H}_k)$.*

**Assumption 3.** *For all $y \in \mathcal{Y}$, $z \mapsto \ell(z,y)$ is $L$-Lipschitz.*

**Assumption 4.** *For all $x,x' \in \mathcal{X}$, we have $k(x,x') \leq \kappa$.*

**Assumption 5.** *The sketch $S$ is $K$-satisfiable with constant $c > 0$.*

Note that we discuss some directions to relax Assumption 2 in Appendix B. Many loss functions satisfy Assumption 3, such as the hinge loss ($L = 1$), used in SVMs (Cortes & Vapnik, 1995), the $\epsilon$-insensitive $\ell_1$ (Drucker et al., 1997), the $\kappa$-Huber loss, known for robust regression (Huber, 1964), the pinball loss, used in quantile regression (Steinwart & Christmann, 2011), or the square loss with bounded outputs. Assumption 4 is standard (e.g., $\kappa = 1$ for the Gaussian kernel). Under Assumptions 1 to 5 we have the following result.

**Theorem 2.** *Let $\tilde{f}_s$ as in Definition 2, suppose that Assumptions 1 to 5 hold, and let $C = 1 + \sqrt{6}c$, with $c$ the constant from Assumption 5. Then, for any $\delta \in (0,1)$ with probability at least $1 - \delta$ we have*

$$\mathbb{E}\left[\ell_{\tilde{f}_s}\right] \leq \mathbb{E}\left[\ell_{f_{\mathcal{H}_k}}\right] + LC\sqrt{\lambda_n + \delta_n^2} + \frac{\lambda_n}{2} + 8L\sqrt{\frac{\kappa}{n}} + 2\sqrt{\frac{8\log(4/\delta)}{n}}. \tag{5}$$

*Furthermore, if $\ell(z,y) = (z-y)^2/2$ and $\mathcal{Y} \subset [0,1]$, with probability at least $1 - \delta$ we have*

$$\mathbb{E}\left[\ell_{\tilde{f}_s}\right] \leq \mathbb{E}\left[\ell_{f_{\mathcal{H}_k}}\right] + \left(C^2 + \frac{1}{2}\right)\lambda_n + C^2\delta_n^2 + 8\frac{\kappa + \sqrt{\kappa}}{\sqrt{n}} + 2\sqrt{\frac{8\log(4/\delta)}{n}}. \tag{6}$$

*Proof sketch.* The proof relies on the decomposition of the excess risk into two generalization error terms and an approximation error term, i.e.,

$$\mathbb{E}[\ell_{\tilde{f}_s}] - \mathbb{E}[\ell_{f_{\mathcal{H}_k}}] = \mathbb{E}[\ell_{\tilde{f}_s}] - \mathbb{E}_n[\ell_{\tilde{f}_s}] + \mathbb{E}_n[\ell_{\tilde{f}_s}] - \mathbb{E}_n[\ell_{f_{\mathcal{H}_k}}] + \mathbb{E}_n[\ell_{f_{\mathcal{H}_k}}] - \mathbb{E}[\ell_{f_{\mathcal{H}_k}}]. \tag{7}$$

The two generalization errors (of $\tilde{f}_s$ and $f_{\mathcal{H}_k}$) can be bounded using Bartlett & Mendelson (2003, Theorem 8) together with Assumptions 1 to 4. For the last term, we can use Jensen's inequality and the Lipschitz continuity of the loss to upper bound this approximation error by the square root of the sum of the square residuals of the Kernel Ridge Regression with targets the $f_{\mathcal{H}_k}(x_i)$. The latter can in turn be upper bounded using Assumptions 1 and 5 and Lemma 2 from Yang et al. (2017). When considering the square loss, Jensen's inequality is not necessary anymore, leading to the improved second term in the right-hand side of the last inequality in Theorem 2. $\qquad \square$

Recall that the rates in Theorem 2 are incomparable as is to that of Yang et al. (2017, Theorem 2), since we focus on the excess risk while the authors study the squared $L^2(\mathbb{P}_N)$ error. Precisely, we recover their results as a particular case with the square loss and bounded outputs, up to the generalization errors. Instead, note that we do recover the rates of Li et al. (2021, Theorem 1), based on a similar framework. Our bounds feature two different terms: a quantity related to the generalization errors, and a quantity governed by $\delta_n$, deriving from the $K$-satisfiability analysis. The behaviour of the critical radius $\delta_n$ crucially depends on the choice of the kernel. In Yang et al. (2017), the authors compute its decay rate for different kernels. For instance, we have $\delta_n^2 = \mathcal{O}(\sqrt{\log(n)}/n)$ for the Gaussian kernel, $\delta_n^2 = \mathcal{O}(1/n)$ for polynomial kernels, or $\delta_n^2 = \mathcal{O}(n^{-2/3})$ for first-order Sobolev kernels. Note finally that by setting $\lambda_n \propto 1/\sqrt{n}$ we attain a rate of $\mathcal{O}(1/\sqrt{n})$, that is minimax for the kernel ridge regression, see Caponnetto & De Vito (2007).

**Remark 1.** *Note that a standard additional assumption on the second order moments of the functions in $\mathcal{H}_k$ (Bartlett et al., 2005) allows to derive refined learning rates for the generalization errors. These refined rates are expressed in terms of $\hat{r}_{\mathcal{H}_k}^\star$, the fixed point of a new sub-root function $\hat{\psi}_n$. In order to make the approximation error of the same order, it is then necessary to prove the $K$-satisfiability of $S$ with respect to $\hat{r}_{\mathcal{H}_k}^{\star 2}$ instead of $\delta_n^2$. Whether it is possible to prove such a $K$-satisfiability for standard sketches is however a nontrivial question, left as future work.*

## 2.2 Matrix-valued Kernel Machines

In this section, we extend our results to multiple output regression, tackled in vector-valued RKHSs. Note that the output space $\mathcal{Y}$ is now a subset of $\mathbb{R}^d$, with $d \geq 2$. We start by recalling important notions about Matrix-Valued Kernels (MVKs) and vector-valued RKHSs (vv-RKHSs).

**Definition 4** (Matrix-valued kernel). *A MVK is an application $\mathcal{K} : \mathcal{X} \times \mathcal{X} \rightarrow \mathcal{L}(\mathbb{R}^d)$, where $\mathcal{L}(\mathbb{R}^d)$ is the set of bounded linear operators on $\mathbb{R}^d$, such that $\mathcal{K}(x,x') = \mathcal{K}(x',x)^\top$ for all $(x,x') \in \mathcal{X}^2$, and such that for all $n \in \mathbb{N}$ and any $(x_i,y_i)_{i=1}^n \in (\mathcal{X} \times \mathcal{Y})^n$ we have $\sum_{i,j=1}^n y_i^\top \mathcal{K}(x_i,x_j) y_j \geqslant 0$.*

**Theorem 3** (Vector-valued RKHS). *Let $\mathcal{K}$ be a MVK. There is a unique Hilbert space $\mathcal{H}_\mathcal{K} \subset \mathcal{F}(\mathcal{X}, \mathbb{R}^d)$, the vv-RKHS of $\mathcal{K}$, such that for all $x \in \mathcal{X}$, $y \in \mathbb{R}^d$ and $f \in \mathcal{H}_\mathcal{K}$ we have $x' \mapsto \mathcal{K}(x,x') y \in \mathcal{H}_\mathcal{K}$, and $\langle f, \mathcal{K}(\cdot,x) y \rangle_\mathcal{H} = f(x)^\top y$.*

Note that we focus in this paper on the finite-dimensional case, i.e., $\mathcal{Y} \subset \mathbb{R}^d$, such that for all $x, x' \in \mathcal{X}$, we have $\mathcal{K}(x, x') \in \mathbb{R}^{d \times d}$. For a training sample $\{x_1, \ldots, x_n\}$, we define the Gram matrix as $\mathbf{K} = (\mathcal{K}(x_i, x_j))_{1 \leq i,j \leq n} \in \mathbb{R}^{nd \times nd}$. A common assumption consists in considering decomposable kernels: we assume that there exist a scalar kernel $k$ and a positive semidefinite matrix $M \in \mathbb{R}^{d \times d}$ such that for all $x, x' \in \mathcal{X}$ we have $\mathcal{K}(x, x') = k(x, x')M$. The Gram matrix can then be written $\mathbf{K} = K \otimes M$, where $K \in \mathbb{R}^{n \times n}$ is the scalar Gram matrix, and $\otimes$ denotes the Kronecker product. Decomposable kernels are widely spread in the literature as they provide a good compromise between computational simplicity and expressivity —note that in particular they encapsulate independent learning, achieved with $M = I_d$. We now discuss two examples of relevant output matrices.

**Example 1.** *In joint quantile regression, one is interested in predicting d different conditional quantiles of an output y given the input x. If $(\tau_i)_{i \leq d} \in (0, 1)$ denote the d different quantile levels, it has been shown in Sangnier et al. (2016) that choosing $M_{ij} = \exp(-\gamma(\tau_i - \tau_j)^2)$ favors close predictions for close quantile levels, while limiting crossing effects.*

**Example 2.** *In multiple output regression, it is possible to leverage prior knowledge on the task relationships to design a relevant output matrix M. For instance, let P be the $d \times d$ adjacency matrix of a graph in which the vertices are the tasks and an edge exists between two tasks if and only if they are (thought to be) related. Denoting by $L_P$ the graph Laplacian associated to P, Evgeniou et al. (2005) and Sheldon (2008) have proposed to use $M = (\mu L_P + (1 - \mu)I_d)^{-1}$, with $\mu \in [0, 1]$. When $\mu = 0$, we have $M = I_d$ and all tasks are considered independent. When $\mu = 1$, we only rely on the prior knowledge encoded in P.*

Given a sample $(x_i, y_i)_{i=1}^n \in (\mathcal{X}, \mathbb{R}^d)^n$ and a decomposable kernel $\mathcal{K} = kM$ (its associated vv-RKHS is $\mathcal{H}_{\mathcal{K}}$), the penalized empirical risk minimisation problem is

$$\min_{f \in \mathcal{H}_{\mathcal{K}}} \frac{1}{n} \sum_{i=1}^n \ell(f(x_i), y_i) + \frac{\lambda_n}{2} \|f\|_{\mathcal{H}_K}^2 \,, \tag{8}$$

where $\ell : \mathbb{R}^d \times \mathbb{R}^d \to \mathbb{R}$ is a loss such that $z \mapsto \ell(z, y)$ is proper, lower semi-continuous and convex for all $y \in \mathbb{R}^d$. By the vector-valued representer theorem (Micchelli & Pontil, 2005), we have that the solution to Problem (8) writes $\hat{f}_n = \sum_{j=1}^n \mathcal{K}(\cdot, x_j)\hat{\alpha}_j = \sum_{j=1}^n k(\cdot, x_j)M\hat{\alpha}_j$, where $\hat{A} = (\hat{\alpha}_1, \ldots, \hat{\alpha}_n)^\top \in \mathbb{R}^{n \times d}$ is the solution to the problem

$$\min_{A \in \mathbb{R}^{n \times d}} \frac{1}{n} \sum_{i=1}^n \ell\left([KAM]_{i:}^\top, y_i\right) + \frac{\lambda_n}{2} \operatorname{Tr}\left(KAMA^\top\right) \,.$$

In this context, sketching consists in making the substitution $A = S^\top \Gamma$, where $S \in \mathbb{R}^{s \times n}$ is a sketch matrix and $\Gamma \in \mathbb{R}^{s \times d}$ is the parameter of reduced dimension to be learned. The solution to the sketched problem is then $\tilde{f}_s = \sum_{j=1}^n k(\cdot, x_j)M[S^\top \tilde{\Gamma}]_{j:}$, with $\tilde{\Gamma} \in \mathbb{R}^{s \times d}$ minimizing

$$\frac{1}{n} \sum_{i=1}^n \ell\left([KS^\top \Gamma M]_{i:}, y_i\right) + \frac{\lambda_n}{2} \operatorname{Tr}\left(SKS^\top \Gamma M \Gamma^\top\right) \,.$$

**Theorem 4.** *Suppose that Assumptions 1 to 5 hold, that $\mathcal{K} = kM$ is a decomposable kernel with M invertible, and let C as in Theorem 2. Then for any $\delta \in (0, 1)$ with probability at least $1 - \delta$ we have*

$$\mathbb{E}\left[\ell_{\tilde{f}_s}\right] \leq \mathbb{E}\left[\ell_{f_{\mathcal{H}_{\mathcal{K}}}}\right] + LC\sqrt{\lambda_n + \|M\|_{\text{op}} \delta_n^2} + \frac{\lambda_n}{2} + 8L\sqrt{\frac{\kappa \operatorname{Tr}(M)}{n}} + 2\sqrt{\frac{8 \log(4/\delta)}{n}} \,. \tag{9}$$

*Furthermore, if $\ell(z, y) = \|z - y\|_2^2 / 2$ and $\mathcal{Y} \subset \mathcal{B}(\mathbb{R}^d)$, with probability at least $1 - \delta$ we have that*

$$\mathbb{E}\left[\ell_{\tilde{f}_s}\right] \leq \mathbb{E}\left[\ell_{f_{\mathcal{H}_k}}\right] + \left(C^2 + \frac{1}{2}\right)\lambda_n + C^2 \|M\|_{\text{op}} \delta_n^2 + 8 \operatorname{Tr}(M)^{1/2} \frac{\kappa \|M\|_{\text{op}}^{1/2} + \kappa^{1/2}}{\sqrt{n}} + 2\sqrt{\frac{8 \log(4/\delta)}{n}} \,. \tag{10}$$

*Proof sketch.* The proof follows that of Theorem 2. The main challenge is to adapt Yang et al. (2017, Lemma 2) to the multiple output setting. To do so, we leverage that $\mathcal{K}$ is decomposable, such that the $K$-satisfiability of $S$ is sufficient, where $K$ the scalar Gram matrix. $\qquad\square$

Note that for $M = I_d$ (independent prior), the third term of the right-hand side of both inequalities becomes of order $\sqrt{d/n}$, that is typical of multiple output problems. If moreover we instantiate the bound for $d = 1$, we recover exactly Theorem 2. Finally, similarly to the scalar case in Theorem 2, looking at the least square case (Equation (10)), by setting $\lambda_n \propto 1/\sqrt{n}$, we attain the minimax rate of $\mathcal{O}(1/\sqrt{n})$, as stated in Caponnetto & De Vito (2007) and Ciliberto et al. (2020, Theorem 5). To the best of our knowledge, Theorem 4 is the first theoretical result about sketched vector-valued kernel machines. We highlight that it applies to generic Lipschitz losses and provides a bound directly on the excess risk.

## 2.3 Algorithmic details

We now discuss how to solve single and multiple output optimization problems. Let $\{(\tilde{\mu}_i, \tilde{\mathbf{v}}_i), i \in [s]\}$ be the eigenpairs of $SKS^\top$ in descending order, $\tilde{U} = [\tilde{U}_{ij}]_{s \times s} = (\tilde{\mathbf{v}}_1, \ldots, \tilde{\mathbf{v}}_s)$, $r = \text{rank}(SKS^\top)$, and $\tilde{K}_r = \tilde{U}_r \tilde{D}_r^{-1/2}$, where $\tilde{D}_r = \text{diag}(\tilde{\mu}_1, \ldots, \tilde{\mu}_r)$, and $\tilde{U}_r = (\tilde{\mathbf{v}}_1, \ldots, \tilde{\mathbf{v}}_r)$.

**Proposition 1.** *Solving Problem* (3) *is equivalent to solving*

$$\min_{\omega \in \mathbb{R}^r} \frac{1}{n} \sum_{i=1}^n \ell \left( \omega^\top \mathbf{z}_S \left( x_i \right), y_i \right) + \frac{\lambda_n}{2} \|\omega\|_2^2, \tag{11}$$

*where* $\mathbf{z}_S(x) = \tilde{K}_r^\top S \left( k(x, x_1), \ldots, k(x, x_n) \right)^\top \in \mathbb{R}^r$.

Problem (11) thus writes as a linear problem with respect to the feature maps induced by the sketch, generalizing the results established in Yang et al. (2012) for sub-sampling sketches. When considering multiple outputs, it is also possible to derive a linear feature map version when the kernel is decomposable. These feature maps are of the form $\mathbf{z}_S \otimes M^{1/2}$, yielding matrices of size $nd \times rd$ that are prohibitive in terms of space, see Appendix E. Note that an alternative way is to see sketching as a projection of the $k(\cdot, x_i)$ into $\mathbb{R}^r$ (Chatalic et al., 2021). Instead, we directly learn $\Gamma$. For both single and multiple output problems, we consider losses not differentiable everywhere in Section 4 and apply ADAM Stochastic Subgradient Descent (Kingma & Ba, 2015) for its ability to handle large datasets.

**Remark 2.** *In the previous sections, sketching is always leveraged in primal problems. However, for some of the loss functions we consider, dual problems are usually more attractive (Cortes & Vapnik, 1995; Laforgue et al., 2020). This naturally raises the question of investigating the interplay between sketching and duality on the algorithmic level. More details can be found in Appendix F.*

## 3 $p$-Sparsified Sketches

We now introduce the $p$-sparsified sketches, and establish their $K$-satisfiability. The $p$-sparsified sketching matrices are composed of i.i.d. Rademacher or centered Gaussian entries, multiplied by independent Bernoulli variables of parameter $p$ (the non-zero entries are scaled to ensure that $S$ defines an isometry in expectation). The sketch sparsity is controlled by $p$, and when the latter becomes small enough, $S$ contains many columns full of zeros. It is then possible to rewrite $S$ as the product of a sub-Gaussian and a sub-sampling sketch of reduced size, which greatly accelerates the computations.

**Definition 5.** *Let $s < n$, and $p \in (0, 1]$. A $p$-Sparsified Rademacher ($p$-SR) sketching matrix is a random matrix $S \in \mathbb{R}^{s \times n}$ whose entries $S_{ij}$ are independent and identically distributed (i.i.d.) as follows*

$$S_{ij} = \begin{cases} \frac{1}{\sqrt{sp}} & \text{with probability} & \frac{p}{2} \\ 0 & \text{with probability} & 1 - p \\ \frac{-1}{\sqrt{sp}} & \text{with probability} & \frac{p}{2} \end{cases} \tag{12}$$

*A $p$-Sparsified Gaussian ($p$-SG) sketching matrix is a random matrix $S \in \mathbb{R}^{s \times n}$ whose entries $S_{ij}$ are i.i.d. as follows*

$$S_{ij} = \begin{cases} \frac{1}{\sqrt{sp}} G_{ij} & \text{with probability} & p \\ 0 & \text{with probability} & 1 - p \end{cases} \tag{13}$$

*where the $G_{ij}$ are i.i.d. standard normal random variables. Note that standard Gaussian sketches are a special case of p-SG sketches, corresponding to $p = 1$.*

Several works partially addressed $p$-SR sketches in the past literature. For instance, Baraniuk et al. (2008) establish that $p$-SR sketches satisfy the Restricted Isometry Property (based on concentration results from Achlioptas (2001)), but only for $p = 1$ and $p = 1/3$. In Li et al. (2006), the authors consider generic $p$-SR sketches, but do not provide any theoretical result outside of a moment analysis. The *i.i.d. sparse embedding matrices* from Cohen (2016) are basically $m/s$-SR sketches, where $m \geq 1$, leading each column to have exactly $m$ nonzero elements in expectation. However, we were not able to reproduce the proof of the Johnson-Linderstrauss property proposed by the author for his sketch (Theorem 4.2 in the paper, equivalent to the first claim of $K$-satisfiability, left-hand side of (4)). More precisely, we think that the assumptions considering "each entry is independently nonzero with probability $m/s$" and "each column has a fixed number of nonzero entries" ($m$ here) are conflicting. As far as we know, this is the first time $p$-SG sketches are introduced in the literature. Note that both (12) and (13) can be rewritten as $S_{ij} = (1/\sqrt{sp})B_{ij}R_{ij}$, where the $B_{ij}$ are i.i.d. Bernouilli random variables of parameter $p$, and the $R_{ij}$ are i.i.d. random variables, independent from the $B_{ij}$, such that $\mathbb{E}[R_{ij}] = 0$ and $\mathbb{E}[R_{ij}R_{i'j'}] = 1$ if $i = i'$ and $j = j'$, and 0 otherwise. Namely, for $p$-SG sketches $R_{ij} = G_{ij}$ is a standard Gaussian variable while for $p$-SR sketches it is a Rademacher random variable. It is then easy to check that $p$-SR and $p$-SG sketches define isometries in expectation. In the next theorem, we show that $p$-sparsified sketches are $K$-satisfiable with high probability.

**Theorem 5.** *Let $S$ be a p-sparsified sketching matrix. Then, there are some universal constants $C_0, C_1 > 0$ and a constant $c(p)$, increasing with $p$, such that for $s \geq \max\left(C_0 d_n/p^2, \delta_n^2 n\right)$ and with a probability at least $1 - C_1 e^{-sc(p)}$, the sketch $S$ is $K$-satisfiable for $c = \frac{2}{\sqrt{p}}\left(1 + \sqrt{\log(5)}\right) + 1$.*

*Proof sketch.* To prove the left-hand side of (4), we use Boucheron et al. (2013, Theorem 2.13), which shows that any i.i.d. sub-Gaussian sketch matrix satisfies the Johnson-Lindenstrauss lemma with high probability. To prove the right-hand side of (4), we work conditionally on a realization of the $B_{ij}$, and use concentration results of Lipschitz functions of Rademacher or Gaussian random variables (Tao, 2012). We highlight that such concentration results do not hold for sub-Gaussian random variables in general, preventing from showing $K$-satisfiability of generic sparsified sub-Gaussian sketches. Note that having $S_{ij} \propto B_{ij}R_{ij}$ is key, and that sub-sampling uniformly at random non-zero entries instead of using i.i.d. Bernoulli variables would make the proof significantly more complex. We highlight that Theorem 5 strictly generalizes Yang et al. (2017, Lemma 5), recovered for $p = 1$, and extends the results to Rademacher sketches. □

**Computational property of $p$-sparsified sketches.** In addition to be statistically accurate, $p$-sparsified sketches are computationally efficient. Indeed, recall that the main quantity one has to compute when sketching a kernel machine is the matrix $SKS^\top$. With standard Gaussian sketches, that are known to be theoretically accurate, this computation takes $\mathcal{O}(sn^2)$ operations. Sub-sampling sketches are notoriously less precise, but since they act as masks over the Gram matrix $K$, computing $SKS^\top$ can be done in $\mathcal{O}(s^2)$ operations only, without having to store the entire Gram matrix upfront. Now, let $S \in \mathbb{R}^{s \times n}$ be a $p$-sparsified sketch, and $s' = \sum_{j=1}^n \mathbb{I}\{S_{:j} \neq 0_s\}$ be the number of columns of $S$ with at least one nonzero element. The crucial observation that makes $S$ computationally efficient is that we have

$$S = S_{\text{SG}}\, S_{\text{SS}}\,, \tag{14}$$

where $S_{\text{SG}} \in \mathbb{R}^{s \times s'}$ is obtained by deleting the null columns from $S$, and $S_{\text{SS}} \in \mathbb{R}^{s' \times n}$ is a sub-Sampling sketch whose sampling indices correspond to the indices of the columns in $S$ with at least one non-zero entry[1]. We refer to (14) as the *decomposition trick*. This decomposition is key, as we can apply first a fast sub-sampling sketch, and then a sub-Gaussian sketch on the sub-sampled Gram matrix of reduced size. Note that $s'$ is a random variable. By independence of the entries, each column is null with probability $(1-p)^s$. Then, by the independence of the columns we have that $s'$ follows a Binomial distribution with parameters $n$ and $1 - (1-p)^s$, such that $\mathbb{E}[s'] = n(1 - (1-p)^s)$.

---

[1]Precisely, $S_{\text{SS}}$ is the identity matrix $I_{s'}$, augmented with $n - s'$ null columns inserted at the indices of the null columns of $S$.

Hence, the sparsity of the $p$-sparsified sketches, controlled by parameter $p$, is an interesting degree of freedom to add: it preserves statistical guarantees (Theorem 5) while speeding-up calculations (14). Of course, there is no free lunch and one looses on one side what is gained on the other: when $p$ decreases (sparser sketches), the lower bound to get guarantees $s \gtrsim d_n/p^2$ increases, but the expected number of non-null columns $s'$ decreases, thus accelerating computations (note that for $p = 1$ we exactly recover the lower bound and number of non-null columns for Gaussian sketches).

**Corollary 1.** *Let $S \in \mathbb{R}^{s \times n}$ be a $p$-sparsified sketching matrix, and $C_0$, $C_1$ and $c(p)$ as in Theorem 5. Then, setting $p \approx 0.7$ and $s = C_0 d_n/(0.7^2)$, $S$ is $K$-satisfiable for $c = 9$, with a probability at least $1 - C_1 e^{-sc(0.7)}$. These values of $p$ and $s$ minimize computations while maintaining the guarantees.*

*Proof.* By substituting $s = C_0 d_n/p^2$ into $\mathbb{E}[s']$, one can show that it is optimal to set $p \approx 0.7$, independently from $C_0$ and $d_n$. □

Corollary 1 gives the optimal values of $p$ and $s$ that ensure $K$-satisfiability of a $p$-sparsified sketching matrix while having some complexity reduction. However, the lower bound in Theorem 5 is a sufficient condition, that might be conservative. Looking at the problem of setting $s$ and $p$ from the practitioner point of view, we also provide more aggressive empirical guidelines. Indeed, although this regime is not covered by Theorem 5, experiments show that setting $s$ as for the Gaussian sketch and $p$ smaller than $1/s$ yield very interesting results, see Figure 1(c). Overall, $p$-sparsified sketches ($i$) generalize Gaussian sketches by introducing sparsity as a new degree of freedom, ($ii$) enjoy a regime in which theoretical guarantees are preserved and computations (slightly) accelerated, ($iii$) empirically yield competitive results also in aggressive regimes not covered by theory, thus achieving a wide range of intesting accuracy/computations tradeoffs.

**Related works.** Sparse sketches have been widely studied in the literature, see Clarkson & Woodruff (2017); Nelson & Nguyên (2013); Derezinski et al. (2021). However these sketches are well-suited when applied to sparse matrices (e.g., matrices induced by graphs). In fact, given a matrix $A$, computing $SA$ with these types of sketching has a time complexity of the order of nnz $(A)$, the number of nonzero elements of $A$. Besides, these sketches usually are constructed such that each column has at least one nonzero element (e.g. CountSketch, OSNAP), hence no *decomposition trick* is possible. Regarding kernel methods, since a Gram matrix is typically dense (e.g., with the Gaussian kernel, nnz $(K) = n^2$), and since no decomposition trick can be applied, one has to compute the whole matrix $K$ and store it, such that time and space complexity implied by such sketches are of the order of $n^2$. In practice, we show that we can set $p$ small enough to computationally outperform classical sparse sketches and still obtain similar statistical performance. Note that an important line of research is devoted to improve the statistical performance of Nyström's approximation, either by adaptive sampling (Kumar et al., 2012; Wang & Zhang, 2013; Gittens & Mahoney, 2013), or leverage scores (Alaoui & Mahoney, 2015; Musco & Musco, 2017; Rudi et al., 2018; Chen & Yang, 2021b). We took the opposite route, as $p$-SG sketches are accelerated but statistically degraded versions of the Gaussian sketch.

## 4   Experiments

We now empirically compare the performance of $p$-sparsified sketches against state-of-he-art approaches, namely Nyström approximation (Williams & Seeger, 2001), Gaussian sketch (Yang et al., 2017), Accumulation sketch (Chen & Yang, 2021a), CountSketch (Clarkson & Woodruff, 2017) and Random Fourier Features (Rahimi & Recht, 2007). We chose not to benchmark ROS sketches as CountSketch has equivalent statistical accuracy while being faster to compute. Results reported are averaged over 30 replicates.

### 4.1   Scalar regression

**Robust regression.** We generate a dataset composed of $n = 10,000$ training datapoints: $9,900$ input points drawn i.i.d. from $\mathcal{U}([0_{10}, \mathbb{1}_{10}])$ and 100 other drawn i.i.d. from $\mathcal{N}(1.5\mathbb{1}_{10}, 0.25I_{10})$. The outputs are generated as $y = f^\star(x) + \epsilon$, where $\epsilon \sim \mathcal{N}(0, 1)$ and

$$f^\star(x) = 0.1e^{4x_1} + \frac{4}{1 + e^{-20(x_2 - 0.5)}} + 3x_3 + 2x_4 + x_5,$$

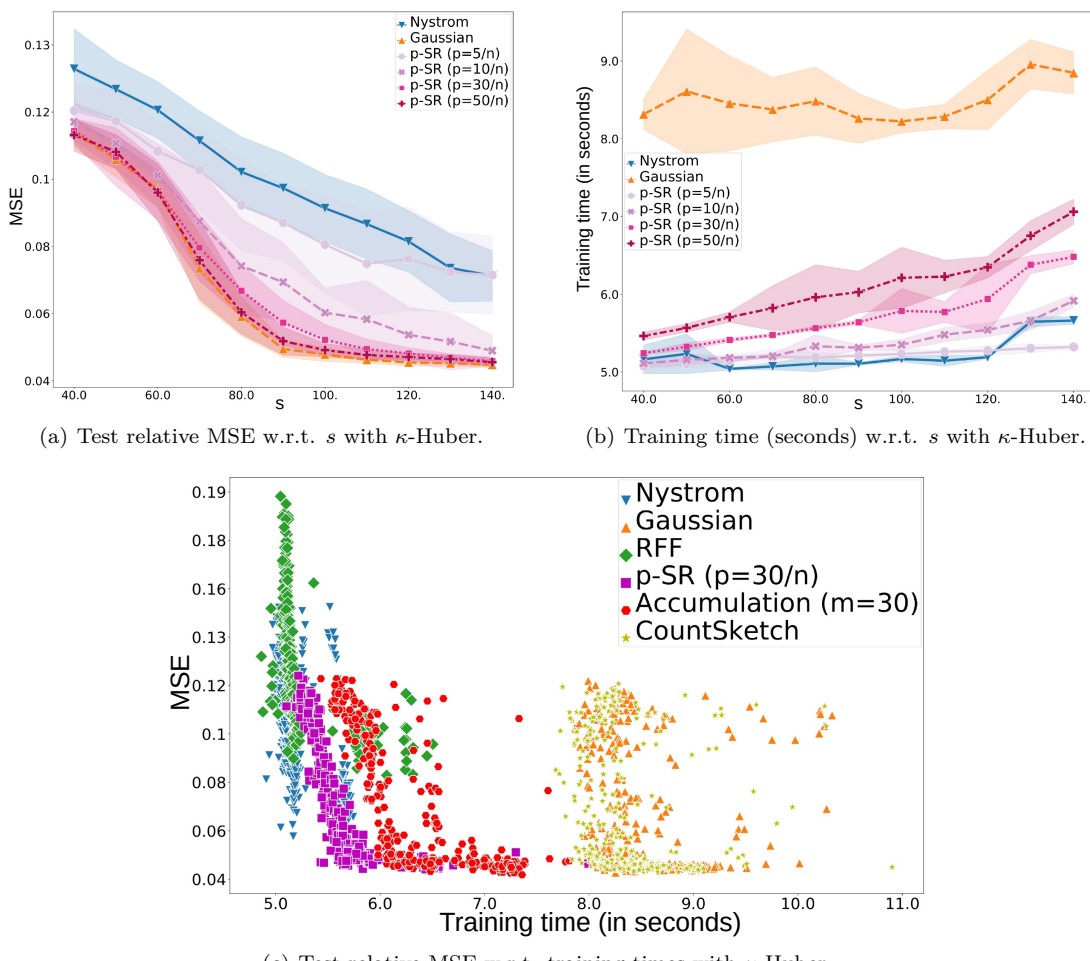

(a) Test relative MSE w.r.t. $s$ with $\kappa$-Huber.

(b) Training time (seconds) w.r.t. $s$ with $\kappa$-Huber.

(c) Test relative MSE w.r.t. training times with $\kappa$-Huber.

Figure 1: Trade-off between Accuracy and Efficiency for $p$-SR sketches with $\kappa$-Huber loss on synthetic dataset.

as introduced in Friedman (1991). We generate a test set of $n_{te} = 10,000$ points in the same way. We use the Gaussian kernel and select its bandwidth —as well as parameters $\lambda_n$ and $\kappa$ (and $\epsilon$ for $\epsilon$-SVR)— via 5-folds cross-validation. We solve this 1D regression problem using the $\kappa$-Huber loss, described in Appendix G. We learn the sketched kernel machines for different values of $s$ (from 40 to 140) and several values of $p$, the probability of being non-null in a $p$-SR sketch. Figure 1(a) presents the test error as a function of the sketch size $s$. Figure 1(b) shows the corresponding computational training time. All methods reduce their test error, measured in terms of the relative Mean Squared Error (MSE) when $s$ increases. Note that increasing $p$ increases both the precision and the training time, as expected. This behaviour recalls the Accumulation sketches, since we observe a form of interpolation between the Nyström and Gaussian approximations. The behaviour of all the different sketched kernel machines is shown in Figure 1(c), where each of them appears as a point (training time, test MSE). We observe that $p$-SR sketches attain the smallest possible error ($MSE \leq 0.05$) at the lowest training time budget (mostly around $5.6 < time < 6.6$). Moreover, $p$-SR sketches obtain a similar precision range as the Accumulation sketches, but for smaller training times (both approaches improve upon CountSketch and Gaussian sketch in that respect). Nyström sketching, which similarly to our approach does not need computing the entire Gram matrix, is fast to compute. The method is however known to be sensitive to the non-homogeneity of the marginal distribution of the input data (Yang et al., 2017, Section 3.3). In contrast, the sub-Gaussian mixing matrix $S_{SG}$ in (14) makes $p$-sparsified sketches more robust, as empirically shown in Figure 1(c). See Appendix H.1 for results on $p$-SG sketches.

## 4.2 Vector-valued regression

Table 1: Test pinball and crossing loss and training times (in seconds) with and without sketching ($s = 50$).

| Dataset | Metrics | w/o Sketch | $20/n_{tr}$-SR | $20/n_{tr}$-SG | Acc. $m = 20$ | CountSketch |
|---|---|---|---|---|---|---|
| | Pinball loss | $\mathbf{51.28 \pm 0.67}$ | $54.75 \pm 0.74$ | $54.78 \pm 0.72$ | $54.73 \pm 0.75$ | $54.60 \pm 0.72$ |
| Boston | Crossing loss | $0.34 \pm 0.13$ | $0.26 \pm 0.08$ | $0.11 \pm 0.07$ | $0.15 \pm 0.07$ | $\mathbf{0.10 \pm 0.05}$ |
| | Training time | $6.97 \pm 0.25$ | $1.43 \pm 0.07$ | $1.38 \pm 0.08$ | $1.48 \pm 0.05$ | $\mathbf{1.23 \pm 0.07}$ |
| | Pinball loss | $2.78$ | $2.66 \pm 0.02$ | $\mathbf{2.64 \pm 0.02}$ | $2.67 \pm 0.03$ | $2.65 \pm 0.02$ |
| otoliths | Crossing loss | $\mathbf{5.18}$ | $5.46 \pm 0.06$ | $5.43 \pm 0.05$ | $5.46 \pm 0.06$ | $5.44 \pm 0.05$ |
| | Training time | $606.8$ | $20.4 \pm 0.5$ | $\mathbf{20.0 \pm 0.3}$ | $22.1 \pm 0.4$ | $20.9 \pm 0.3$ |

**Joint quantile regression.** We choose the quantile levels as follows $\tau = (0.1, 0.3, 0.5, 0.7, 0.9)$. We apply a subgradient algorithm to minimize the pinball loss described in Appendix G with ridge regularization and a kernel $\mathcal{K} = kM$ with $M$ discussed in Example 1, and $k$ a Gaussian kernel. We select regularisation parameter $\lambda_n$ and bandwidth of kernel $\sigma^2$ via a 5-fold cross-validation. We showcase the behaviour of the proposed algorithm for Joint Sketched Quantile Regression on two datasets: the Boston Housing dataset (Harrison Jr & Rubinfeld, 1978), composed of 506 data points devoted to house price prediction, and the Fish Otoliths dataset (Moen et al., 2018; Ordoñez et al., 2020), dedicated to fish age prediction from images of otoliths (calcium carbonate structures), composed of a train and test sets of size 3780 and 165 respectively. The results are averages over 10 random $70\% - 30\%$ train-test splits for Boston dataset. For the Otoliths dataset we kept the initial given train-test split. The results are reported in Table 1. Sketching allows for a massive reduction of the training times while preserving the statistical performances. As a comparison, according to the results of Sangnier et al. (2016), the best benchmark result for the Boston dataset in terms of test pinball loss is 47.4, while best test crossing loss is 0.48, which shows that our implementation does not compete in terms of quantile prediction but preserves the non-crossing property.

Table 2: ARRMSE and training times (in sec) with square loss and $s = 100$ when using Sketching.

| Dataset | Metrics | w/o Sketch | $20/n_{tr}$-SR | $20/n_{tr}$-SG | Acc. $m = 20$ | CountSketch |
|---|---|---|---|---|---|---|
| rf1 | ARRMSE | $\mathbf{0.575}$ | $0.584 \pm 0.003$ | $0.583 \pm 0.003$ | $0.592 \pm 0.001$ | $\mathbf{0.575 \pm 0.0005}$ |
| | Training time | $1.73$ | $\mathbf{0.22 \pm 0.025}$ | $0.25 \pm 0.005$ | $0.60 \pm 0.0004$ | $0.66 \pm 0.013$ |
| rf2 | ARRMSE | $\mathbf{0.578}$ | $0.671 \pm 0.009$ | $0.656 \pm 0.006$ | $0.796 \pm 0.006$ | $0.715 \pm 0.011$ |
| | Training time | $1.77$ | $0.28 \pm 0.003$ | $\mathbf{0.27 \pm 0.003}$ | $0.82 \pm 0.003$ | $0.62 \pm 0.001$ |
| scm1d | ARRMSE | $\mathbf{0.418}$ | $0.422 \pm 0.002$ | $0.423 \pm 0.001$ | $0.423 \pm 0.001$ | $0.420 \pm 0.001$ |
| | Training time | $9.36$ | $\mathbf{0.45 \pm 0.022}$ | $\mathbf{0.45 \pm 0.019}$ | $0.86 \pm 0.006$ | $2.49 \pm 0.035$ |
| scm20d | ARRMSE | $0.755$ | $0.754 \pm 0.003$ | $0.754 \pm 0.003$ | $\mathbf{0.753 \pm 0.001}$ | $0.754 \pm 0.002$ |
| | Training time | $6.16$ | $\mathbf{0.38 \pm 0.016}$ | $\mathbf{0.38 \pm 0.017}$ | $0.70 \pm 0.032$ | $1.91 \pm 0.047$ |

**Multi-output regression.** We finally conducted experiments on multi-output kernel ridge regression. We used decomposable kernels, and took the largest datasets introduced in Spyromitros-Xioufis et al. (2016). They consist in four datasets, divided in two groups: River Flow (rf1 and rf2) both composed of 4108 training data, and Supply Chain Management (scm1d and scm20d) composed of 8145 and 7463 training data respectively (more details and additional results can be found in Appendix H.2). We compare our non-sketched decomposable matrix-valued kernel machine with the sketched version. For the sake of conciseness, we only report here the Average Relative Root Mean Squared Error (ARRMSE), see Table 2 and Appendix H.2. For all datasets, sketching shows strong computational improvements while maintaining the accuracy of non-sketched approaches.

Note that for both joint quantile regression and multi-output regression the results obtained after sketching (no matter the sketch chosen) are almost the same as that attained without sketching. It might be explained by two factors. First, the datasets studied have relatively small training sizes (from 354 training data for Boston to 8145 for scm1d). Second, predicting jointly multiple outputs is a complex task, so that it appears more natural to obtain less differences and variance using various types of sketches (or no sketch). However, in all cases sketching induces a huge time saver.

## 5 Conclusion

We proposed excess-risk bounds for sketched kernel machines in the context of Lipschitz-continuous losses, with results valid for both scalar and matrix-valued kernels. We introduced a novel sketching scheme that leverages the good empirical statistical guarantees of the Gaussian Sketching while combining them with the low cost of Nyström sketching. Numerical experiments show that this novel scheme opens the door to many applications beyond the squared loss. Improvements on multi-output regression can certainly be obtained by applying low-rank considerations in the output space as well.

### Acknowledgments

The authors thank Olivier Fercoq for insightful discussions. This work was supported by the Télécom Paris research chair on Data Science and Artificial Intelligence for Digitalized Industry and Services (DSAIDIS) and by the French National Research Agency (ANR) through the ANR-18-CE23-0014 APi project.

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

## A   Technical Proofs

In this section are gathered all the technical proofs of the results stated in the article.

**Notation.**   We recall that we assume that training data $(x_i, y_i)_{i=1}^n$ are i.i.d. realisations sampled from a joint probability density $P(x, y)$. We define

$$\mathbb{E}_n[\ell_f] = \frac{1}{n}\sum_{i=1}^n \ell(f(x_i), y_i),$$

$$\mathbb{E}[\ell_f] = \mathbb{E}_P[\ell(f(X), Y].$$

For a class of functions $F$, the empirical Rademacher complexity (Bartlett & Mendelson, 2003) is defined as

$$\hat{R}_n(F) = \mathbb{E}\left[\sup_{f \in F}\left|\frac{2}{n}\sum_{i=1}^n \sigma_i f(x_i)\right| \Big| x_1, \ldots, x_n\right],$$

where $\sigma_1, \ldots, \sigma_n$ are independent Rademacher random variables such that $\mathbb{P}\{\sigma_i = 1\} = \mathbb{P}\{\sigma_i = -1\} = 1/2$. The corresponding Rademacher complexity is then defined as the expectation of the empirical Rademacher complexity

$$R_n(F) = \mathbb{E}\left[\hat{R}_n(F)\right].$$

### A.1   Proof of Theorem 2

We first prove the first inequality in Theorem 2 for generic Lipschitz losses.

*Proof.* The proof follows that of Li et al. (2021, Theorem 3). We decompose the expected learning risk as

$$\mathbb{E}[\ell_{\tilde{f}_s}] - \mathbb{E}[\ell_{f_{\mathcal{H}_k}}] = \mathbb{E}[\ell_{\tilde{f}_s}] - \mathbb{E}_n[\ell_{\tilde{f}_s}] + \mathbb{E}_n[\ell_{\tilde{f}_s}] - \mathbb{E}_n[\ell_{f_{\mathcal{H}_k}}] + \mathbb{E}_n[\ell_{f_{\mathcal{H}_k}}] - \mathbb{E}[\ell_{f_{\mathcal{H}_k}}]. \tag{7}$$

We then use Bartlett & Mendelson (2003, Theorem 8) to bound $\mathbb{E}[\ell_{\tilde{f}_s}] - \mathbb{E}_n[\ell_{\tilde{f}_s}]$ and $\mathbb{E}_n[\ell_{f_{\mathcal{H}_k}}] - \mathbb{E}[\ell_{f_{\mathcal{H}_k}}]$.

**Lemma 1.** *(Bartlett & Mendelson, 2003, Theorem 8) Let $\{x_i, y_i\}_{i=1}^n$ be i.i.d samples from $P$ and let $\mathcal{H}$ be the space of functions mapping from $\mathcal{X}$ to $\mathbb{R}$. Denote a loss function with $l : \mathcal{Y} \times \mathbb{R} \to [0, 1]$ and recall the learning risk function for all $f \in \mathcal{H}$ is $\mathbb{E}[l_f]$, together with the corresponding empirical risk function $\mathbb{E}_n[l_f] = (1/n)\sum_{i=1}^n l(y_i, f(x_i))$. Then, for a sample of size $n$, for all $f \in \mathcal{H}$ and $\delta \in (0, 1)$, with probability $1 - \delta/2$, we have that*

$$\mathbb{E}[l_f] \leq \mathbb{E}_n[l_f] + R_n(l \circ \mathcal{H}) + \sqrt{\frac{8\log(4/\delta)}{n}}$$

*where $l \circ \mathcal{H} = \{(x, y) \to l(y, f(x)) - l(y, 0) \mid f \in \mathcal{H}\}$.*

Thus, since $\tilde{f}_s$ lies in the unit ball $\mathcal{B}(\mathcal{H}_k)$ of $\mathcal{H}_k$ by Assumption 2, we obtain thanks to the above lemma, with a probability at least $1 - \delta$

$$\mathbb{E}\left[\ell_{\tilde{f}_s}\right] - \mathbb{E}_n\left[\ell_{\tilde{f}_s}\right] \leq R_n\left(\ell \circ \mathcal{B}(\mathcal{H}_k)\right) + \sqrt{\frac{8\log(2/\delta)}{n}}.$$

Then, by the Lipschitz continuity of $\ell$ (Assumption 3) and point 4 of Theorem 12 from Bartlett & Mendelson (2003), we have that

$$R_n\left(\ell \circ \mathcal{B}(\mathcal{H}_k)\right) \leq 2LR_n\left(\mathcal{B}(\mathcal{H}_k)\right).$$

Finally, Assumption 4 combined with Lemma 22 from Bartlett & Mendelson (2003) then yields

$$R_n\left(\mathcal{B}(\mathcal{H}_k)\right) \leq \frac{2}{n}\sqrt{\sum_{i=1}^n k(x_i, x_i)} \leq 2\sqrt{\frac{\kappa}{n}}.$$

As a consequence, we obtain

$$\mathbb{E}\left[\ell_{\tilde{f}_s}\right] - \mathbb{E}_n\left[\ell_{\tilde{f}_s}\right] \leq \frac{4L\sqrt{\kappa}}{\sqrt{n}} + \sqrt{\frac{8\log(4/\delta)}{n}}, \tag{15}$$

and the exact same result applies to $\mathbb{E}_n[\ell_{f_{\mathcal{H}_k}}] - \mathbb{E}[\ell_{f_{\mathcal{H}_k}}]$, by Assumption 2 and the opposite side of Lemma 1.

We now focus on the last quantity to bound. Let $\mathcal{H}_S = \left\{f = \sum_{i=1}^n \left[S^\top\gamma\right]_i k\left(\cdot, x_i\right) \mid \gamma \in \mathbb{R}^s\right\}$. By Assumptions 2 and 3 and Jensen's inequality we have

$$\begin{aligned}
\mathbb{E}_n\left[\ell_{\tilde{f}_s}\right] - \mathbb{E}_n\left[\ell_{f_{\mathcal{H}_k}}\right] &= \frac{1}{n}\sum_{i=1}^n \ell\left(\tilde{f}_s(x_i), y_i\right) - \frac{1}{n}\sum_{i=1}^n \ell\left(f_{\mathcal{H}_k}(x_i), y_i\right) \\
&\leq \frac{1}{n}\sum_{i=1}^n \ell\left(\tilde{f}_s(x_i), y_i\right) + \frac{\lambda_n}{2}\left\|\tilde{f}_s\right\|_{\mathcal{H}_k}^2 - \frac{1}{n}\sum_{i=1}^n \ell\left(f_{\mathcal{H}_k}(x_i), y_i\right) \\
&= \inf_{\substack{f \in \mathcal{H}_S \\ \|f\|_{\mathcal{H}_k} \leq 1}} \frac{1}{n}\sum_{i=1}^n \ell\left(f(x_i), y_i\right) - \frac{1}{n}\sum_{i=1}^n \ell\left(f_{\mathcal{H}_k}(x_i), y_i\right) + \frac{\lambda_n}{2}\|f\|_{\mathcal{H}_k}^2 \\
&\leq \inf_{\substack{f \in \mathcal{H}_S \\ \|f\|_{\mathcal{H}_k} \leq 1}} \frac{L}{n}\sum_{i=1}^n \left|f\left(x_i\right) - f_{\mathcal{H}_k}\left(x_i\right)\right| + \frac{\lambda_n}{2} \\
&\leq L \inf_{\substack{f \in \mathcal{H}_S \\ \|f\|_{\mathcal{H}_k} \leq 1}} \sqrt{\frac{1}{n}\sum_{i=1}^n \left|f\left(x_i\right) - f_{\mathcal{H}_k}\left(x_i\right)\right|^2} + \frac{\lambda_n}{2} \\
&= L\sqrt{\inf_{\substack{f \in \mathcal{H}_S \\ \|f\|_{\mathcal{H}_k} \leq 1}} \frac{1}{n}\left\|f^X - f_{\mathcal{H}_k}^X\right\|_2^2} + \frac{\lambda_n}{2},
\end{aligned}$$

where, for any $f \in \mathcal{H}_k$, $f^X = (f(x_1), \ldots, f(x_n)) \in \mathbb{R}^n$. Let $\tilde{f}_s^R = \sum_{i=1}^n \left[S^\top\tilde{\gamma}^R\right]_i k\left(\cdot, x_i\right)$, where $\tilde{\gamma}^R$ is a solution to

$$\inf_{\gamma \in \mathbb{R}^s} \frac{1}{n}\left\|KS^\top\gamma - f_{\mathcal{H}_k}^X\right\|_2^2 + \lambda_n \gamma^\top SKS^\top\gamma.$$

It is easy to check that $\tilde{f}_s^R$ is also a solution to

$$\inf_{\substack{f \in \mathcal{H}_S \\ \|f\|_{\mathcal{H}_k} \leq \|\tilde{f}_s^R\|_{\mathcal{H}_k}}} \frac{1}{n}\left\|f^X - f_{\mathcal{H}_k}^X\right\|_2^2.$$

Since we have $\|\tilde{f}_s^R\|_{\mathcal{H}_k} \leq 1$ by Assumption 2, it holds

$$\begin{aligned}
\inf_{\substack{f \in \mathcal{H}_S \\ \|f\|_{\mathcal{H}_k} \leq 1}} \frac{1}{n}\left\|f^X - f_{\mathcal{H}_k}^X\right\|_2^2 &\leq \inf_{\substack{f \in \mathcal{H}_S \\ \|f\|_{\mathcal{H}_k} \leq \|\tilde{f}_s^R\|_{\mathcal{H}_k}}} \frac{1}{n}\left\|f^X - f_{\mathcal{H}_k}^X\right\|_2^2 \\
&= \inf_{\gamma \in \mathbb{R}^s} \frac{1}{n}\left\|KS^\top\gamma - f_{\mathcal{H}_k}^X\right\|_2^2 + \lambda_n \gamma^\top SKS^\top\gamma.
\end{aligned}$$

As a consequence,

$$\mathbb{E}_n\left[\ell_{\tilde{f}_s}\right] - \mathbb{E}_n\left[\ell_{f_{\mathcal{H}_k}}\right] \leq L\sqrt{\inf_{\gamma \in \mathbb{R}^s} \frac{1}{n}\left\|KS^\top\gamma - f_{\mathcal{H}_k}^X\right\|_2^2 + \lambda_n \gamma^\top SKS^\top\gamma} + \frac{\lambda_n}{2}.$$

Finally, since $S$ is a $K$-satisfiable sketch matrix, using Lemma 2 from Yang et al. (2017),

$$\mathbb{E}_n\left[\ell_{\tilde{f}_s}\right] - \mathbb{E}_n\left[\ell_{f_{\mathcal{H}_k}}\right] \leq LC\sqrt{\lambda_n + \delta_n^2} + \frac{\lambda_n}{2}, \tag{16}$$

where $C = 1 + \sqrt{6}c$ and $c$ is a universal constant coming from $K$-satisfiable property. The desired bound is obtained by combining Equations (7), (15) and (16). $\qquad\square$

We now give the proof of the second claim i.e., the excess risk bound for kernel ridge regression.

*Proof.* We now assume that the outputs are bounded, hence, without loss of generality, $\mathcal{Y} \subset [0,1]$. First, we prove Lipschitz-continuity of the square loss under Assumptions 2 and 4. Let $\mathcal{H}_k(\mathcal{X}) = \{f(x) : f \in \mathcal{H}_k, x \in \mathcal{X}\}$ and $g : z \in \mathcal{H}_k(\mathcal{X}) \mapsto \frac{1}{2}(z-y)^2$, for $y \in [0,1]$. Hence, $g'(z) = z - y$ and by Assumptions 2 and 4

$$|g'(z)| \leq |z| + |y| \leq |f(x)| + 1 \leq |\langle f, k(\cdot, x)\rangle_{\mathcal{H}_k}| + 1 \leq \|f\|_{\mathcal{H}_k} \kappa^{1/2} + 1 = \kappa^{1/2} + 1,$$

for some $f \in \mathcal{H}_k$ and $x \in \mathcal{X}$ since $z \in \mathcal{H}_k(\mathcal{X})$. As a consequence, we obtain that $g$ is $(\kappa^{1/2} + 1)$-Lipschitz, i.e.

$$|\ell(f(x), y) - \ell(f'(x'), y)| \leq \left(\kappa^{1/2} + 1\right) |f(x) - f'(x')|.$$

We can then obtain the same generalisation bounds as above. Finally, looking at the approximation term,

$$
\begin{aligned}
\mathbb{E}_n\left[\ell_{\tilde{f}_s}\right] - \mathbb{E}_n\left[\ell_{f_{\mathcal{H}_k}}\right] &= \frac{1}{2n}\left\|\tilde{f}_s^X - Y\right\|_2^2 - \frac{1}{2n}\left\|f_{\mathcal{H}_k}^X - Y\right\|_2^2 \\
&\leq \frac{1}{2n}\left\|\tilde{f}_s^X - f_{\mathcal{H}_k}^X\right\|_2^2 \\
&\leq \inf_{\substack{f \in \mathcal{H}_S \\ \|f\|_{\mathcal{H}_k} \leq 1}} \frac{1}{2n}\left\|f^X - f_{\mathcal{H}_k}^X\right\|_2^2 + \frac{\lambda_n}{2} \\
&\leq \inf_{\gamma \in \mathbb{R}^s} \frac{1}{n}\left\|KS^\top\gamma - f_{\mathcal{H}_k}^X\right\|_2^2 + \lambda_n \gamma^\top SKS^\top \gamma + \frac{\lambda_n}{2} \\
&\leq \left(C^2 + \frac{1}{2}\right)\lambda_n + C^2\delta_n^2.
\end{aligned}
$$

We obtain the same bound as above without the square root since we do not need to invoke Jensen inequality as the studied loss is the square loss. Gathering all arguments, we obtain last inequality in Theorem 2. $\square$

## A.2 Refined analysis in the scalar case

As said in Remark 1, and similarly to Li et al. (2021), we can conduct a refined analysis, leading to faster convergence rates for the generalization errors, with the following additional assumption.

**Assumption 6.** *There is a constant $B$ such that, for all $f \in \mathcal{H}_k$ we have*

$$\mathbb{E}[f - f_{\mathcal{H}_k}]^2 \leq B\,\mathbb{E}\left[\ell_f - \ell_{f_{\mathcal{H}_k}}\right]. \tag{17}$$

It has be shown that many loss functions satisfy this assumption such as Hinge loss (Steinwart & Christmann, 2008b; Bartlett et al., 2006), truncated quadratic or sigmoid loss (Bartlett et al., 2006). Under Assumptions 1 to 5 and 6, the following result holds:

**Theorem 6.** *We define, for $\delta \in (0,1)$, the following sub-root function $\hat{\psi}_n$*

$$\hat{\psi}_n(r) = 2LC_1\left(\frac{2}{n}\sum_{i=1}^n \min\{b_2 r, \mu_i\}\right)^{1/2} + \frac{C_2}{n}\log\frac{1}{\delta},$$

*and let $\hat{r}_{\mathcal{H}_k}^*$ be the fixed point of $\hat{\psi}_n$, i.e., $\hat{\psi}_n\left(\hat{r}_{\mathcal{H}_k}^*\right) = \hat{r}_{\mathcal{H}_k}^*$. Then, we have for all $D > 1$ and $\delta \in (0,1)$ with probability greater than $1 - \delta$,*

$$\mathbb{E}\left[\ell_{\tilde{f}_s}\right] \leq \mathbb{E}\left[\ell_{f_{\mathcal{H}_k}}\right] + \frac{D}{D-1}\left(LC\sqrt{\lambda_n + \delta_n^2} + \frac{\lambda_n}{2}\right) + \frac{12D}{B}\hat{r}_{\mathcal{H}_k}^\star + \frac{2C_3}{n}\log\frac{1}{\delta}, \tag{18}$$

*where $C$ is as in Theorem 2 and $C_1$, $C_2$, $C_3$ and $b_2$ are some constants and $\hat{r}_{\mathcal{H}_k}^\star$ can be upper bounded by*

$$\hat{r}_{\mathcal{H}_k}^\star \leq \min_{0 \leq h \leq n}\left(b_0\frac{h}{n} + \sqrt{\frac{1}{n}\sum_{i>h}\mu_i}\right),$$

*where $B$ and $b_0$ are some constants.*

Hence, we see that, in order to obtain faster learning rates than Theorem 2 as Li et al. (2021), we need to replace $\delta_n^2$ by $\hat{r}_{\mathcal{H}_k}^{\star 2}$. However, according to the expression of $\hat{\psi}_n$ and its dependencies to non-explicit constants, it appears very difficult to prove that $\left(\frac{1}{n}\sum_{i=1}^n \min(\hat{r}_{\mathcal{H}_k}^{\star 2}, \mu_i)\right)^{1/2} \leq \hat{r}_{\mathcal{H}_k}^{\star 2}$, which is a necessary condition to prove that a sketch matrix $S$ is $K$-satisfiable. We still prove the above result following the proof of Theorem 4 in Li et al. (2021), and leave as an open problem to find faster rates than $\delta_n$.

*Proof.* We use the decomposition of the expected learning risk from Li et al. (2021)

$$
\mathbb{E}[\ell_{\tilde{f}_s}] - \mathbb{E}[\ell_{f_{\mathcal{H}_k}}] = \mathbb{E}[\ell_{\tilde{f}_s}] - \frac{D}{D-1}\mathbb{E}_n[\ell_{\tilde{f}_s}]
$$
$$
+ \frac{D}{D-1}\left(\mathbb{E}_n[\ell_{\tilde{f}_s}] - \mathbb{E}_n[\ell_{f_{\mathcal{H}_k}}]\right)
$$
$$
+ \frac{D}{D-1}\mathbb{E}_n[\ell_{f_{\mathcal{H}_k}}] - \mathbb{E}[\ell_{f_{\mathcal{H}_k}}],
$$

for $D > 1$. The generalization errors can be bounded as in Li et al. (2021) by Theorem 6. The approximation error is bounded using (16). $\qquad\square$

## A.3 Proof of Theorem 4

Here, the proof uses the same decomposition of the excess risk (Equation (7)) as in single output settings. Since some works (Maurer, 2016) exist to easily extend generalisation bounds of functions in scalar-valued RKHS to functions in vector-valued RKHS, the main challenge here is to derive an approximation error for the multiple output settings. Hence, let us first state a needed intermediate results that we will prove later.

**Lemma 2.** *For all $f \in \mathcal{H}_{\mathcal{K}}$, such that $\|f\|_{\mathcal{H}_{\mathcal{K}}} \leq 1$, we have $z^\top \left(K^{-1} \otimes M^{-1}\right) z \leq 1$, where $z = \left(f(x_1)^\top, \ldots, f(x_n)^\top\right)^\top \in \mathbb{R}^{nd}$.*

We are now equipped to state the main result that generalises Lemma 2 from Yang et al. (2017).

**Lemma 3.** *Let $Z^\star = (f^\star(x_1), \ldots, f^\star(x_n))^\top \in \mathbb{R}^{n \times d}$ for any $f^\star \in \mathcal{H}_{\mathcal{K}}$ such that $\|f^\star\|_{\mathcal{H}_{\mathcal{K}}} \leq 1$, where $\mathcal{K} = kM$, and $S \in \mathbb{R}^{s \times n}$ a $K$- satisfiable matrix. Then we have*

$$
\inf_{\Gamma \in \mathbb{R}^{s \times d}} \frac{1}{n}\|KS^\top \Gamma M - Z^\star\|_F^2 + \lambda_n \operatorname{Tr}\left(KS^\top \Gamma M \Gamma^\top S\right) \leq C^2 \left(\|M\|_{\mathrm{op}}\delta_n^2 + \lambda_n\right), \tag{19}
$$

*where $C = 1 + \sqrt{6}c$ and $c$ is the universal constant from Definition 3.*

*Proof.* We adapt the proof of Lemma 2 from Yang et al. (2017) to the multidimensional case. If we are able to find a $\Gamma \in \mathbb{R}^{s \times d}$ such that

$$
\frac{1}{n}\|KS^\top \Gamma M - Z^\star\|_F^2 + \lambda_n \operatorname{Tr}\left(KS^\top \Gamma M \Gamma^\top S\right) \leq C^2 \left(\|M\|_{\mathrm{op}}\delta_n^2 + \lambda_n\right),
$$

then in particular it also holds true for the minimizer. We recall the eigendecompositions $\frac{1}{n}K = K_{norm} = UDU^\top$ and $M = V\Delta V^\top$. Then the above problem rewrites as

$$
\|D\tilde{S}^\top \Gamma V\Delta - \Theta^\star\|_F^2 + \lambda_n \operatorname{Tr}\left(\tilde{S}D\tilde{S}^\top \Gamma M\Gamma\right) \leq C^2 \left(\|M\|_{\mathrm{op}}\delta_n^2 + \lambda_n\right),
$$

where $\tilde{S} = SU$ and $\Theta^\star = \frac{1}{n^{1/2}}U^\top Z^\star V$. We can rewrite $\theta^\star = (\Theta_{1:}^\star, \ldots, \Theta_{n:}^\star)^\top = \frac{1}{n^{1/2}}\left(U^\top \otimes V^\top\right)z^\star$, hence $\|\left(D^{-1/2} \otimes \Delta^{-1/2}\right)\theta^\star\|_2^2 = z^{\star\top}\left(K^{-1} \otimes M^{-1}\right)z^\star$, with $z^\star = (Z_{1:}^\star, \ldots, Z_{n:}^\star)^\top = \left(f^\star(x_1)^\top, \ldots, f^\star(x_n)^\top\right)^\top$. By Lemma 2, we have that $\|\left(D^{-1/2} \otimes \Delta^{-1/2}\right)\theta^\star\|_2 \leq 1$, and using the notation $\gamma = (\Gamma_{1:}, \ldots, \Gamma_{s:})^\top \in \mathbb{R}^{sd}$, we can rewrite the above problem as finding a $\gamma$ such that

$$
\|\theta^\star - \left(D\tilde{S}^\top \otimes \Delta V^\top\right)\gamma\|_2^2 + \lambda_n\gamma^\top\left(\tilde{S}D\tilde{S}^\top \otimes M\right)\gamma \leq C^2 \left(\|M\|_{\mathrm{op}}\delta_n^2 + \lambda_n\right).
$$

As in (Yang et al., 2017), we partition vector $\theta^\star \in \mathbb{R}^{nd}$ into two sub-vectors, namely $\theta_1^\star \in \mathbb{R}^{d_n d}$ and $\theta_2^\star \in \mathbb{R}^{(n-d_n)d}$, the diagonal matrix $D$ into two blocks $D_1 \in \mathbb{R}^{d_n \times d_n}$ and $D_2 \in \mathbb{R}^{(n-d_n) \times (n-d_n)}$ and finally, under the condition $s > d_n$, we let $\tilde{S}_1 \in \mathbb{R}^{s \times d_n}$ and $\tilde{S}_2 \in \mathbb{R}^{s \times (n-d_n)}$ denote the left and right block of $\tilde{S}$ respectively. By the $K$-satisfiablility of $S$ we have

$$\|\tilde{S}_1^\top \tilde{S}_1 - I_{d_n}\|_{\mathrm{op}} \leq \frac{1}{2} \quad \text{and} \quad \|\tilde{S}_2 D_2^{1/2}\|_{\mathrm{op}} \leq c\delta_n^2 \,. \tag{20}$$

By the first inequality, we have that $\tilde{S}_1^\top \tilde{S}_1$ is invertible. In fact, assuming that there exists $x \in \mathbb{R}^{d_n}$ such that $\|x\|_2 = 1$ and $\tilde{S}_1^\top \tilde{S}_1 X = 0$, then $\left\|(\tilde{S}_1^\top \tilde{S}_1 - I_{d_n})x\right\| = 1 > \frac{1}{2}$. Then, we can define

$$\hat{\gamma} = \left( \tilde{S}_1 \left( \tilde{S}_1^\top \tilde{S}_1 \right)^{-1} D_1^{-1} \otimes V\Delta^{-1} \right) \theta_1^\star \,. \tag{21}$$

Hence,

$$\left\| \theta^\star - (D\tilde{S}^\top \otimes \Delta V^\top)\hat{\gamma} \right\|_2^2 = \left\| \theta_1^\star - (D_1 \tilde{S}_1^\top \otimes \Delta V^\top)\hat{\gamma} \right\|_2^2 + \left\| \theta_2^\star - (D_2 \tilde{S}_2^\top \otimes \Delta V^\top)\hat{\gamma} \right\|_2^2,$$

and we have

$$\begin{aligned}
\left\| \theta_1^\star - (D_1 \tilde{S}_1^\top \otimes \Delta V^\top)\hat{\gamma} \right\|_2^2 &= \left\| \theta_1^\star - (D_1 \tilde{S}_1^\top \otimes \Delta V^\top)\left( \tilde{S}_1 (\tilde{S}_1^\top \tilde{S}_1)^{-1} D_1^{-1} \otimes V\Delta^{-1} \right)\theta_1^\star \right\|_2^2 \\
&= \left\| \theta_1^\star - \left( D_1 \tilde{S}_1^\top \tilde{S}_1 (\tilde{S}_1^\top \tilde{S}_1)^{-1} D_1^{-1} \otimes \Delta V^\top V\Delta^{-1} \right)\theta_1^\star \right\|_2^2 \\
&= \left\| \theta_1^\star - \theta_1^\star \right\|_2^2 \\
&= 0 \,,
\end{aligned}$$

and

$$\begin{aligned}
&\left\| \theta_2^\star - \left( D_2 \tilde{S}_2^\top \otimes \Delta V^\top \right) \hat{\gamma} \right\|_2 \\
&= \left\| \theta_2^\star - \left( D_2 \tilde{S}_2^\top \tilde{S}_1 \left( \tilde{S}_1^\top \tilde{S}_1 \right)^{-1} D_1^{-1} \otimes I_p \right) \theta_1^\star \right\|_2 \\
&\leq \|\theta_2^\star\|_2 + \left\| \left( D_2 \tilde{S}_2^\top \tilde{S}_1 \left( \tilde{S}_1^\top \tilde{S}_1 \right)^{-1} D_1^{-1/2} D_1^{-1/2} \otimes \Delta^{1/2}\Delta^{-1/2} \right) \theta_1^\star \right\|_2 \\
&= \|\theta_2^\star\|_2 + \left\| \left( D_2 \tilde{S}_2^\top \tilde{S}_1 \left( \tilde{S}_1^\top \tilde{S}_1 \right)^{-1} D_1^{-1/2} \otimes \Delta^{1/2} \right)\left( D_1^{-1/2} \otimes \Delta^{-1/2} \right) \theta_1^\star \right\|_2 \\
&\leq \|\theta_2^\star\|_2 + \left\| \left( D_2 \tilde{S}_2^\top \tilde{S}_1 \left( \tilde{S}_1^\top \tilde{S}_1 \right)^{-1} D_1^{-1/2} \otimes \Delta^{1/2} \right) \right\|_{\mathrm{op}} \left\| \left( D_1^{-1/2} \otimes \Delta^{-1/2} \right) \theta_1^\star \right\|_2 \\
&= \|\theta_2^\star\|_2 + \left\| D_2 \tilde{S}_2^\top \tilde{S}_1 \left( \tilde{S}_1^\top \tilde{S}_1 \right)^{-1} D_1^{-1/2} \right\|_{\mathrm{op}} \left\| \Delta^{1/2} \right\|_{\mathrm{op}} \left\| \left( D_1^{-1/2} \otimes \Delta^{-1/2} \right) \theta_1^\star \right\|_2 \\
&\leq \|\theta_2^\star\|_2 + \left\| D_2^{1/2} \right\|_{\mathrm{op}} \left\| \tilde{S}_2 D_2^{1/2} \right\|_{\mathrm{op}} \left\| \tilde{S}_1 \right\|_{\mathrm{op}} \left\| \left( \tilde{S}_1^\top \tilde{S}_1 \right)^{-1} \right\|_{\mathrm{op}} \left\| D_1^{-1/2} \right\|_{\mathrm{op}} \left\| \Delta^{1/2} \right\|_{\mathrm{op}} \left\| \left( D_1^{-1/2} \otimes \Delta^{-1/2} \right) \theta_1^\star \right\|_2 \,.
\end{aligned} \tag{22}$$

We now bound all terms involved in (22). Since $\| \left( D^{-1/2} \otimes \Delta^{-1/2} \right) \theta^\star \|_2 \leq 1$, then $\| \left( D_1^{-1/2} \otimes \Delta^{-1/2} \right) \theta_1^\star \|_2 \leq 1$ and,

$$\begin{aligned}
\|\theta_2^\star\|_2^2 &= \sum_{i=1}^{d} \sum_{j=d_n+1}^{n} \left( \theta_{2_{ji}}^\star \right)^2 \\
&\leq \delta_n^2 \|M\|_{\mathrm{op}} \sum_{i=1}^{d} \frac{1}{\Delta_{ii}} \sum_{j=d_n+1}^{n} \frac{\left( \theta_{2_{ji}}^\star \right)^2}{\mu_j} \\
&\leq \delta_n^2 \|M\|_{\mathrm{op}} \sum_{i=1}^{d} \sum_{j=1}^{n} \frac{\left( \theta_{2_{ji}}^\star \right)^2}{\mu_j \Delta_{ii}} \\
&= \delta_n^2 \|M\|_{\mathrm{op}} \left\| \left( D^{-1/2} \otimes \Delta^{-1/2} \right) \theta^\star \right\|_2^2 \\
&\leq \delta_n^2 \|M\|_{\mathrm{op}},
\end{aligned}$$

since $\mu_j \leq \delta_n^2$, for all $j \geq d_n + 1$ and $\Delta_{ii} \leq \|M\|_{\text{op}}$ for all $1 \leq i \leq d$. Moreover, since $\|\tilde{S}_1^\top \tilde{S}_1 - I_{d_n}\|_{\text{op}} \leq \frac{1}{2}$, $\|\tilde{S}_1^\top \tilde{S}_1\|_{\text{op}} \leq \frac{3}{2}$, then $\|\tilde{S}_1\|_{\text{op}} \leq \sqrt{\frac{3}{2}}$. Besides, for all $x \in \mathbb{R}^{d_n}$ such that $\|x\|_2 = 1$, we have

$$\big| \|\tilde{S}_1^\top \tilde{S}_1 x\|_2 - 1 \big| = \big| \|\tilde{S}_1^\top \tilde{S}_1 x\|_2 - \|x\|_2 \big| \leq \left\| \left( \tilde{S}_1^\top \tilde{S}_1 - I_{d_n} \right) x \right\|_2 \leq \frac{1}{2},$$

Then, we obtain that $\|\tilde{S}_1^\top \tilde{S}_1 x\|_2 - 1 \geq -\frac{1}{2}$ and then $\|\tilde{S}_1^\top \tilde{S}_1 x\|_2 \geq \frac{1}{2}$, taking $x$ the eigenvector of $\tilde{S}_1^\top \tilde{S}_1$ corresponding to its smallest eigenvalue, we obtain that $\| \left( \tilde{S}_1^\top \tilde{S}_1 \right)^{-1} \|_{\text{op}}^{-1} \geq \frac{1}{2}$, and finally $\| \left( \tilde{S}_1^\top \tilde{S}_1 \right)^{-1} \|_{\text{op}} \leq 2$. Moreover we have

$$\|D_1^{-1/2}\|_{\text{op}} \leq \frac{1}{\delta_n},$$
$$\|D_2^{1/2}\|_{\text{op}} \leq \delta_n,$$
$$\|\tilde{S}_2 D_2^{1/2}\|_{\text{op}} \leq c\delta_n.$$

Thus,

$$\left\| \theta_2^\star - \left( D_2 \tilde{S}_2^\top \otimes \Delta V^\top \right) \hat{\gamma} \right\|_2 \leq \left( \delta_n^2 \|M\|_{\text{op}} \right)^{1/2} + \delta_n c \delta_n \left( \frac{3}{2} \right)^{1/2} 2 \frac{1}{\delta_n} \|M\|_{\text{op}}^{1/2}$$
$$= \left( \delta_n^2 \|M\|_{\text{op}} \right)^{1/2} \left( 1 + c\sqrt{6} \right)$$

Finally,

$$\left\| \theta^\star - \left( D\tilde{S}^\top \otimes \Delta V^\top \right) \hat{\gamma} \right\|_2^2 \leq \delta_n^2 \|M\|_{\text{op}} \left( 1 + c\sqrt{6} \right)^2. \tag{23}$$

Furthermore, looking into the second term,

$$\hat{\gamma}^\top \left( \tilde{S} D \tilde{S}^\top \otimes M \right) \hat{\gamma} = \left\| \left( D^{1/2} \tilde{S}^\top \otimes \Delta^{1/2} V^\top \right) \hat{\gamma} \right\|_2^2$$
$$= \left\| \left( D_1^{1/2} \tilde{S}_1^\top \otimes \Delta^{1/2} V^\top \right) \hat{\gamma} \right\|_2^2 + \left\| \left( D_2^{1/2} \tilde{S}_2^\top \otimes \Delta^{1/2} V^\top \right) \hat{\gamma} \right\|_2^2$$
$$= \left\| \left( D_1^{-1/2} \otimes \Delta^{-1/2} \right) \theta_1^\star \right\|_2^2 + \left\| \left( D_2^{1/2} \tilde{S}_2^\top \tilde{S}_1 \left( \tilde{S}_1^\top \tilde{S}_1 \right)^{-1} D_1^{-1} \otimes \Delta^{-1/2} \right) \theta_1^\star \right\|_2^2$$
$$\leq 1 + \left\| \tilde{S}_2 D_2^{1/2} \right\|_{\text{op}}^2 \left\| \tilde{S}_1 \right\|_{\text{op}}^2 \left\| \left( \tilde{S}_1^\top \tilde{S}_1 \right)^{-1} \right\|_{\text{op}}^2 \left\| D_1^{-1/2} \right\|_{\text{op}}^2 \left\| \left( D_1^{-1/2} \otimes \Delta^{-1/2} \right) \theta_1^\star \right\|_2^2$$
$$\leq 1 + c^2 \delta_n^2 \frac{3}{2} 4 \frac{1}{\delta_n^2}$$
$$= 1 + 6c^2$$
$$= \left( 1 + \sqrt{6}c \right)^2 - 2\sqrt{6}c$$
$$\leq \left( 1 + \sqrt{6}c \right)^2.$$

Finally, we obtain that

$$\left\| \theta^\star - \left( D\tilde{S}^\top \otimes \Delta V^\top \right) \hat{\gamma} \right\|_2^2 + \lambda_n \hat{\gamma}^\top \left( \tilde{S} D \tilde{S}^\top \otimes M \right) \hat{\gamma} \leq \left( 1 + \sqrt{6}c \right)^2 \left( \|M\|_{\text{op}} \delta_n^2 + \lambda_n \right), \tag{24}$$

and as a conclusion

$$\inf_{\Gamma \in \mathbb{R}^{s \times d}} \frac{1}{n} \|KS^\top \Gamma M - Z^\star\|_F^2 + \lambda_n \operatorname{Tr} \left( KS^\top \Gamma M \Gamma^\top S \right) \leq C^2 \left( \|M\|_{\text{op}} \delta_n^2 + \lambda_n \right), \tag{25}$$

where $C = 1 + \sqrt{6}c$. $\qquad\square$

Now, as for the proof of Theorem 2, let us prove equation first inequality in Theorem 4.

*Proof.* For any function in $\mathcal{B}(\mathcal{H}_{\mathcal{K}}) = \{f \in \mathcal{H}_{\mathcal{K}} : \|f\|_{\mathcal{H}_{\mathcal{K}}} \leq 1\}$, Lemma 1 still holds, then

$$\mathbb{E}[\ell_f] \leq \mathbb{E}_n[\ell_f] + R_n(l \circ \mathcal{B}(\mathcal{H}_{\mathcal{K}})) + \sqrt{\frac{8\log(2/\delta)}{n}}. \tag{26}$$

Then, using Corollary 1 from Maurer (2016), we have that:

$$R_n(\ell \circ \mathcal{B}(\mathcal{H}_{\mathcal{K}})) \leq \sqrt{2} L \mathcal{R}_n(\mathcal{B}(\mathcal{H}_{\mathcal{K}})), \tag{27}$$

where

$$\mathcal{R}_n(F) = \mathbb{E}\left[\sup_{f \in F} \left|\frac{2}{n}\sum_{i=1}^{n}\sum_{j=1}^{d}\sigma_{ij}f(x_i)_j\right| \mid x_1, \ldots, x_n\right]$$

$$= \mathbb{E}\left[\sup_{f \in F} \left|\frac{2}{n}\sum_{i=1}^{n}\langle \boldsymbol{\sigma}_i, f(x_i)\rangle_{\mathbb{R}^d}\right| \mid x_1, \ldots, x_n\right],$$

where $\sigma_{11}, \ldots, \sigma_{np}$ are $nd$ independent Rademacher variables, and for all $1 \leq i \leq n$, $\boldsymbol{\sigma}_i = (\sigma_{i1}, \ldots, \sigma_{id})^{\top}$. Hence

$$\mathcal{R}_n(\mathcal{B}(\mathcal{H}_{\mathcal{K}})) = \mathbb{E}\left[\sup_{\|f\|_{\mathcal{H}_{\mathcal{K}}} \leq 1} \left|\frac{2}{n}\sum_{i=1}^{n}\langle \boldsymbol{\sigma}_i, f(x_i)\rangle_{\mathbb{R}^d}\right| \mid x_1, \ldots, x_n\right]$$

$$= \mathbb{E}\left[\sup_{\|f\|_{\mathcal{H}_{\mathcal{K}}} \leq 1} \left|\left\langle \frac{2}{n}\sum_{i=1}^{n}\mathcal{K}_{x_i}\boldsymbol{\sigma}_i, f\right\rangle_{\mathcal{H}_{\mathcal{K}}}\right| \mid x_1, \ldots, x_n\right]$$

$$\leq \frac{2}{n}\mathbb{E}\left[\left\|\sum_{i=1}^{n}\mathcal{K}_{x_i}\boldsymbol{\sigma}_i\right\|_{\mathcal{H}_{\mathcal{K}}}^2 \mid x_1, \ldots, x_n\right]^{1/2}$$

$$= \frac{2}{n}\mathbb{E}\left[\sum_{i,j=1}^{n}\langle \boldsymbol{\sigma}_i, \mathcal{K}(x_i, x_j)\boldsymbol{\sigma}_j\rangle_{\mathbb{R}^d} \mid x_1, \ldots, x_n\right]^{1/2}$$

$$= \frac{2}{n}\mathbb{E}\left[\sum_{i,j=1}^{n}k(x_i, x_j)\langle \boldsymbol{\sigma}_i, M\boldsymbol{\sigma}_j\rangle_{\mathbb{R}^d} \mid x_1, \ldots, x_n\right]^{1/2}$$

$$= \frac{2}{n}\left(\sum_{i,j=1}^{n}k(x_i, x_j)\sum_{i',j'=1}^{d}\mathbb{E}[M_{i'j'}\sigma_{ii'}\sigma_{jj'}|x_1, \ldots, x_n]\right)^{1/2}$$

$$= \frac{2}{n}\left(\sum_{i=1}^{n}k(x_i, x_i)\sum_{i'=1}^{d}M_{i'i'}\right)^{1/2}$$

$$= \frac{2}{n}\left(\mathrm{Tr}(K \otimes M)\right)^{1/2}$$

$$\mathcal{R}_n(\mathcal{B}(\mathcal{H}_{\mathcal{K}})) \leq \frac{2}{n^{1/2}}\kappa^{1/2}\,\mathrm{Tr}(M)^{1/2}.$$

Finally, for any function $f \in \mathcal{B}(\mathcal{H}_{\mathcal{K}})$, for all $\delta \in (0,1)$, we have for a probability at least $1 - \delta$,

$$|\mathbb{E}[\ell_f] - \mathbb{E}_n[\ell_f]| \leq 4L\sqrt{\frac{2\kappa}{n}\,\mathrm{Tr}(M)} + 2\sqrt{\frac{8\log(2/\delta)}{n}}. \tag{28}$$

Now, for the approximation error term, we proceed as in the proof of Theorem 2. Let $\mathcal{H}_S = \left\{ f = \sum_{i=1}^n k(\cdot, x_i) M \left[ S^\top \tilde{\Gamma} \right]_i \mid \gamma \in \mathbb{R}^{s \times d} \right\}$. By Assumptions 2 and 3 and Jensen's inequality,

$$
\begin{aligned}
\mathbb{E}_n \left[ \ell_{\tilde{f}_s} \right] - \mathbb{E}_n \left[ \ell_{f_{\mathcal{H}_\mathcal{K}}} \right] &= \frac{1}{n} \sum_{i=1}^n \ell \left( \tilde{f}_s(x_i), y_i \right) - \frac{1}{n} \sum_{i=1}^n \ell \left( f_{\mathcal{H}_\mathcal{K}}(x_i), y_i \right) \\
&\leq \frac{1}{n} \sum_{i=1}^n \ell \left( \tilde{f}_s(x_i), y_i \right) + \frac{\lambda_n}{2} \left\| \tilde{f}_s \right\|_{\mathcal{H}_\mathcal{K}}^2 - \frac{1}{n} \sum_{i=1}^n \ell \left( f_{\mathcal{H}_\mathcal{K}}(x_i), y_i \right) \\
&= \inf_{\substack{f \in \mathcal{H}_S \\ \|f\|_{\mathcal{H}_\mathcal{K}} \leq 1}} \frac{1}{n} \sum_{i=1}^n \ell \left( f(x_i), y_i \right) - \frac{1}{n} \sum_{i=1}^n \ell \left( f_{\mathcal{H}_\mathcal{K}}(x_i), y_i \right) + \frac{\lambda_n}{2} \|f\|_{\mathcal{H}_\mathcal{K}}^2 \\
&\leq \inf_{\substack{f \in \mathcal{H}_S \\ \|f\|_{\mathcal{H}_\mathcal{K}} \leq 1}} \frac{L}{n} \sum_{i=1}^n \left\| f(x_i) - f_{\mathcal{H}_\mathcal{K}}(x_i) \right\|_2 + \frac{\lambda_n}{2} \\
&\leq L \inf_{\substack{f \in \mathcal{H}_S \\ \|f\|_{\mathcal{H}_\mathcal{K}} \leq 1}} \sqrt{\frac{1}{n} \sum_{i=1}^n \left\| f(x_i) - f_{\mathcal{H}_\mathcal{K}}(x_i) \right\|_2^2} + \frac{\lambda_n}{2} \\
&= L \sqrt{\inf_{\substack{f \in \mathcal{H}_S \\ \|f\|_{\mathcal{H}_\mathcal{K}} \leq 1}} \frac{1}{n} \left\| f^X - f_{\mathcal{H}_\mathcal{K}}^X \right\|_F^2} + \frac{\lambda_n}{2},
\end{aligned}
$$

where, for any $f \in \mathcal{H}_\mathcal{K}$, $f^X = (f(x_1), \ldots, f(x_n))^\top \in \mathbb{R}^{n \times d}$. Let $\tilde{f}_s^R = \sum_{i=1}^n k(\cdot, x_i) M \left[ S^\top \tilde{\Gamma}^R \right]_i$, where $\tilde{\Gamma}^R$ is a solution to

$$
\inf_{\Gamma \in \mathbb{R}^{s \times d}} \frac{1}{n} \left\| K S^\top \Gamma M - f_{\mathcal{H}_\mathcal{K}}^X \right\|_F^2 + \lambda_n \operatorname{Tr} \left( K S^\top \Gamma M \Gamma^\top S \right).
$$

It is easy to check that $\tilde{f}_s^R$ is also a solution to

$$
\inf_{\substack{f \in \mathcal{H}_S \\ \|f\|_{\mathcal{H}_\mathcal{K}} \leq \|\tilde{f}_s^R\|_{\mathcal{H}_\mathcal{K}}}} \frac{1}{n} \left\| f^X - f_{\mathcal{H}_\mathcal{K}}^X \right\|_F^2.
$$

Since we have $\|\tilde{f}_s^R\|_{\mathcal{H}_\mathcal{K}} \leq 1$ by Assumption 2, it holds

$$
\begin{aligned}
\inf_{\substack{f \in \mathcal{H}_S \\ \|f\|_{\mathcal{H}_\mathcal{K}} \leq 1}} \frac{1}{n} \left\| f^X - f_{\mathcal{H}_\mathcal{K}}^X \right\|_F^2 &\leq \inf_{\substack{f \in \mathcal{H}_S \\ \|f\|_{\mathcal{H}_\mathcal{K}} \leq \|\tilde{f}_s^R\|_{\mathcal{H}_\mathcal{K}}}} \frac{1}{n} \left\| f^X - f_{\mathcal{H}_\mathcal{K}}^X \right\|_F^2 \\
&= \inf_{\Gamma \in \mathbb{R}^{s \times d}} \frac{1}{n} \left\| K S^\top \Gamma M - f_{\mathcal{H}_\mathcal{K}}^X \right\|_F^2 + \lambda_n \operatorname{Tr} \left( K S^\top \Gamma M \Gamma^\top S \right).
\end{aligned}
$$

As a consequence, we have

$$
\mathbb{E}_n[\ell_{\tilde{f}_s}] - \mathbb{E}_n[\ell_{f_{\mathcal{H}_k}}] \leq L \sqrt{\inf_{\Gamma \in \mathbb{R}^{s \times d}} \frac{1}{n} \| K S^\top \Gamma M - f_{\mathcal{H}_k}^X \|_F^2 + \lambda_n \operatorname{Tr} \left( K S^\top \Gamma M \Gamma^\top S \right)} + \frac{\lambda_n}{2}.
$$

Finally, by Lemma 3 and Equation (28), we obtain the result stated. $\qquad \square$

Furthermore, we give the proof of the second claim, i.e. the excess risk bound for kernel ridge multi-output regression.

*Proof.* We now assume that the outputs are bounded, hence, without loss of generality, $\mathcal{Y} \subset \mathcal{B} \left( \mathbb{R}^d \right)$. First, we prove Lipschitz-continuity of the square loss under Assumptions 2 and 4. Let $g : z \in \mathcal{H}_\mathcal{K}(\mathcal{X}) \mapsto \frac{1}{2} \| z - y \|_2^2$.

We have that $\nabla g(z) = z - y$, and hence $\|\nabla g(z)\|_2 \leq \|f(x)\|_2 + 1$, for some $f \in \mathcal{H}_\mathcal{K}$ and $x \in \mathcal{X}$. By Assumptions 2 and 4 and Cauchy-Schwartz inequality, it is easy to check that

$$\|f(x)\|_2^2 \leq \left( \kappa \|M\|_{\mathrm{op}} \|f(x)\|_2^2 \right)^{1/2} ,$$

which gives us that $\|f(x)\|_2 \leq \kappa^{1/2} \|M\|_{\mathrm{op}}^{1/2}$ and then $\|\nabla g(z)\|_2 \leq \kappa^{1/2} \|M\|_{\mathrm{op}}^{1/2} + 1$. We finally obtain that

$$|\ell(f(x), y) - \ell(f'(x'), y)| \leq \left( \kappa^{1/2} \|M\|_{\mathrm{op}}^{1/2} + 1 \right) \|f(x) - f'(x')\|_2 .$$

We can then obtain the same generalisation bounds as above. Finally, looking at the approximation term,

$$
\begin{aligned}
\mathbb{E}_n \left[ l_{\tilde{f}_s} \right] - \mathbb{E}_n \left[ l_{f_{\mathcal{H}_k}} \right] &= \frac{1}{2n} \left\| \tilde{f}_s^X - Y \right\|_2^2 - \frac{1}{2n} \left\| f_{\mathcal{H}_k}^X - Y \right\|_2^2 \\
&\leq \frac{1}{2n} \left\| \tilde{f}_s^X - f_{\mathcal{H}_k}^X \right\|_2^2 \\
&\leq \inf_{\substack{f \in \mathcal{H}_S \\ \|f\|_{\mathcal{H}_k} \leq 1}} \frac{1}{2n} \left\| f^X - f_{\mathcal{H}_k}^X \right\|_2^2 + \frac{\lambda_n}{2} \\
&\leq \inf_{\gamma \in \mathbb{R}^s} \frac{1}{n} \left\| K S^\top \gamma - f_{\mathcal{H}_k}^X \right\|_2^2 + \lambda_n \gamma^\top S K S^\top \gamma + \frac{\lambda_n}{2} \\
&\leq \left( C^2 + \frac{1}{2} \right) \lambda_n + C^2 \delta_n^2 .
\end{aligned}
$$

Here again, as in second claim of Theorem 2, we can directly use bound (19) and then, in combination with (28), we obtain the stated second claim in Theorem 4. $\qquad\square$

Finally, we here prove Lemma 2.

*Proof.* Let $f \in \mathcal{H}_\mathcal{K}$ such that $\|f\|_{\mathcal{H}_\mathcal{K}} \leq 1$ and $z = \left( f(x_1)^\top, \ldots, f(x_n)^\top \right)^\top \in \mathbb{R}^{nd}$. We define the linear operator $S_X : \mathcal{H}_\mathcal{K} \to \mathbb{R}^{nd}$ such that $S_X(f) = \left( f(x_1)^\top, \ldots, f(x_n)^\top \right)^\top$ for all $f \in \mathcal{H}_\mathcal{K}$. Then for all $f \in \mathcal{H}_\mathcal{K}$ and $z = \left( z_1^\top, \ldots, z_n^\top \right)^\top \in \mathbb{R}^{nd}$ we have

$$\langle S_X(f), z \rangle_{\mathbb{R}^{nd}} = \sum_{i=1}^n \langle f(x_i), z_i \rangle_{\mathbb{R}^d} = \sum_{i=1}^n \langle f, \mathcal{K}_{x_i} z_i \rangle_{\mathcal{H}_\mathcal{K}} = \left\langle f, \sum_{i=1}^n \mathcal{K}_{x_i} z_i \right\rangle_{\mathcal{H}_\mathcal{K}} = \langle f, S_X^\star(z) \rangle_{\mathcal{H}_\mathcal{K}} .$$

Hence

$$z^\top \left( K^{-1} \otimes M^{-1} \right) z = \left\langle (K \otimes M)^{-1} S_X(f), S_x(f) \right\rangle_{\mathbb{R}^{nd}} = \left\langle S_X^\star \left( (K \otimes M)^{-1} S_X(f) \right), f \right\rangle_{\mathcal{H}_\mathcal{K}} .$$

We recall the eigendecompositions of $K$ and $M$

$$K = U(nD) U^\top = \sum_{i=1}^n n\mu_i u_i u_i^\top$$

$$M = V \Delta V^\top = \sum_{j=1}^d \mu_i v_j v_j^\top .$$

Then,

$$
\begin{aligned}
K \otimes M &= \left( \sum_{i=1}^{n} n\mu_i u_i u_i^\top \right) \otimes \left( \sum_{j=1}^{d} \mu_j v_j v_j^\top \right) \\
&= \sum_{i=1}^{n} \sum_{j=1}^{d} n\mu_i \mu_i \left( u_i u_i^\top \right) \otimes \left( v_j v_j^\top \right) \\
&= \sum_{i=1}^{n} \sum_{j=1}^{d} n\mu_i \mu_i \left( u_i \right) \otimes \left( v_j \right) \left( u_i^\top \right) \otimes \left( v_j^\top \right) \\
&= \sum_{i=1}^{n} \sum_{j=1}^{d} n\mu_i \mu_i \left( u_i \right) \otimes \left( v_j \right) \left( (u_i) \otimes (v_j) \right)^\top,
\end{aligned}
$$

and for all $1 \le i, i' \le n$ and $1 \le j, j' \le d$, if $(i, i') \ne (j, j')$, then $(u_i) \otimes (v_j)^\top ((u_{i'}) \otimes (v_{j'})) = 0$ and otherwise $(u_i) \otimes (v_j)^\top ((u_{i'}) \otimes (v_{j'})) = 1$. Then, this allows to show that the operator norm of a Kronecker product is the product of the operator norms, and that

$$
(K \otimes M)^{-1} = \sum_{i=1}^{n} \sum_{j=1}^{d} (n\mu_i \mu_i)^{-1} (u_i) \otimes (v_j) \left( (u_i) \otimes (v_j) \right)^\top. \tag{29}
$$

We define, for all $1 \le i \le n$ and $1 \le j \le d$,

$$
\varphi_{ij} = \frac{1}{\sqrt{n\mu_i \mu_j}} \sum_{l=1}^{n} u_{i_l} \mathcal{K}_{x_l} v_j. \tag{30}
$$

By definition, $\mathrm{span}\left( (\varphi_{ij})_{1 \le i \le n, 1 \le j \le d} \right) \subset \mathrm{span}\left( (\mathcal{K}_{x_i} v_j)_{1 \le i \le n, 1 \le j \le d} \right)$ and we show that the $\varphi_{ij}$s are orthonormal,

$$
\begin{aligned}
\langle \varphi_{ij}, \varphi_{i'j'} \rangle_{\mathcal{H}_\mathcal{K}} &= \left\langle \frac{1}{\sqrt{n\mu_i \mu_j}} \sum_{l=1}^{n} u_{i_l} \mathcal{K}_{x_l} v_j, \frac{1}{\sqrt{n\mu_{i'} \mu_j}} \sum_{l'=1}^{n} u_{i'_{l'}} \mathcal{K}_{x_{l'}} v_{j'} \right\rangle_{\mathcal{H}_\mathcal{K}} \\
&= \frac{1}{\sqrt{n\mu_i \mu_j}} \frac{1}{\sqrt{n\mu_{i'} \mu_{j'}}} \sum_{l,l'}^{n} u_{i_l} u_{i'_{l'}} \left\langle \mathcal{K}_{x_l} v_j, \mathcal{K}_{x_{l'}} v_{j'} \right\rangle_{\mathcal{H}_\mathcal{K}} \\
&= \frac{1}{\sqrt{n\mu_i \mu_j}} \frac{1}{\sqrt{n\mu_{i'} \mu_{j'}}} \sum_{l,l'}^{n} u_{i_l} u_{i'_{l'}} \left\langle v_j, \mathcal{K}_{x_l, x_{l'}} v_{j'} \right\rangle_{\mathbb{R}^d} \\
&= \frac{1}{\sqrt{n\mu_i \mu_j}} \frac{1}{\sqrt{n\mu_{i'} \mu_{j'}}} \sum_{l,l'}^{n} u_{i_l} u_{i'_{l'}} k(x_l, x_{l'}) \left\langle v_j, M v_{j'} \right\rangle_{\mathbb{R}^d} \\
&= \frac{1}{\sqrt{n\mu_i \mu_j}} \frac{1}{\sqrt{n\mu_{i'} \mu_{j'}}} \sum_{l,l'}^{n} u_{i_l} u_{i'_{l'}} k(x_l, x_{l'}) \mu_{j'} \left\langle v_j, v_{j'} \right\rangle_{\mathbb{R}^d} \\
&= 0 \quad \text{if} \quad j \ne j'.
\end{aligned}
$$

Otherwise, if $j = j'$,

$$
\begin{aligned}
\langle \varphi_{ij}, \varphi_{i'j'} \rangle_{\mathcal{H}_\mathcal{K}} &= \frac{1}{\sqrt{n\mu_i}} \frac{1}{\sqrt{n\mu_{i'}}} \sum_{l,l'}^{n} u_{i_l} u_{i'_{l'}} k(x_l, x_{l'}) \\
&= \frac{1}{\sqrt{n\mu_i}} \frac{1}{\sqrt{n\mu_{i'}}} \left\langle K u_i, u_{i'} \right\rangle_{\mathbb{R}^n} \\
&= \frac{1}{\sqrt{n\mu_i}} \frac{1}{\sqrt{n\mu_{i'}}} n\mu_i \left\langle u_i, u_{i'} \right\rangle_{\mathbb{R}^n} \\
&= 0 \quad \text{if} \quad i \ne i'.
\end{aligned}
$$

Hence, $\langle \varphi_{ij}, \varphi_{i'j'} \rangle_{\mathcal{H}_\mathcal{K}} = 0$ if $(i,i') \neq (j,j')$ and if $(i,i') = (j,j')$,

$$\langle \varphi_{ij}, \varphi_{i'j'} \rangle_{\mathcal{H}_\mathcal{K}} = 1.$$

Finally, $\mathrm{span}\left( (\varphi_{ij})_{1 \leq i \leq n, 1 \leq j \leq d} \right) \subset \mathrm{span}\left( (\mathcal{K}_{x_i} v_j)_{1 \leq i \leq n, 1 \leq j \leq d} \right)$ and $\dim\left( (\varphi_{ij})_{1 \leq i \leq n, 1 \leq j \leq d} \right) = nd = \dim\left( (\mathcal{K}_{x_i} v_j)_{1 \leq i \leq n, 1 \leq j \leq d} \right)$, hence $(\varphi_{ij})_{1 \leq i \leq n, 1 \leq j \leq d}$ yields an orthonormal basis of $\mathrm{span}\left( (\mathcal{K}_{x_i} v_j)_{1 \leq i \leq n, 1 \leq j \leq d} \right)$. As a consequence, all $f \in \mathcal{H}_\mathcal{K}$ can be decomposed as $f = f_1 + f_2$, with $f_1 \in \mathrm{span}\left( (\mathcal{K}_{x_i} v_j)_{1 \leq i \leq n, 1 \leq j \leq d} \right)$ and $f_2 \in \mathrm{span}\left( (\mathcal{K}_{x_i} v_j)_{1 \leq i \leq n, 1 \leq j \leq d} \right)^\perp$. Thus, for all $y \in \mathbb{R}^d$, $y$ can be written as $y = \sum_{j=1}^d y_j v_j$ and

$$
\begin{aligned}
\langle S_X(f), z \rangle_{\mathbb{R}^{nd}} &= \sum_{i=1}^n \langle f(x_i), z_i \rangle_{\mathbb{R}^d} \\
&= \sum_{i=1}^n \sum_{j=1}^d z_{i_j} \langle f(x_i), v_j \rangle_{\mathbb{R}^d} \\
&= \sum_{i=1}^n \sum_{j=1}^d z_{i_j} \langle f, \mathcal{K}_{x_i} v_j \rangle_{\mathcal{H}_\mathcal{K}} \\
&= \sum_{i=1}^n \sum_{j=1}^d z_{i_j} \langle f_1, \mathcal{K}_{x_i} v_j \rangle_{\mathcal{H}_\mathcal{K}} + \sum_{i=1}^n \sum_{j=1}^d z_{i_j} \langle f_2, \mathcal{K}_{x_i} v_j \rangle_{\mathcal{H}_\mathcal{K}} \\
&= \sum_{i=1}^n \sum_{j=1}^d z_{i_j} \langle f_1, \mathcal{K}_{x_i} v_j \rangle_{\mathcal{H}_\mathcal{K}} \\
&= \langle S_X(f_1), z \rangle_{\mathbb{R}^{nd}}.
\end{aligned}
$$

Hence, let $f \in \mathcal{H}_\mathcal{K}$ such that $\|f\|_{\mathcal{H}_\mathcal{K}} \leq 1$, written as $f = \sum_{i=1}^n \sum_{j=1}^d f_{ij} \varphi_{ij} + f^\perp$, with $f_{ij} \in \mathbb{R}$ for all $1 \leq i \leq n$ and $1 \leq j \leq d$ and such that $\sum_{i=1}^n \sum_{j=1}^d f_{ij}^2 \leq 1$ and $f^\perp \in \mathrm{span}\left( (\mathcal{K}_{x_i} v_j)_{1 \leq i \leq n, 1 \leq j \leq d} \right)^\perp$ such that $\|f^\perp\|_{\mathcal{H}_\mathcal{K}} \leq 1$ (since $\|f\|_{\mathcal{H}_\mathcal{K}} = \sum_{i=1}^n \sum_{j=1}^d f_{ij}^2 + \|f^\perp\|_{\mathcal{H}_\mathcal{K}} \leq 1$, we have that

$$S_X(f) = \sum_{i=1}^n \sum_{j=1}^d f_{ij} S_X(\varphi_{ij}),$$

and, for all $1 \leq l \leq n$,

$$
\begin{aligned}
\varphi_{ij}(x_l) &= \frac{1}{\sqrt{n\mu_i \mu_j}} \sum_{l'=1}^n u_{i_{l'}} k(x_{l'}, x_l) M v_j \\
&= \sqrt{\frac{\mu_j}{n\mu_i}} K_{l:}^\top u_i v_j,
\end{aligned}
$$

and then

$$S_X(\varphi_{ij}) = \sqrt{\frac{\mu_j}{n\mu_i}} (K u_i) \otimes v_j = \sqrt{n\mu_i \mu_j} u_i \otimes v_j. \tag{31}$$

Finally,

$$S_X(f) = \sum_{i=1}^n \sum_{j=1}^d f_{ij} (n\mu_i \mu_j)^{1/2} u_i \otimes v_j. \tag{32}$$

Besides,

$$(K \otimes M)^{-1} S_X(f) = \left( \sum_{i=1}^{n} \sum_{j=1}^{d} (n\mu_i\mu_i)^{-1} (u_i) \otimes (v_j) ((u_i) \otimes (v_j))^{\top} \right)$$

$$\times \left( \sum_{i'=1}^{n} \sum_{j'=1}^{d} f_{i'j'} (n\mu_{i'}\mu_{j'})^{1/2} u_{i'} \otimes v_{j'} \right)$$

$$= \sum_{i=1}^{n} \sum_{j=1}^{d} f_{ij} (n\mu_i\mu_i)^{-1/2} u_i \otimes v_j.$$

Then,

$$S_X^{\star} \left( (K \otimes M)^{-1} S_X(f) \right) = \sum_{i=1}^{n} \sum_{j=1}^{d} f_{ij} (n\mu_i\mu_i)^{-1/2} S_X^{\star} (u_i \otimes v_j)$$

$$= \sum_{i=1}^{n} \sum_{j=1}^{d} f_{ij} (n\mu_i\mu_i)^{-1/2} \sum_{i'=1}^{n} \mathcal{K}_{x_i}(u_{i_{i'}}, v_j),$$

and finally,

$$\left\langle S_X^{\star} \left( (K \otimes M)^{-1} S_X(f) \right), f \right\rangle_{\mathcal{H}_{\mathcal{K}}} = \sum_{i,i'=1}^{n} \sum_{j,j'=1}^{d} \sum_{l=1}^{n} f_{ij} f_{i'j'} (n\mu_i\mu_i)^{-1/2} u_{i_l} \left\langle \mathcal{K}_{x_l} v_j, \varphi_{i'j'} \right\rangle_{\mathcal{H}_{\mathcal{K}}}$$

$$= \sum_{i,i'=1}^{n} \sum_{j,j'=1}^{d} f_{ij} f_{i'j'} \left\langle (n\mu_i\mu_i)^{-1/2} \sum_{l=1}^{n} u_{i_l} \mathcal{K}_{x_l} v_j, \varphi_{i'j'} \right\rangle_{\mathcal{H}_{\mathcal{K}}}$$

$$= \sum_{i,i'=1}^{n} \sum_{j,j'=1}^{d} f_{ij} f_{i'j'} \left\langle \varphi_{ij}, \varphi_{i'j'} \right\rangle_{\mathcal{H}_{\mathcal{K}}}$$

$$= \sum_{i=1}^{n} \sum_{j=1}^{d} f_{ij}^2$$

$$\left\langle S_X^{\star} \left( (K \otimes M)^{-1} S_X(f) \right), f \right\rangle_{\mathcal{H}_{\mathcal{K}}} \leq 1.$$

Thus, we do have the ellipse constraint

$$\| \left( K^{-1/2} \otimes M^{-1/2} \right) z \|_2 \leq 1. \tag{33}$$

$\square$

## A.4 Proof of Theorem 5

### A.4.1 First claim of K-satisfiability

Let us now prove the first claim (l.h.s. of equation (4)) of the K-satisfiability for $p$-SR and $p$-SG sketches. It is articulated around the following two lemmas.

**Lemma 4.** *Let $M \in \mathbb{R}^{d \times d}$ be a symmetric matrix, $\varepsilon \in (0,1)$, and $\mathcal{C}_{\varepsilon}$ be an $\varepsilon$-cover of $\mathcal{B}^d$. Then we have*

$$\|M\|_{\mathrm{op}} \leq \frac{1}{1 - 2\varepsilon - \varepsilon^2} \sup_{v \in \mathcal{C}_{\varepsilon}} \left| \langle v, Mv \rangle \right|.$$

*Proof.* Let $M$, $\varepsilon$ and $\mathcal{C}_{\varepsilon}$ as in Lemma 4. Let $u \in \mathcal{B}^d$. By definition, there exist $v \in \mathcal{C}_{\varepsilon}$ and $w \in \mathcal{B}^d$ such that $u = v + \varepsilon w$. We thus have

$$\langle u, Mu \rangle = \langle v, Mv \rangle + 2\varepsilon \langle v, Mw \rangle + \varepsilon^2 \langle w, Mw \rangle. \tag{34}$$

Taking the supremum on both sides of (34) we obtain

$$
\begin{aligned}
\sup_{u \in \mathcal{B}^d} |\langle u, Mu \rangle| &= \sup_{v \in \mathcal{C}_\varepsilon,\, w \in \mathcal{B}^d} \Big( |\langle v, Mv \rangle| + 2\varepsilon |\langle v, Mw \rangle| + \varepsilon^2 |\langle w, Mw \rangle| \Big) \\
&\leq \sup_{v \in \mathcal{C}_\varepsilon} |\langle v, Mv \rangle| + 2\varepsilon \sup_{v \in \mathcal{C}_\varepsilon,\, w \in \mathcal{B}^d} |\langle v, Mw \rangle| + \varepsilon^2 \sup_{w \in \mathcal{B}^d} |\langle w, Mw \rangle| \\
&\leq \sup_{v \in \mathcal{C}_\varepsilon} |\langle v, Mv \rangle| + 2\varepsilon \sup_{v' \in \mathcal{B}^d,\, w \in \mathcal{B}^d} |\langle v', Mw \rangle| + \varepsilon^2 \|M\|_{\mathrm{op}} \\
&= \sup_{v \in \mathcal{C}_\varepsilon} |\langle v, Mv \rangle| + \left(2\varepsilon + \varepsilon^2\right) \|M\|_{\mathrm{op}},
\end{aligned}
$$

or again

$$
\|M\|_{\mathrm{op}} \leq \frac{1}{1 - 2\varepsilon - \varepsilon^2} \sup_{v \in \mathcal{C}_\varepsilon} \big|\langle v, Mv \rangle\big|.
$$

$\square$

**Lemma 5.** *Let $S \in \mathbb{R}^{s \times n}$ be a $p$-SR or a $p$-SG sketch. Let $v \in \mathcal{B}^n$, then for every $t > 0$, we have*

$$
\mathbb{P}\left\{ \big| \|Sv\|_2^2 - \|v\|_2^2 \big| > \frac{4}{p}\sqrt{\frac{2t}{s}} + \frac{4t}{sp} \right\} \leq 2e^{-t}.
$$

*Proof.* The proof of Lemma 5 is largely adapted from the proof of Theorem 2.13 in Boucheron et al. (2013). Let $S \in \mathbb{R}^{s \times n}$ be a $p$-SR or a $p$-SG sketch, and $v \in \mathcal{B}^n$. It is easy to check that for all $i \leq s$ we have $\mathbb{E}\left[[Sv]_i^2\right] = \frac{1}{s}\|v\|_2^2$, such that

$$
\big| \|Sv\|_2^2 - \|v\|_2^2 \big| = \left| \sum_{i=1}^{s} \left( [Sv]_i^2 - \frac{1}{s}\|v\|_2^2 \right) \right|.
$$

The proof then consists in applying Bernstein's inequality (Boucheron et al., 2013, Theorem 2.10) to the random variables $[Sv]_i^2$. We now have to find some constants $\nu$ and $c$ such that $\sum_{i=1}^{s} \mathbb{E}\left[[Sv]_i^4\right] \leq \nu$ and

$$
\sum_{i=1}^{s} \mathbb{E}\left[[Sv]_i^{2q}\right] \leq \frac{q!}{2}\nu c^{q-2} \quad \text{for all } q \geq 3.
$$

From (12) and (13), it is easy to check that the $S_{ij}$ are independent and $1/(sp)$ sub-Gaussian. Then, for all $\lambda \in \mathbb{R}$, we have

$$
\begin{aligned}
\mathbb{E}\left[\exp\left(\lambda [Sv]_i\right)\right] &= \mathbb{E}\left[\exp\left(\lambda \sum_{j=1}^{n} S_{ij} v_j\right)\right] \\
&= \prod_{j=1}^{n} \mathbb{E}\left[\exp\left(\lambda S_{ij} v_j\right)\right] \\
&\leq \exp\left(\frac{\lambda^2}{2sp}\|v\|_2^2\right) \\
&\leq \exp\left(\frac{\lambda^2}{2sp}\right).
\end{aligned}
$$

The random variable $[Sv]_i$ is therefore $1/(sp)$ sub-Gaussian, and Theorem 2.1 from Boucheron et al. (2013) yields that for every integer $q \geq 2$ it holds

$$
\mathbb{E}\left[[Sv]_i^{2q}\right] \leq \frac{q!}{2} 4 \left(\frac{2}{sp}\right)^q \leq \frac{q!}{2}\left(\frac{4}{sp}\right)^q.
$$

Choosing $q = 2$, we obtain

$$\sum_{i=1}^{s} \mathbb{E}\left[[Sv]_i^4\right] \le \sum_{i=1}^{s} \left(\frac{4}{sp}\right)^2 = \frac{16}{sp^2},$$

such that we can choose $\nu = 16/(sp^2)$ and $c = 4/(sp)$. Applying Theorem 2.10 from Boucheron et al. (2013) to the random variables $[Sv]_i^2$ finally gives that for any $t > 0$ it holds

$$\mathbb{P}\left\{\left|\,\|Sv\|_2^2 - \|v\|_2^2\,\right| > \frac{4}{p}\sqrt{\frac{2t}{s}} + \frac{4t}{sp}\right\} \le 2e^{-t}.$$

$\square$

**Proof of the first claim of the K-satisfiability.** Let $K \in \mathbb{R}^{n \times n}$ be a Gram matrix, and $S \in \mathbb{R}^{s \times n}$ be a $p$-SR or a $p$-SG sketch. Recall that we want to prove that there exists $c_0 > 0$ such that

$$\mathbb{P}\left\{\left\|U_1^\top S^\top S U_1 - I_{d_n}\right\|_{\mathrm{op}} > \frac{1}{2}\right\} \le 2e^{-c_0 s},$$

where $K/n = UDU^\top$ is the SVD of $K$, and $U_1 \in \mathbb{R}^{n \times d_n}$ contains the left part of $U$. Let $\varepsilon \in (0, 1)$, and $\mathcal{C}_\varepsilon = \{v^1, \ldots, v^{\mathcal{N}_\varepsilon}\}$ be an $\varepsilon$-cover of $\mathcal{B}^{d_n}$. We know that such a covering exists with cardinality $\mathcal{N}_\varepsilon \le \left(1 + \frac{2}{\varepsilon}\right)^{d_n}$, see e.g., Matoušek (2013). Let $Q = U_1^\top S^\top S U_1 - I_{d_n}$, applying Lemma 4, we have

$$\begin{aligned}
\mathbb{P}\left\{\|Q\|_{\mathrm{op}} > \frac{1}{2}\right\} &\le \mathbb{P}\left\{\sup_{i \le \mathcal{N}_\varepsilon} \left|\langle v^i, Qv^i\rangle\right| > \frac{1 - 2\varepsilon - \varepsilon^2}{2}\right\} \\
&\le \sum_{i \le \mathcal{N}_\varepsilon} \mathbb{P}\left\{\left|\langle v^i, Qv^i\rangle\right| > \frac{1 - 2\varepsilon - \varepsilon^2}{2}\right\} \\
&= \sum_{i \le \mathcal{N}_\varepsilon} \mathbb{P}\left\{\left|\,\|Sw^i\|_2^2 - \|w^i\|_2^2\,\right| > \frac{1 - 2\varepsilon - \varepsilon^2}{2}\right\},
\end{aligned}\tag{35}$$

where $w^i = U_1 v^i \in \mathcal{B}^n$. Now, by Lemma 5, for any $w \in \mathcal{B}^n$, we have

$$\mathbb{P}\left\{\left|\,\|Sw\|_2^2 - \|w\|_2^2\,\right| > \frac{4}{p}\sqrt{\frac{2t}{s}} + \frac{4t}{sp}\right\} \le 2e^{-t}.$$

Let $s \ge 32t/(\alpha^2 p^2)$, for some $\alpha \le 1$. Then, we have $\frac{4}{p}\sqrt{\frac{2t}{s}} + \frac{4t}{sp} \le \alpha + \frac{\alpha^2 p}{8} \le 2\alpha$, and therefore

$$\mathbb{P}\left\{\left|\,\|Sw\|_2^2 - \|w\|_2^2\,\right| > 2\alpha\right\} \le 2e^{-t}.$$

If we take $\alpha = (1 - 2\varepsilon - \varepsilon^2)/4$, we obtain

$$\mathbb{P}\left\{\left|\,\|Sw\|_2^2 - \|w\|_2^2\,\right| > \frac{1 - 2\varepsilon - \varepsilon^2}{2}\right\} \le 2e^{-t}$$

as long as $s \ge \frac{512t}{p^2(1-2\varepsilon-\varepsilon^2)^2}$. Now, let $t = \frac{p^2(1-2\varepsilon-\varepsilon^2)^2}{1024}s + \log(\mathcal{N}_\varepsilon)$, and $s \ge 1024\frac{\log(1+2/\varepsilon)}{p^2(1-2\varepsilon-\varepsilon^2)^2}d_n$. We do have

$$\frac{512t}{p^2(1-2\varepsilon-\varepsilon^2)^2} = \frac{s}{2} + \frac{512}{p^2(1-2\varepsilon-\varepsilon^2)^2}\log(\mathcal{N}_\varepsilon) \le \frac{s}{2} + \frac{s}{2} = s,$$

such that

$$\mathbb{P}\left\{\left|\,\|Sw\|_2^2 - \|w\|_2^2\,\right| > \frac{1 - 2\varepsilon - \varepsilon^2}{2}\right\} \le 2e^{-t} = \frac{2e^{-c_0 s}}{\mathcal{N}_\varepsilon},$$

where $c_0 = \frac{p^2(1-2\varepsilon-\varepsilon^2)}{1024}$. Plugging this result into (35), we get that as soon as $s \ge 1024\frac{\log(1+2/\varepsilon)}{p^2(1-2\varepsilon-\varepsilon^2)^2}d_n$ it holds

$$\mathbb{P}\left\{\|Q\|_{\mathrm{op}} > \frac{1}{2}\right\} \le 2e^{-c_0 s}.$$

Finally, we can tune $\varepsilon$ to optimize the lower bound on $s$. If we take $\varepsilon = 0.1$, we obtain $s \ge 5120d_n/p^2$, and $c_0 \ge p^2/2560$. $\square$

### A.4.2   Second claim of K-satisfiability

We now turn to the proof of the second claim (r.h.s. of equation (4)) of the K-satisfiability for $p$-SR and $p$-SG sketches. It builds upon the following two intermediate results, about the concentration of Lipschitz functions of Rademacher or Gaussian random variables.

**Lemma 6.** *Let $K > 0$, and let $X_1, \dots, X_n$ be independent real random variables with $|X_i| \leq K$ for all $1 \leq i \leq n$. Let $F : \mathbb{R}^n \to \mathbb{R}$ be a $L$-Lipschitz convex function. Then, there exist $C, c > 0$ such that for any $\lambda$ one has*

$$\mathbb{P}\{|F(X) - \mathbb{E}\,F(X)| \geq K\lambda\} \leq C' \exp\left(-c'\lambda^2/L^2\right) .$$

**Lemma 7.** *Let $X_1, \dots, X_n$ be i.i.d. standard Gaussian random variables. Let $F : \mathbb{R}^n \to \mathbb{R}$ be a $L$-Lipschitz function. Then, there exist $C, c > 0$ such that for any $\lambda$ one has*

$$\mathbb{P}\{|F(X) - \mathbb{E}\,F(X)| \geq \lambda\} \leq C' \exp\left(-c'\lambda^2/L^2\right) .$$

The above two lemmas are taken from Tao (2012), see Theorems 2.1.12 and 2.1.13 therein, but are actually well known results in the literature. In particular, Lemma 6 is adapted from Talagrand's inequality (Talagrand, 1995), while Lemma 7 is stated as Theorem 5.6 in Boucheron et al. (2013), with explicit constants. We however choose the writing by Tao (2012) in order to be consistent with the Rademacher case.

**Remark 3.** *Note that thanks to Lemma 6, we are even able to prove K-satisfiability for any sketch matrix $S$ whose entries are i.i.d. centered and reduced bounded random variables.*

**Proof of the second claim of the K-satisfiability.** Let $K \in \mathbb{R}^{n \times n}$ be a Gram matrix, and $S \in \mathbb{R}^{n \times s}$ be a $p$-SR or a $p$-SG sketch. Recall that we want to prove that there exist positive constants $c, c_1, c_2 > 0$ such that

$$\mathbb{P}\left\{ \left\| SU_2 D_2^{1/2} \right\|_{\mathrm{op}} > c\delta_n \right\} \leq c_1 e^{-c_2 s} ,$$

where $K/n = UDU^\top$ is the SVD of $K$, $U_2 \in \mathbb{R}^{n \times (n-d_n)}$ is the right part of $U$, and $D_2 \in \mathbb{R}^{(n-d_n) \times (n-d_n)}$ is the right bottom part of $D$. Note that we have $SU_2 D_2^{1/2} = SU\bar{D}^{1/2}$, where $\bar{D} = \mathrm{diag}\left(0_{d_n}, D_2\right) \in \mathbb{R}^{n \times n}$. Following Yang et al. (2017), we have

$$\left\| SU\bar{D}^{1/2} \right\|_{\mathrm{op}} = \sup_{u \in \mathcal{B}^s,\, v \in \mathcal{E}} |\langle u, Sv \rangle| ,$$

where $\mathcal{E} = \left\{ v \in \mathbb{R}^n \colon \exists\, w \in \mathcal{S}^{n-1}, v = U\bar{D}^{1/2}w \right\}$. Now, let $u^1, \dots u^\mathcal{N}$ be a 1/2-cover of $\mathcal{B}^s$. We know that such a covering exists with cardinality $\mathcal{N} \leq 5^s$. We then have

$$
\begin{aligned}
\left\| SU\bar{D}^{1/2} \right\|_{\mathrm{op}} &= \sup_{u \in \mathcal{B}^s,\, v \in \mathcal{E}} |\langle u, Sv \rangle| \\
&\leq \max_{i \leq \mathcal{N}} \sup_{v \in \mathcal{E}} \left| \left\langle u^i, Sv \right\rangle \right| + \frac{1}{2} \sup_{u \in \mathcal{B}^s,\, v \in \mathcal{E}} |\langle u, Sv \rangle| \\
&= \max_{i \leq \mathcal{N}} \sup_{v \in \mathcal{E}} \left| \left\langle u^i, Sv \right\rangle \right| + \frac{1}{2} \left\| SU\bar{D}^{1/2} \right\|_{\mathrm{op}} ,
\end{aligned}
$$

and rearranging implies that

$$\left\| SU\bar{D}^{1/2} \right\|_{\mathrm{op}} \leq 2 \max_{i \leq \mathcal{N}} \sup_{v \in \mathcal{E}} \left| \left\langle u^i, Sv \right\rangle \right| .$$

Hence, for every $c > 0$ we have

$$
\begin{aligned}
\mathbb{P}\left( \left\| SU_2 D_2^{1/2} \right\|_{\mathrm{op}} > c\delta_n \right) &\leq \mathbb{P}\left( \max_{i \leq \mathcal{N}} \sup_{v \in \mathcal{E}} \left| \left\langle u^i, Sv \right\rangle \right| > \frac{c}{2}\delta_n \right) \\
&\leq \sum_{i \leq \mathcal{N}} \mathbb{P}\left\{ \sup_{v \in \mathcal{E}} \left| \left\langle u^i, Sv \right\rangle \right| > \frac{c}{2}\delta_n \right\} .
\end{aligned}
\tag{36}
$$

Now, recall that

$$S = \frac{1}{\sqrt{sp}} \, B \circ R \,,$$

where $B \in \mathbb{R}^{n \times s}$ is filled with i.i.d. Bernoulli random variables with parameter $p$, $R \in \mathbb{R}^{n \times s}$ is filled with i.i.d. Rademacher or Gaussian random variables for $p$-SR and $p$-SG sketches respectively, and $\circ$ denotes the Hadamard (termwise) matrix product. The next step of the proof consists in controlling the right-hand side of (36) by showing that, conditionally on $B$, we have Lipschitz functions of Rademacher or Gaussian random variables, whose deviations can be bounded using Lemmas 6 and 7. Therefore, from now on we assume $B$ to be fixed, and only consider the randomness with respect to $R$. Let $u \in \mathcal{B}^s$, and define $F \colon \mathbb{R}^{n \times s} \to \mathbb{R}$ as

$$F(R) = \frac{1}{\sqrt{sp}} \sup_{v \in \mathcal{E}} \big| \, \langle u, (B \circ R)v \rangle \, \big| \,.$$

It is direct to check that $F$ is a convex function. Moreover, we have

$$\begin{aligned}
\sqrt{sp}\, F(R) &= \sup_{v \in \mathcal{E}} |\langle u, (B \circ R)v \rangle| \\
&= \sup_{v \in \mathcal{S}^{n-1}} |\langle u, (B \circ R)U\bar{D}^{1/2}v \rangle| \\
&= \sup_{v \in \mathcal{S}^{n-1}} |\langle \bar{D}^{1/2}U^{\top}(B \circ R)^{\top}u, v \rangle| \\
&= \left\| \bar{D}^{1/2}U^{\top}(B \circ R)^{\top}u \right\|_2 \,.
\end{aligned}$$

Thus, for any $R, R'$ we have

$$\begin{aligned}
\sqrt{sp}\, \big| F(R) - F(R') \big| &= \Big| \, \left\| \bar{D}^{1/2}U^{\top}(B \circ R)^{\top}u \right\|_2 - \left\| \bar{D}^{1/2}U^{\top}(B \circ R')^{\top}u \right\|_2 \, \Big| \\
&\leq \left\| \bar{D}^{1/2}U^{\top}\big(B \circ (R - R')\big)^{\top}u \right\|_2 \\
&\leq \left\| \bar{D}^{1/2} \right\|_{\mathrm{op}} \left\| U^{\top} \right\|_{\mathrm{op}} \left\| B \circ (R - R') \right\|_{\mathrm{op}} \left\| u \right\|_2 \\
&\leq \delta_n \left\| B \circ (R - R') \right\|_F \\
&\leq \delta_n \left\| R - R' \right\|_F \,,
\end{aligned} \tag{37}$$

such that $F$ is $\sqrt{\delta_n^2/(sp)}$-Lipschitz. Moreover, we have

$$\begin{aligned}
\sqrt{sp} \; \mathbb{E}\left[F(R)\right] &= \mathbb{E}\left[ \left\| D_2^{1/2}U_2^{\top}(B \circ R)^{\top}u \right\|_2 \right] \\
&\leq \sqrt{\mathbb{E}\left[ u^{\top}(B \circ R)U_2 D_2 U_2^{\top}(B \circ R)^{\top}u \right]} \\
&= \sqrt{ \sum_{k,k'=1}^{s} u_k u_{k'} \, \mathbb{E}\left[ \, \left[(B \circ R)U_2 D_2 U_2^{\top}(B \circ R)^{\top}\right]_{kk'} \, \right] } \\
&= \sqrt{ \sum_{k,k'=1}^{s} \sum_{l,l'=1}^{n} u_k u_{k'} [U_2 D_2 U_2^{\top}]_{ll'} \, \mathbb{E}\left[ [B \circ R]_{kl}[B \circ R]_{k'l'} \right] } \\
&= \sqrt{ \sum_{k=1}^{s} \sum_{l=1}^{n} B_{kl}^2 u_k^2 [U_2 D_2 U_2^{\top}]_{ll} } \\
&\leq \sqrt{\mathrm{Tr}(D_2)} \,,
\end{aligned}$$

which implies

$$\mathbb{E}\left[F(R)\right] \leq \sqrt{\frac{n}{sp}} \sqrt{\frac{\sum_{j=d_n+1}^{n} \mu_j}{n}} \leq \sqrt{\frac{n}{sp}} \sqrt{\frac{1}{n} \sum_{j=d_n+1}^{n} \min(\mu_j, \delta_n^2)} \leq \sqrt{\frac{\delta_n^2}{p}} \,, \tag{38}$$

where we have used the definition of $\delta_n^2$ and the assumption $s \geq \delta_n^2 n$. Coming back to (36), we obtain

$$\mathbb{P}\left\{\left\|SU_2 D_2^{1/2}\right\|_{\text{op}} > c\delta_n\right\} \leq 5^s\, \mathbb{E}\left[\mathbb{P}\left\{\sup_{v\in\mathcal{E}} |\langle u, Sv\rangle| > \frac{c}{2}\delta_n \mid B\right\}\right]$$

$$= 5^s\, \mathbb{E}\left[\mathbb{P}\left\{F(R) > \frac{c}{2}\delta_n\right\}\right]$$

$$\leq 5^s\, \mathbb{E}\left[\mathbb{P}\left\{F(R) - \mathbb{E}[F(R)] > \delta_n\left(\frac{c}{2} - \frac{1}{\sqrt{p}}\right)\right\}\right] \tag{39}$$

$$\leq C\, 5^s \exp\left(-c'\left(\frac{c}{2} - \frac{1}{\sqrt{p}}\right)^2 \delta_n^2 \frac{sp}{\delta_n^2}\right) \tag{40}$$

$$\leq C \exp\left(-c'\left(\left(\frac{c}{2} - \frac{1}{\sqrt{p}}\right)^2 p - \log(5)\right)s\right),$$

where (39) comes from the upper bound on $\mathbb{E}[F(R)]$ we derived in (38), and (40) derives from Lemmas 6 and 7 applied to the function $F$ whose Lipschitz constant has been established in (37). Therefore, taking $c = \frac{2}{\sqrt{p}}\left(1 + \sqrt{\log(5)}\right) + 1$, we have

$$\mathbb{P}\left\{\left\|SU_2 D_2^{1/2}\right\|_{\text{op}} > c\delta_n\right\} \leq c_1 e^{-c_2 s}$$

with $c_1 = C'$ and $c_2 = c'\left(\sqrt{p\log(5)} + \frac{p}{4}\right)$. $\qquad\square$

## A.5 Proof of Proposition 1

We prove Proposition 1 thanks to duality properties.

*Proof.* Since problems (3) and (11) are convex problems under Slater's constraints, strong duality holds and we will show that they admit the same dual problem

$$\min_{\zeta\in\mathbb{R}^n} \sum_{i=1}^{n} \ell_i^\star(-\zeta_i) + \frac{1}{2\lambda_n n}\zeta^\top K S^\top (SKS^\top)^\dagger SK\zeta\,. \tag{53}$$

First, we compute dual problem of (3), that can be rewritten

$$\min_{\gamma\in\mathbb{R}^s, u\in\mathbb{R}^n} \sum_{i=1}^{n} \ell_i(u_i) + \frac{\lambda_n n}{2}\gamma^\top SKS^\top\gamma$$

$$\text{s.t. } u = KS^\top\gamma.$$

Therefore the Lagrangian writes

$$\mathcal{L}(\gamma, u, \zeta) = \sum_{i=1}^{n} \ell_i(u_i) + \frac{\lambda_n n}{2}\gamma^\top SKS^\top\gamma + \sum_{i=1}^{n} \zeta_i(u_i - [KS^\top\gamma]_i)$$

$$= \sum_{i=1}^{n} \ell_i(u_i) + \frac{\lambda_n n}{2}\gamma^\top SKS^\top\gamma + \sum_{i=1}^{n} \zeta_i u_i - \zeta^\top KS^\top\gamma.$$

Differentiating with respect to $\gamma$ and using the definition of the Fenchel-Legendre transform, one gets

$$g(\zeta) = \inf_{\gamma\in\mathbb{R}^s, u\in\mathbb{R}^n} \mathcal{L}(\gamma, u, \zeta)$$

$$= \sum_{i=1}^{n} \inf_{u_i\in\mathbb{R}} \{\ell_i(u_i) + \zeta_i u_i\} + \inf_{\gamma\in\mathbb{R}^s} \left\{\frac{\lambda_n n}{2}\gamma^\top SKS^\top\gamma - \zeta^\top KS^\top\gamma\right\}$$

$$= \sum_{i=1}^{n} -\ell_i^\star(-\zeta_i) - \frac{1}{2\lambda_n n}\zeta^\top KS^\top (SKS^\top)^\dagger SK\zeta$$

together with the equality $SKS^\top\tilde{\gamma} = \frac{1}{\lambda_n n}SK\tilde{\zeta}$, implying $\tilde{\gamma} = \frac{1}{\lambda_n n}(SKS^\top)^\dagger SK\tilde{\zeta}$, where $\tilde{\zeta} \in \mathbb{R}^n$ is the solution of the following dual problem

$$\min_{\beta \in \mathbb{R}^n} \sum_{i=1}^{n} \ell_i^\star(-\beta_i) + \frac{1}{2\lambda_n n}\beta^\top KS^\top(SKS^\top)^\dagger SK\beta. \tag{53}$$

Then, we compute dual problem of (11), that can be rewritten

$$\min_{\omega \in \mathbb{R}^r, u \in \mathbb{R}^n} \sum_{i=1}^{n} \ell(u_i, y_i) + \frac{\lambda_n n}{2}\|\omega\|_2^2$$

$$\text{s.t. } u = KS^\top \tilde{K}_r\omega.$$

Therefore the Lagrangian writes

$$\mathcal{L}(\omega, u, \zeta) = \sum_{i=1}^{n} \ell_i(u_i) + \frac{\lambda_n n}{2}\|\omega\|_2^2 + \sum_{i=1}^{n} \zeta_i(u_i - [KS^\top \tilde{K}_r\omega]_i)$$

$$= \sum_{i=1}^{n} \ell_i(u_i) + \frac{\lambda_n n}{2}\|\omega\|_2^2 + \sum_{i=1}^{n} \zeta_i^\top u_i - \omega^\top \tilde{K}_r^\top SK\zeta.$$

Differentiating with respect to $\omega$ and using the definition of the Fenchel-Legendre transform, one gets

$$g(\zeta) = \inf_{\omega \in \mathbb{R}^r, u \in \mathbb{R}^n} \mathcal{L}(\omega, u, \zeta)$$

$$= \sum_{i=1}^{n} \inf_{u_i \in \mathbb{R}} \{\ell_i(u_i) + \zeta_i u_i\} + \inf_{\omega \in \mathbb{R}^r} \left\{\frac{\lambda_n n}{2}\|\omega\|_2^2 - \omega^\top \tilde{K}^{-1/2^\top} SK\zeta\right\}.$$

We have that

$$\frac{\partial}{\partial \omega}\left(\|\omega\|_2^2\right) = 2\omega$$

$$\frac{\partial}{\partial \omega}\left(\omega^\top \tilde{K}_r^\top SK\zeta\right) = \tilde{K}_r^\top SK\zeta,$$

Then, setting the gradient to zero, we obtain

$$\tilde{\omega} = \frac{1}{\lambda_n n}\ \tilde{K}_r^\top SK\tilde{\zeta}. \tag{41}$$

Hence, putting it into the Lagrangian,

$$-\frac{1}{\lambda_n n}\zeta^\top KS^\top \tilde{K}^{-1/2}\tilde{K}_r^\top SK\zeta = -\frac{1}{\lambda_n n}KS^\top\left(SKS^\top\right)^\dagger SK\zeta,$$

and

$$\frac{1}{2\lambda_n n}\zeta^\top KS^\top \tilde{K}_r\tilde{K}_r^\top SK\zeta = \frac{1}{2\lambda_n n}KS^\top\left(SKS^\top\right)^\dagger SK\zeta.$$

Hence, $\tilde{\zeta} \in \mathbb{R}^n$ is the solution to the following dual problem

$$\min_{\beta \in \mathbb{R}^n} \sum_{i=1}^{n} \ell_i^\star(-\beta_i) + \frac{1}{2\lambda_n n}\beta^\top KS^\top(SKS^\top)^\dagger SK\beta. \tag{53}$$

Finally, since both problems are convex and strong duality holds, we obtain through KKT conditions

$$\tilde{\omega} = \left(SKS^\top\right)^{-1/2^\top}\left(SKS^\top\right)\tilde{\gamma} = \left(\tilde{D}_r^{1/2}0_{r\times s-r}\right)\tilde{V}^\top\tilde{\gamma}$$

and

$$\min_{\gamma \in \mathbb{R}^s} \frac{1}{n}\sum_{i=1}^{n} \ell([KS^\top\gamma]_i, y_i) + \frac{\lambda_n}{2}\gamma^\top SKS^\top\gamma = \min_{\omega \in \mathbb{R}^r} \sum_{i=1}^{n} \ell\left(\omega^\top \mathbf{z}_S(x_i), y_i\right) + \frac{\lambda_n}{2}\|\omega\|_2^2.$$

$\square$

# B   On relaxing Assumption 2

In this section, we detail the discussion about relaxing Assumption 2, i.e. the restriction of the hypothesis set to the unit ball of the RKHS. Assumption 2 is a classical assumption in kernel literature to apply generalisation bounds based on Rademacher complexity of a bounded ball of a RKHS. Moreover, it is also useful in our case to derive an approximation error bound, describing how $K$-satisfiability of a sketch matrix allows to obtain a good approximation of the minimiser of the risk. However, let us discuss the consequences of relaxing this assumption. Indeed, all we need is a bound on the norm of the estimators $\tilde{f}_s$ – minimiser of the regularised ERM sketched problem – and $\tilde{f}_s^R$ – minimiser of the regularised ERM sketched denoised KRR problem. By definition, noting $\mathcal{H}_S = \left\{ f = \sum_{i=1}^{n} \left[ S^\top \gamma \right]_i k\left(\cdot, x_i\right) \mid \gamma \in \mathbb{R}^s \right\}$, we have that

$$\tilde{f}_s = \underset{f \in \mathcal{H}_S}{\arg\min}\ \frac{1}{n} \sum_{i=1}^{n} \ell(f(x_i), y_i) + \frac{\lambda_n}{2} \|f\|_{\mathcal{H}_k}^2 \,.$$

Hence,

$$\frac{\lambda_n}{2} \|\tilde{f}_s\|_{\mathcal{H}_k}^2 \leq \frac{1}{n} \sum_{i=1}^{n} \ell(\tilde{f}_s(x_i), y_i) + \frac{\lambda_n}{2} \|\tilde{f}_s\|_{\mathcal{H}_k}^2 \leq \frac{1}{n} \sum_{i=1}^{n} \ell(0, y_i) \leq 1 \,,$$

if we assume that $\max_{1 \leq i \leq n} \ell(0, y_i) \leq 1$ to simplify the derivations. As a consequence, we obtain that

$$\|\tilde{f}_s\|_{\mathcal{H}_k} \leq \sqrt{\frac{2}{\lambda_n}} \,. \tag{42}$$

Similarly, we have that

$$\tilde{f}_s^R = \underset{f \in \mathcal{H}_S}{\arg\min}\ \frac{1}{n} \left\| f^X - f_{\mathcal{H}_k}^X \right\|_2^2 + \frac{\lambda_n}{2} \|f\|_{\mathcal{H}_k}^2 \,,$$

that gives

$$\|\tilde{f}_s^R\|_{\mathcal{H}_k} \leq \left( \frac{1}{\lambda_n n} \|f_{\mathcal{H}_k}\|_{\mathcal{H}_k}^2 \right)^{1/2} \,.$$

By Assumptions 2 and 4,

$$\frac{1}{n} \|f_{\mathcal{H}_k}\|_{\mathcal{H}_k}^2 = \frac{1}{n} \sum_{i=1}^{n} \langle f_{\mathcal{H}_k}, k\left(\cdot, x_i\right) \rangle_{\mathcal{H}_k}$$

$$\leq \frac{1}{n} \sum_{i=1}^{n} \|f_{\mathcal{H}_k}\|_{\mathcal{H}_k}^2\, k\left(x_i, x_i\right)$$

$$\leq \kappa \,,$$

and finally

$$\|\tilde{f}_s^R\|_{\mathcal{H}_k} \leq \sqrt{\frac{\kappa}{\lambda_n}} \,. \tag{43}$$

**Remark 4.** *Note that in the multiple output settings, we obtain*

$$\|\tilde{f}_s^R\|_{\mathcal{H}_k} \leq \sqrt{\frac{\kappa \operatorname{Tr}(M)}{\lambda_n}} \,. \tag{44}$$

We are now equipped to derive the generalisation error bound $\mathbb{E}\left[\ell_{\tilde{f}_s}\right] - \mathbb{E}_n\left[\ell_{\tilde{f}_s}\right]$ and the approximation error bound $\mathbb{E}_n\left[\ell_{\tilde{f}_s}\right] - \mathbb{E}_n\left[\ell_{f_{\mathcal{H}_k}}\right]$. We first focus on the generalisation bound, and following the proof given in Appendix A.1 and given the new norm upper bound $\sqrt{\frac{2}{\lambda_n}}$, for any $\delta \in (0, 1)$, with probability $1 - \delta/2$, we have that

$$\mathbb{E}\left[\ell_{\tilde{f}_s}\right] - \mathbb{E}_n\left[\ell_{\tilde{f}_s}\right] \leq \frac{4L\sqrt{2\kappa}}{\sqrt{\lambda_n n}} + \sqrt{\frac{8\log(4/\delta)}{n}} \,. \tag{45}$$

This dependence in $1/\sqrt{\lambda_n}$ shows that, as expected by a regularisation penalty, with a fixed $n$, when $\lambda_n$ increases, the generalisation bound decreases and then we obtain a better generalisation performance. However, this behaviour does not reflect completely the role of $\lambda_n$, since there exists a tradeoff between overfitting and underfitting, and then it should not be set too large. We now focus on the approximation error bound. As in Appendix A.1, we obtain that

$$\mathbb{E}_n\left[\ell_{\tilde{f}_s}\right] - \mathbb{E}_n\left[\ell_{f_{\mathcal{H}_k}}\right] = \frac{1}{n}\sum_{i=1}^{n}\ell\left(\tilde{f}_s(x_i), y_i\right) - \frac{1}{n}\sum_{i=1}^{n}\ell\left(f_{\mathcal{H}_k}(x_i), y_i\right) \tag{46}$$

$$= \inf_{\substack{f \in \mathcal{H}_S \\ \|f\|_{\mathcal{H}_k} \leq \|\tilde{f}_s\|_{\mathcal{H}_k}}} \frac{1}{n}\sum_{i=1}^{n}\ell\left(f(x_i), y_i\right) - \frac{1}{n}\sum_{i=1}^{n}\ell\left(f_{\mathcal{H}_k}(x_i), y_i\right)$$

$$\leq \inf_{\substack{f \in \mathcal{H}_S \\ \|f\|_{\mathcal{H}_k} \leq \|\tilde{f}_s\|_{\mathcal{H}_k}}} \frac{L}{n}\sum_{i=1}^{n}\left|f\left(x_i\right) - f_{\mathcal{H}_k}\left(x_i\right)\right|$$

$$\leq L \inf_{\substack{f \in \mathcal{H}_S \\ \|f\|_{\mathcal{H}_k} \leq \|\tilde{f}_s\|_{\mathcal{H}_k}}} \sqrt{\frac{1}{n}\sum_{i=1}^{n}\left|f\left(x_i\right) - f_{\mathcal{H}_k}\left(x_i\right)\right|^2}$$

$$= L \sqrt{\inf_{\substack{f \in \mathcal{H}_S \\ \|f\|_{\mathcal{H}_k} \leq \|\tilde{f}_s\|_{\mathcal{H}_k}}} \frac{1}{n}\left\|f^X - f_{\mathcal{H}_k}^X\right\|_2^2}, \tag{47}$$

where, for any $f \in \mathcal{H}_k$, $f^X = (f(x_1), \ldots, f(x_n)) \in \mathbb{R}^n$. Let $\tilde{f}_s^R = \sum_{i=1}^{n}\left[S^\top\tilde{\gamma}^R\right]_i k\left(\cdot, x_i\right)$, where $\tilde{\gamma}^R$ is a solution to

$$\inf_{\gamma \in \mathbb{R}^s} \frac{1}{n}\left\|KS^\top\gamma - f_{\mathcal{H}_k}^X\right\|_2^2 + \lambda_n\gamma^\top SKS^\top\gamma.$$

It is easy to check that $\tilde{f}_s^R$ is also a solution to

$$\inf_{\substack{f \in \mathcal{H}_S \\ \|f\|_{\mathcal{H}_k} \leq \|\tilde{f}_s^R\|_{\mathcal{H}_k}}} \frac{1}{n}\left\|f^X - f_{\mathcal{H}_k}^X\right\|_2^2. \tag{48}$$

Now, comparing (47) and (48), as done in Appendix A.1, essentially boils down to comparing $\left\|\tilde{f}_s\right\|_{\mathcal{H}_k}$ and $\left\|\tilde{f}_s^R\right\|_{\mathcal{H}_k}$, which is a highly nontrivial question. In particular, the upper bounds (42) and (43) are not informative enough. Another solution could consist in adding $\frac{\lambda_n}{2}\left\|\tilde{f}_s\right\|_{\mathcal{H}_k}$ to (46). However, the upper bound (42) then transforms this term into a constant bias. This can be explained as (42) is very crude. Instead, having $\left\|\tilde{f}_s\right\|_{\mathcal{H}_k}$ bounded by $\lambda_n^\alpha$ for $\alpha \geq -1/2$ would be enough to exhibit a bias term that vanishes as $\lambda_n$ goes to 0. Note that it would still degrade the tradeoff with the genealisation term. Hence, if generalization errors can be dealt with when removing Assumption 2, it is much more complex to control (46).

## C  Some background on statistical properties when using a $p$-sparsified sketch

In this section, we focus on $p$-sparsified sketches, and give, according to different standard choices of kernels (Gaussian, polynomial and first-order Sobolev), the different learning rates obtained for the excess risk as well as the condition on $s$.

We can first derive the following corollaries for the excess risk of $p$-sparsified sketched estimator in both single and multiple output settings.

**Corollary 2.** *For a p-sparsified sketch matrix $S$ with $s \geq \max\left(C_0 d_n/p^2, \delta_n^2 n\right)$, we have with probability greater than $1 - C_1 e^{-sc(p)}$ in the single output setting for a generic Lipschitz loss,*

$$\mathbb{E}\left[\ell_{\tilde{f}_s}\right] \leq \mathbb{E}\left[\ell_{f_{\mathcal{H}_k}}\right] + LC\sqrt{\lambda_n + \delta_n^2} + \frac{\lambda_n}{2} + 8L\sqrt{\frac{\kappa}{n}} + \mathcal{O}\left(\sqrt{\frac{s}{n}}\right), \tag{49}$$

*and if* $\ell(z, y) = (z - y)^2 / 2$ *and* $\mathcal{Y} \subset [0, 1]$*, with probability at least* $1 - \delta$ *we have that*

$$\mathbb{E}\left[\ell_{\tilde{f}_s}\right] \le \mathbb{E}\left[\ell_{f_{\mathcal{H}_k}}\right] + \left(C^2 + \frac{1}{2}\right)\lambda_n + C^2\delta_n^2 + 8\frac{\kappa + \sqrt{\kappa}}{\sqrt{n}} + \mathcal{O}\left(\sqrt{\frac{s}{n}}\right). \tag{50}$$

**Corollary 3.** *For a* $p$*-sparsified sketch matrix* $S$ *with* $s \ge \max\left(C_0 d_n / p^2, \delta_n^2 n\right)$*, we have with probability greater than* $1 - C_1 e^{-sc(p)}$ *in the multiple output setting for a generic Lipschitz loss,*

$$\mathbb{E}\left[\ell_{\tilde{f}_s}\right] \le \mathbb{E}\left[\ell_{f_{\mathcal{H}_k}}\right] + LC\sqrt{\lambda_n + \|M\|_{\mathrm{op}}\delta_n^2} + \frac{\lambda_n}{2} + 8L\sqrt{\frac{\kappa \operatorname{Tr}(M)}{n}} + \mathcal{O}\left(\sqrt{\frac{s}{n}}\right). \tag{51}$$

*and if* $\ell(z, y) = \|z - y\|_2^2 / 2$ *and* $\mathcal{Y} \subset \mathcal{B}\left(\mathbb{R}^d\right)$*, with probability at least* $1 - \delta$ *we have that*

$$\mathbb{E}\left[\ell_{\tilde{f}_s}\right] \le \mathbb{E}\left[\ell_{f_{\mathcal{H}_k}}\right] + \left(C^2 + \frac{1}{2}\right)\lambda_n + C^2\|M\|_{\mathrm{op}}\delta_n^2 + 8\operatorname{Tr}(M)^{1/2}\frac{\kappa\|M\|_{\mathrm{op}}^{1/2} + \kappa^{1/2}}{\sqrt{n}} + \mathcal{O}\left(\sqrt{\frac{s}{n}}\right). \tag{52}$$

We summarize in Table 3 the different behaviours of $\delta_n^2$ and $d_n$ in the different spectrum regimes considered, in order to explicit the exact condition on $s$ in each case. More specifically, for a $D$th-order polynomial kernel, $d_n$, for any $n$ is at most $D + 1$, leading to $s$ of order $D + 1$ to be sufficient. Finally, we can derive the learning rate obtained as well as the exact condition on $s$ for each scenario, see Table 3. Compared with Random Fourier Features (Li et al., 2021), we see that we obtain slightly degraded learning rates for Gaussian and first-order Sobolev kernels, in comparison with the $\mathcal{O}(1/\sqrt{n})$ rate the authors obtain. Our rates remain however very close.

Table 3: Statistical dimension, lower bound obtained on $s$, and learning rate obtained for excess risk with $p$-sparsified sketches for different kernels.

| Kernel | $\delta_n^2$ | $d_n$ | $s$ | Learning rate |
|---|---|---|---|---|
| Gaussian | $\mathcal{O}\left(\frac{\sqrt{\log(n)}}{n}\right)$ | $\propto \sqrt{\log(n)}$ | $\Omega\left(\sqrt{\log(n)}/p^2\right)$ | $\mathcal{O}\left(\frac{(\log(n))^{1/4}}{n^{1/2}}\right)$ |
| Polynomial | $\mathcal{O}\left(\frac{1}{n}\right)$ | $\propto 1$ | $\Omega\left(\frac{1}{p^2}\right)$ | $\mathcal{O}\left(\frac{1}{\sqrt{n}}\right)$ |
| Sobolev | $\mathcal{O}\left(\frac{1}{n^{2/3}}\right)$ | $\propto n^{1/3}$ | $\Omega\left(n^{1/3}/p^2\right)$ | $\mathcal{O}\left(\frac{1}{n^{1/3}}\right)$ |

## D   Detailed algorithm of the generation and the decomposition of a $p$-sparsified sketch

In this section, we detail the process of generating a $p$-sparsified sketch and decomposing it as a product of a sub-Gaussian sketch $S_{\mathrm{SG}}$ and a sub-Sampling sketch $S_{\mathrm{SS}}$.

---
**Algorithm 1** Generation of a $p$-sparsified sketch
---
**input:** $s$, $n$ and $p$

Generate a $s \times n$ matrix $B$ whose entries are i.i.d. Bernouilli random variables of parameter $p$.

  indices $\longleftarrow$ indices of non-null columns of $B$.

  $B' \longleftarrow B$ where all null columns have been deleted.

Generate a matrix $M_{\mathrm{SG}}$ of the same size as $B'$ whose entries are either i.i.d. Gaussian or Rademacher random variables.

$S_{\mathrm{SG}} \longleftarrow M_{\mathrm{SG}} \circ B'$, where $\circ$ denotes the component-wise Hadamard matrix product.

**return** $S_{\mathrm{SG}}$ *and* indices
---

## E  Some background on complexity for single and multiple output regression

In this section, we detail the complexity in time and space of various matrix operations and iterations of stochastic subgradient descent in both single and multiple output settings for various sketching types.

We first recall the time and space complexities for elementary matrix products. The main advantage of using Sub-Sampling matrices is that computing $SK$ is equivalent to sampling $s$ training inputs and construct a $s \times n$ Gram matrix, hence we gain huge time complexity since we do not compute a matrix multiplication, as well as space complexity since we do not compute a $n \times n$ Gram matrix. As a consequence, the main advantage of our $p$-sparsified sketches is their ability to be decomposed as $S = S_{\text{SG}}S_{\text{SS}}$, where $S_{\text{SG}} \in \mathbb{R}^{s \times s'}$ is a sparse sub-Gaussian sketch and $S_{\text{SS}} \in \mathbb{R}^{s' \times n}$ is a sub-Sampling sketch, as explained in Section 3 and Appendix D. This *decomposition trick* is particularly interesting when $p$ is small, and since $s'$ follows a Binomial distribution of parameters $n$ and $1 - (1 - p)^s$ and we assume in our settings that $n$ is large, hence we have that $s' \approx \mathbb{E}[s'] \underset{p \to 0}{\sim} nsp$. In the following, we take $s' = nsp$. We recall that Accumulation matrices from Chen & Yang (2021a) writes as $S = \sum_{i=1}^{m} S_{(i)}$, where the $S_{(i)}$s are sub-sampling matrices whose each row is multiplied by independent Rademacher variables. Hence, both $p$-sparsified and Accumulation sketches are interesting since it completely benefits from computational efficiency of sub-sampling matrices. See table 4 for complexity analysis of matrix multiplications.

Going into the complexity of the learning algorithms, the main difference between single and multiple output settings are the computation of feature maps, relying on the construction of $SKS^\top$ and the computation of the square root of its pseudo-inverse for the single output setting which is not present in the multiple output settings. We assume in our framework that $d$ and even $d^2$ are typically very small in comparison with $n$. Hence, we have that the complexity in the single output case is dominated by the complexity of the operation $SKS^\top$, whereas in the multiple output case it is dominated by the complexity of the operation $SK$. We see that from a time complexity perspective, $p$-sparsified sketches outperform Accumulation sketches in single output settings as long as $p \leq m/n\sqrt{s}$, and in multiple output settings as long as $p \leq m/ns$. From a space complexity perspective, Accumulation is always better as $nsp$ is typically greater than $s$, otherwise it shows poor performance. However, $p$ is usually chosen such that $nsp$ is not very large compared with $s$.

Table 4: Complexity of matrix operations for each sketching type.

| Sketching type | Complexity type | $SK$ | $SKS^\top$ |
|---|---|---|---|
| Gaussian | time | $\mathcal{O}\left(n^2 s\right)$ | $\mathcal{O}\left(n^2 s\right)$ |
| | space | $\mathcal{O}\left(n^2\right)$ | $\mathcal{O}\left(n^2\right)$ |
| $p$-sparsified | time | $\mathcal{O}\left(n^2 s^2 p\right)$ | $\mathcal{O}\left(n^2 s^3 p^2\right)$ |
| | space | $\mathcal{O}\left(n^2 sp\right)$ | $\mathcal{O}\left(n^2 s^2 p^2\right)$ |
| Accumulation | time | $\mathcal{O}\left(nsm\right)$ | $\mathcal{O}\left(s^2 m^2\right)$ |
| | space | $\mathcal{O}\left(ns\right)$ | $\mathcal{O}\left(s^2\right)$ |
| CountSketh | time | $\mathcal{O}\left(n^2\right)$ | $\mathcal{O}\left(n^2\right)$ |
| | space | $\mathcal{O}\left(n^2\right)$ | $\mathcal{O}\left(n^2\right)$ |
| Sub-Sampling | time | $\mathcal{O}\left(ns\right)$ | $\mathcal{O}\left(s^2\right)$ |
| | space | $\mathcal{O}\left(ns\right)$ | $\mathcal{O}\left(s^2\right)$ |

## F  Discussion with dual implementation

In this section we detail the discussion about duality of kernel machines when using sketching. A first idea consists in computing the dual problem to the sketched problem (3). It writes

$$\min_{\zeta \in \mathbb{R}^n} \sum_{i=1}^{n} \ell_i^\star\left(-\zeta_i\right) + \frac{1}{2\lambda_n n} \zeta^\top K S^\top (SKS^\top)^\dagger SK\zeta, \tag{53}$$

where $\ell_i = \ell(\cdot, y_i)$, and $f^\star$ denotes the Fenchel-Legendre transform of $f$, such that $f^\star(\theta) = \sup_x \langle \theta, x \rangle - f(x)$. First note that sketching with a subsampling matrix in the primal is thus equivalent to using a Nyström approximation in the dual. This remark generalizes for any loss function the observation made in Yang et al. (2017) for the kernel Ridge regression. However, although the $\ell_i^\star$ might be easier to optimize, solving (53) seems not a meaningful option, as duality brought us back to optimizing over $\mathbb{R}^n$, what we initially intended to avoid. The natural alternative thus appears to use duality first, and then sketching. The resulting problem writes

$$\min_{\theta \in \mathbb{R}^s} \; \sum_{i=1}^n \ell_i^\star \left( -[S^\top \theta]_i \right) + \frac{1}{2\lambda_n n} \theta^\top SKS^\top \theta \,. \tag{54}$$

It is interesting to note that (54) is also the sketched version of Problem (53), which we recall is itself the dual to the sketched primal problem. Hence, sketching in the dual can be seen as a double approximation. As a consequence, the objective value reached by minimizing (54) is always larger than that achieved by minimizing (3), and theoretical guarantees for such an approach are likely to be harder to obtain. Another limitation of (54) regards the $\ell_i^\star(-[S^\top \theta]_i)$. Indeed, these terms generally contain the non-differentiable part of the objective function (for the $\epsilon$-insensitive Ridge regression we have $\sum_i \ell_i^\star(\theta_i) = \frac{1}{2}\|\theta\|_2^2 + \langle \theta, y \rangle + \epsilon\|\theta\|_1$ for instance), and are usually minimized by proximal gradient descent. However, using a similar approach for (54) is impossible, since the proximal operator of $\ell_i^\star(S^\top \cdot)$ is only computable if $S^\top S = I_n$, which is never the case. Instead, one may use a primal-dual algorithm (Chambolle et al., 2018; Vu, 2011; Condat, 2013), which solves the saddle-point optimization problem of the Lagrangian, but maintain a dual variable in $\mathbb{R}^n$. Coordinate descent versions of such algorithms (Fercoq & Bianchi, 2019; Alacaoglu et al., 2020) may also be considered, as they leverage the possible sparsity of $S$ to reduce the per-iteration cost. In order to converge, these algorithms however require a number of iteration that is of the order of $n$, making them hardly relevant in the large scale setting we consider.

For all the reasons listed above, we thus believe that minimizing (53) or (54) is not theoretically relevant nor computationally attractive, and that running stochastic (sub-)gradient descent on the primal problem, as detailed at the beginning of the section, is the best way to proceed algorithmically despite the possibly more elegant dual formulations. Finally, we highlight that although the condition $S^\top S = I_n$ is almost surely not verified (we have $S \in \mathbb{R}^{s \times n}$ with $s < n$), we still have $\mathbb{E}[S^\top S] = I_n$ for most sketching matrices. An interesting research direction could thus consist in running a proximal gradient descent assuming that $S^\top S = I_n$, and controlling the error incurred by such an approximation.

## G   Examples of Lipschitz-continuous losses

**Robust losses for multiple output regression:**   For $p \geq 1$, $\mathcal{Y} \subset \mathbb{R}^p$, and for all $y, y' \in \mathcal{Y}$, $\ell(y, y') = g(y - y')$, where $g$ is:

**For $\kappa$-Huber**: For $\kappa > 0$:

$$\forall y \in \mathcal{Y}, g(y) = \begin{cases} \frac{1}{2}\|y\|_\mathcal{Y}^2 & \text{if } \|y\|_\mathcal{Y} \leq \kappa \\ \kappa \left( \|y\|_\mathcal{Y} - \frac{\kappa}{2} \right) & \text{otherwise} \end{cases} .$$

**For $\epsilon$-SVR** (i.e. $\epsilon$-insensitive $\ell_1$ loss): For $\epsilon > 0$:

$$\forall y \in \mathcal{Y}, g(y) = \begin{cases} \|y\|_\mathcal{Y} - \epsilon & \text{if } \|y\|_\mathcal{Y} \geq \epsilon \\ 0 & \text{otherwise} \end{cases} .$$

**The pinball loss (Koenker, 2005) for joint quantile regression:**   For $d$ quantile levels, $\tau_1 < \tau_2 < \ldots < \tau_d$ with $\tau_i \in (0, 1)$, we define:

$$\ell_\tau(f(x), y) = L_\tau(f(x) - y\mathbb{1}_d),$$

with the following definition for $L_\tau$ the extension of pinball loss to $\mathbb{R}^d$ (Sangnier et al., 2016):
For $r \in \mathbb{R}^d$:

$$L_\tau(r) = \sum_{j=1}^{d} \left\{ \begin{array}{ll} \tau_j r_j & \text{if } r_j \geq 0, \\ (\tau_j - 1)r_j & \text{if } r_j < 0. \end{array} \right.$$

## H  Additional experiments

In this section, we bring some additional experiments and details.

### H.1  Simulated dataset for single output regression

First, we report the plots obtained with $\kappa$-Huber for $p$-SG sketches (see Figure 2) and note that we observe a behaviour similar to $p$-SR sketches when varying $p$ and in comparison to other types of sketching and RFFs. However, we see that the MSE obtained is slightly worse than $p$-SR sketches. An explanation might be that, in a very sparse regime, i.e. very low $p$, a $p$-SG sketch is too different than a Gaussian sketch, making it lose some good statistical properties of Gaussian sketches. We however observe that the larger $p$ is, the smaller is the statistical performance between $p$-SR and $p$-SG sketches.

We then report in the following the corresponding plots obtained with $\epsilon$-SVR, that witnesses the same phenomenon observed earlier with $\kappa$-Huber about the interpolation between Nyström method and Gaussian sketching while varying the probability of being different than 0 $p$, with $p$-SR sketches (see Figure 3) and $p$-SG sketches (see Figure 4).

### H.2  Multi-Output Regression on real datasets

We here first a brief presentation on River Flow and Supply Chain Management:

1. River Flow datasets aim at predicting the river network flows for 48 hours in the future at specific locations. These locations are 8 sites in the Mississippi River network in the United States and were obtained from the US National Weather Service. Dataset rf2 extends rf1 since it contains additional precipitation forecast information for each of the 8 sites.

2. The datasets scm1d and sm20d come from the Trading Agent Competition in Supply Chain Management (TAC SCM) tournament from 2010. More details about data preprocessing can be found in Groves & Gini (2015). The dataset contains prices of products at specific days, and the task is to predict to the next day mean price (scm1d) or mean price for 20-days in the future (scm20d) for each product in the simulation.

To conduct these experiments, the train-test splits used are the ones available at http://mulan.sourceforge.net/datasets-mtr.html. Besides, we used multi-output Kernel Ridge Regression framework and an input Gaussian kernel and an operator $M = I_d$. We selected regularisation parameter $\lambda_n$ and bandwidth of kernel $\sigma^2$ via a 5-fold cross-validation. Results are averages over 30 replicates for sketched models.

Table 5: Numbers of training samples ($n_{tr}$), test samples ($n_{te}$), features ($q$) and targets ($d$).

| Dataset | $n_{tr}$ | $n_{te}$ | $q$ | $d$ |
|---|---|---|---|---|
| Boston | 354 | 152 | 13 | 5 |
| otoliths | 3780 | 165 | 4096 | 5 |
| rf1 | 4108 | 5017 | 64 | 8 |
| rf2 | 4108 | 5017 | 576 | 8 |
| scm1d | 8145 | 1658 | 280 | 16 |
| scm20d | 7463 | 1503 | 61 | 16 |

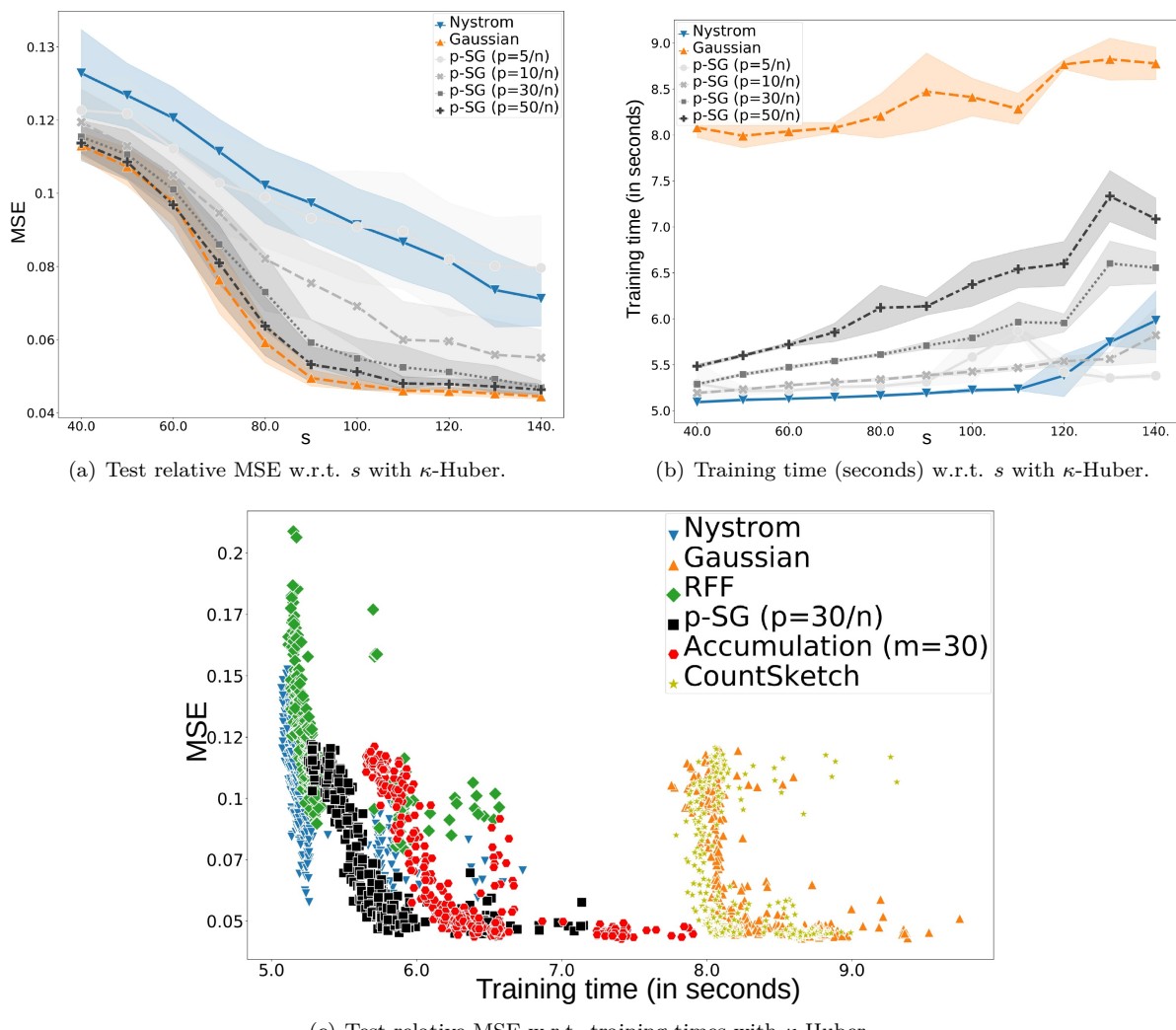

(a) Test relative MSE w.r.t. $s$ with $\kappa$-Huber.

(b) Training time (seconds) w.r.t. $s$ with $\kappa$-Huber.

(c) Test relative MSE w.r.t. training times with $\kappa$-Huber.

Figure 2: Trade-off between Accuracy and Efficiency for $p$-SG sketches with $\kappa$-Huber loss on synthetic dataset.

We compare our non-sketched framework with the sketched one, and we furthermore compare our $p$-sparsified sketches with Accumulation sketch from Chen & Yang (2021a) and CountSketch from Clarkson & Woodruff (2017). Moreover, we report the range of results obtained by SOTA methods available at Spyromitros-Xioufis et al. (2016). All results in terms of Test RRMSE are reported in Table 6, we see that our $p$-sparsified sketches allow to ally statistical and computational performance, since we maintain an accuracy of the same order as without sketching, and these sketches outperform Accumulation in terms of training times (see Table 2). In comparison to SOTA, our framework does not compete with the best results obtained in Spyromitros-Xioufis et al. (2016), but almost always remains within the range of results obtained with SOTA methods.

Table 6: Test RRMSE and ARRMSE for different methods on the MTR datasets. For decomposable kernel-based models, loss here is square loss and $s = 100$ when performing Sketching.

| Dataset | Targets | w/o Sketch | $20/n_{tr}$-SR | $20/n_{tr}$-SG | Acc. $m = 20$ | CountSketch | SOTA |
|---|---|---|---|---|---|---|---|
| rf1 | Mean | **0.575** | $0.584 \pm 0.003$ | $0.583 \pm 0.003$ | $0.592 \pm 0.001$ | $\mathbf{0.575 \pm 0.0005}$ | [0.091, 0.983] |
| | chsi2 | 0.351 | $0.356 \pm 0.005$ | $0.357 \pm 0.004$ | $0.361 \pm 0.002$ | $\mathbf{0.350 \pm 0.002}$ | [0.033, 0.797] |
| | nasi2 | 1.085 | $1.124 \pm 0.003$ | $1.124 \pm 0.003$ | $\mathbf{1.082 \pm 0.0004}$ | $1.110 \pm 0.0003$ | [0.376, 1.951] |
| | eadm7 | 0.388 | $0.397 \pm 0.004$ | $0.398 \pm 0.003$ | $\mathbf{0.387 \pm 0.001}$ | $\mathbf{0.387 \pm 0.001}$ | [0.039, 1.019] |
| | sclm7 | 0.660 | $0.659 \pm 0.008$ | $0.661 \pm 0.005$ | $0.681 \pm 0.002$ | $\mathbf{0.648 \pm 0.002}$ | [0.047, 1.503] |
| | clkm7 | 0.283 | $\mathbf{0.281 \pm 0.001}$ | $0.282 \pm 0.001$ | $0.293 \pm 0.0005$ | $\mathbf{0.281 \pm 0.0004}$ | [0.031, 0.587] |
| | vali2 | 0.614 | $0.633 \pm 0.008$ | $0.635 \pm 0.010$ | $0.656 \pm 0.003$ | $\mathbf{0.611 \pm 0.003}$ | [0.037, 0.571] |
| | napm7 | **0.593** | $0.628 \pm 0.020$ | $0.614 \pm 0.016$ | $0.627 \pm 0.003$ | $0.601 \pm 0.003$ | [0.038, 0.909] |
| | dldi4 | 0.629 | $\mathbf{0.597 \pm 0.004}$ | $\mathbf{0.597 \pm 0.003}$ | $0.646 \pm 0.001$ | $0.614 \pm 0.002$ | [0.029, 0.534] |
| rf2 | Mean | **0.578** | $0.671 \pm 0.009$ | $0.656 \pm 0.006$ | $0.796 \pm 0.006$ | $0.715 \pm 0.011$ | [0.095, 1.103] |
| | chsi2 | **0.318** | $0.382 \pm 0.016$ | $0.358 \pm 0.010$ | $0.478 \pm 0.006$ | $0.426 \pm 0.013$ | [0.034, 0.737] |
| | nasi2 | 1.099 | $1.084 \pm 0.005$ | $1.092 \pm 0.006$ | $\mathbf{1.018 \pm 0.003}$ | $1.036 \pm 0.002$ | [0.384, 3.143] |
| | eadm7 | **0.342** | $0.390 \pm 0.013$ | $0.369 \pm 0.007$ | $0.456 \pm 0.004$ | $0.417 \pm 0.010$ | [0.040, 0.737] |
| | sclm7 | **0.610** | $0.719 \pm 0.030$ | $0.672 \pm 0.021$ | $0.948 \pm 0.014$ | $0.852 \pm 0.030$ | [0.049, 0.970] |
| | clkm7 | **0.311** | $0.328 \pm 0.009$ | $0.330 \pm 0.009$ | $0.614 \pm 0.005$ | $0.436 \pm 0.006$ | [0.041, 0.891] |
| | vali2 | **0.712** | $0.960 \pm 0.044$ | $0.894 \pm 0.043$ | $0.890 \pm 0.017$ | $0.939 \pm 0.028$ | [0.047, 0.956] |
| | napm7 | **0.589** | $0.812 \pm 0.014$ | $0.831 \pm 0.017$ | $1.110 \pm 0.023$ | $0.856 \pm 0.032$ | [0.039, 0.617] |
| | dldi4 | **0.646** | $0.696 \pm 0.010$ | $0.701 \pm 0.011$ | $0.855 \pm 0.004$ | $0.761 \pm 0.007$ | [0.032, 0.770] |
| scm1d | Mean | **0.418** | $0.422 \pm 0.002$ | $0.423 \pm 0.001$ | $0.423 \pm 0.001$ | $0.420 \pm 0.001$ | [0.330, 0.457] |
| | lbl | **0.358** | $0.365 \pm 0.003$ | $0.364 \pm 0.002$ | $0.367 \pm 0.001$ | $0.363 \pm 0.001$ | [0.294, 0.409] |
| | mtlp2 | **0.352** | $0.360 \pm 0.003$ | $0.362 \pm 0.003$ | $0.362 \pm 0.001$ | $0.358 \pm 0.001$ | [0.308, 0.436] |
| | mtlp3 | **0.409** | $0.419 \pm 0.003$ | $0.416 \pm 0.002$ | $0.417 \pm 0.001$ | $0.416 \pm 0.002$ | [0.315, 0.442] |
| | mtlp4 | **0.417** | $0.427 \pm 0.002$ | $0.426 \pm 0.003$ | $0.426 \pm 0.001$ | $0.423 \pm 0.002$ | [0.325, 0.461] |
| | mtlp5 | 0.495 | $\mathbf{0.491 \pm 0.006}$ | $0.492 \pm 0.006$ | $0.502 \pm 0.002$ | $0.492 \pm 0.003$ | [0.349, 0.530] |
| | mtlp6 | 0.534 | $\mathbf{0.524 \pm 0.008}$ | $0.527 \pm 0.006$ | $0.537 \pm 0.002$ | $0.527 \pm 0.002$ | [0.347, 0.540] |
| | mtlp7 | 0.531 | $\mathbf{0.519 \pm 0.008}$ | $0.523 \pm 0.006$ | $0.534 \pm 0.002$ | $0.523 \pm 0.003$ | [0.338, 0.526] |
| | mtlp8 | 0.542 | $\mathbf{0.536 \pm 0.010}$ | $0.540 \pm 0.008$ | $0.547 \pm 0.002$ | $0.537 \pm 0.003$ | [0.345, 0.504] |
| | mtlp9 | **0.385** | $0.395 \pm 0.003$ | $0.395 \pm 0.002$ | $0.390 \pm 0.001$ | $0.390 \pm 0.002$ | [0.323, 0.456] |
| | mtlp10 | **0.389** | $0.398 \pm 0.003$ | $0.397 \pm 0.003$ | $0.394 \pm 0.002$ | $0.394 \pm 0.001$ | [0.339, 0.456] |
| | mtlp11 | **0.424** | $0.430 \pm 0.003$ | $0.429 \pm 0.003$ | $0.426 \pm 0.001$ | $0.426 \pm 0.001$ | [0.327, 0.445] |
| | mtlp12 | **0.420** | $0.422 \pm 0.003$ | $0.421 \pm 0.004$ | $0.423 \pm 0.001$ | $0.421 \pm 0.002$ | [0.350, 0.466] |
| | mtlp13 | **0.349** | $0.358 \pm 0.004$ | $0.354 \pm 0.004$ | $0.351 \pm 0.001$ | $0.351 \pm 0.001$ | [0.322, 0.419] |
| | mtlp14 | **0.347** | $0.364 \pm 0.004$ | $0.363 \pm 0.003$ | $0.350 \pm 0.001$ | $0.355 \pm 0.002$ | [0.356, 0.472] |
| | mtlp15 | **0.361** | $0.371 \pm 0.004$ | $0.370 \pm 0.003$ | $0.363 \pm 0.001$ | $0.364 \pm 0.001$ | [0.314, 0.406] |
| | mtlp16 | **0.376** | $0.382 \pm 0.003$ | $0.384 \pm 0.003$ | $\mathbf{0.376 \pm 0.001}$ | $0.378 \pm 0.001$ | [0.322, 0.407] |
| scm20d | Mean | 0.755 | $0.754 \pm 0.003$ | $0.754 \pm 0.003$ | $\mathbf{0.753 \pm 0.001}$ | $0.754 \pm 0.002$ | [0.394, 0.763] |
| | lbl | **0.613** | $0.618 \pm 0.002$ | $0.618 \pm 0.002$ | $0.614 \pm 0.001$ | $\mathbf{0.613 \pm 0.001}$ | [0.356, 0.678] |
| | mtlp2a | **0.628** | $0.635 \pm 0.002$ | $0.634 \pm 0.003$ | $0.632 \pm 0.001$ | $0.631 \pm 0.002$ | [0.352, 0.688] |
| | mtlp3a | **0.603** | $0.608 \pm 0.002$ | $0.608 \pm 0.003$ | $0.607 \pm 0.001$ | $0.605 \pm 0.002$ | [0.363, 0.683] |
| | mtlp4a | **0.635** | $0.645 \pm 0.002$ | $0.645 \pm 0.003$ | $0.644 \pm 0.001$ | $0.638 \pm 0.002$ | [0.374, 0.730] |
| | mtlp5a | **0.974** | $0.977 \pm 0.008$ | $0.977 \pm 0.007$ | $0.978 \pm 0.003$ | $0.975 \pm 0.006$ | [0.413, 0.846] |
| | mtlp6a | **0.981** | $0.986 \pm 0.009$ | $0.992 \pm 0.008$ | $1.002 \pm 0.004$ | $0.989 \pm 0.008$ | [0.424, 0.843] |
| | mtlp7a | **0.996** | $1.001 \pm 0.008$ | $1.004 \pm 0.007$ | $1.005 \pm 0.006$ | $1.000 \pm 0.009$ | [0.404, 0.833] |
| | mtlp8a | 0.995 | $0.997 \pm 0.010$ | $0.997 \pm 0.011$ | $1.008 \pm 0.005$ | $\mathbf{0.994 \pm 0.005}$ | [0.407, 0.851] |
| | mtlp9a | 0.708 | $0.704 \pm 0.003$ | $0.702 \pm 0.003$ | $\mathbf{0.698 \pm 0.001}$ | $0.705 \pm 0.002$ | [0.382, 0.737] |
| | mtlp10a | 0.718 | $0.722 \pm 0.004$ | $0.722 \pm 0.004$ | $\mathbf{0.716 \pm 0.001}$ | $0.723 \pm 0.003$ | [0.413, 0.753] |
| | mtlp11a | 0.729 | $0.730 \pm 0.003$ | $0.729 \pm 0.003$ | $\mathbf{0.725 \pm 0.001}$ | $0.728 \pm 0.002$ | [0.402, 0.769] |
| | mtlp12a | 0.720 | $0.718 \pm 0.004$ | $0.717 \pm 0.004$ | $\mathbf{0.712 \pm 0.002}$ | $0.716 \pm 0.003$ | [0.429, 0.787] |
| | mtlp13a | 0.711 | $0.703 \pm 0.005$ | $0.699 \pm 0.004$ | $\mathbf{0.697 \pm 0.001}$ | $0.705 \pm 0.003$ | [0.400, 0.751] |
| | mtlp14a | 0.683 | $0.673 \pm 0.004$ | $0.670 \pm 0.003$ | $\mathbf{0.668 \pm 0.001}$ | $0.675 \pm 0.002$ | [0.411, 0.779] |
| | mtlp15a | 0.684 | $0.674 \pm 0.004$ | $0.671 \pm 0.004$ | $\mathbf{0.666 \pm 0.001}$ | $0.678 \pm 0.002$ | [0.384, 0.727] |
| | mtlp16a | 0.689 | $0.677 \pm 0.005$ | $0.676 \pm 0.005$ | $\mathbf{0.672 \pm 0.001}$ | $0.682 \pm 0.003$ | [0.386, 0.754] |

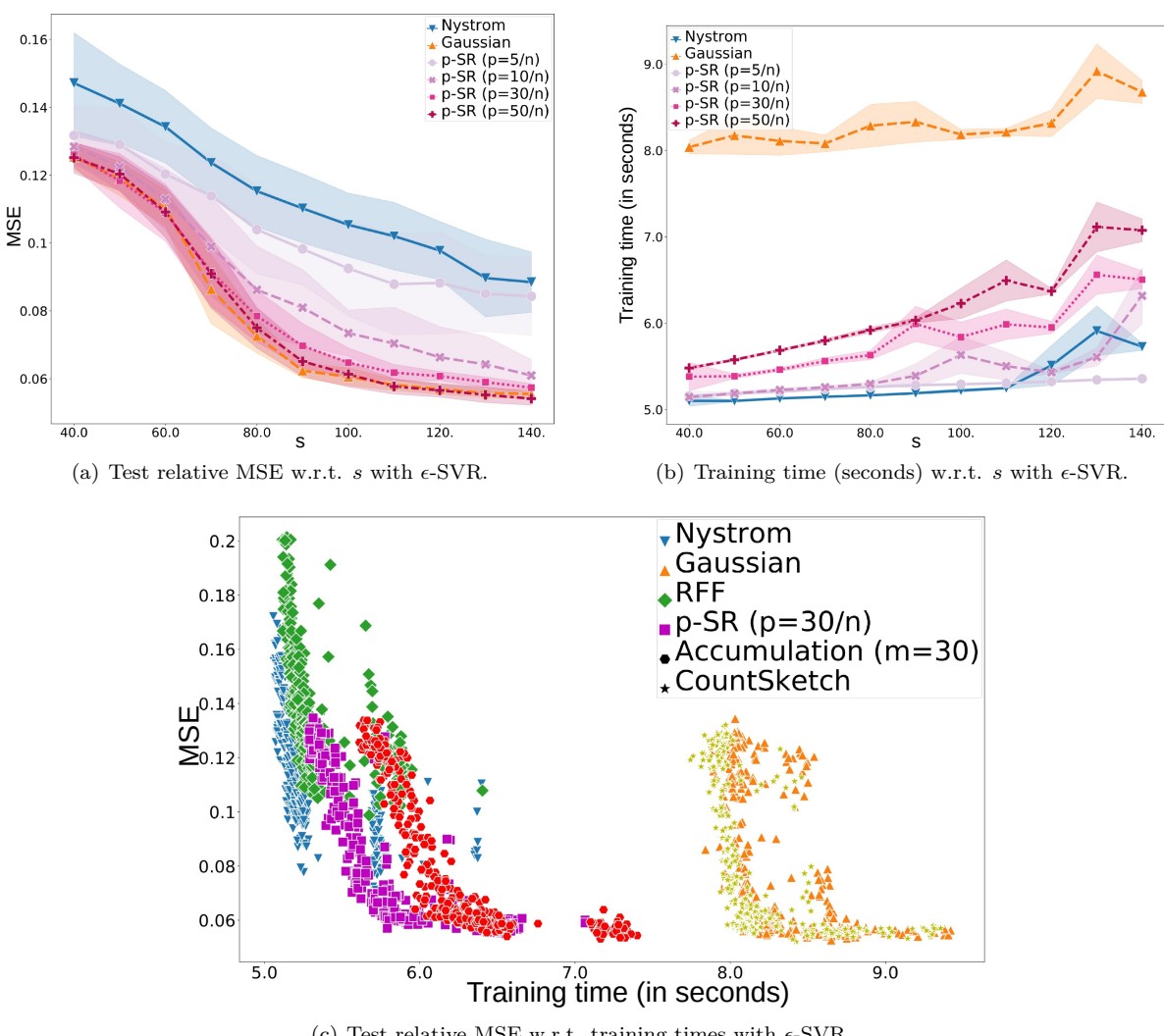

(a) Test relative MSE w.r.t. $s$ with $\epsilon$-SVR.

(b) Training time (seconds) w.r.t. $s$ with $\epsilon$-SVR.

(c) Test relative MSE w.r.t. training times with $\epsilon$-SVR.

Figure 3: Trade-off between Accuracy and Efficiency for $p$-SR sketches with $\epsilon$-SVR on synthetic dataset.

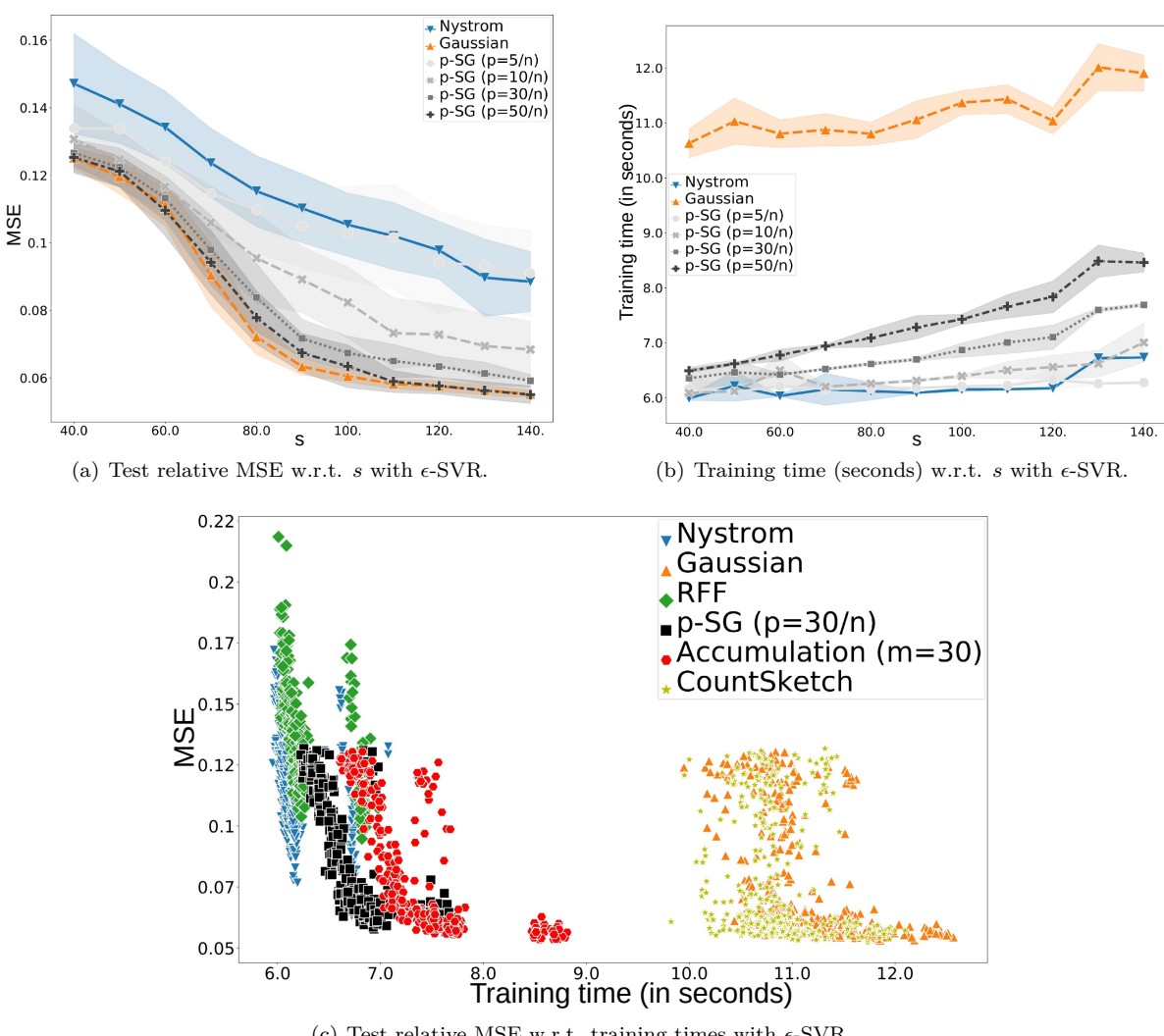

(a) Test relative MSE w.r.t. $s$ with $\epsilon$-SVR.

(b) Training time (seconds) w.r.t. $s$ with $\epsilon$-SVR.

(c) Test relative MSE w.r.t. training times with $\epsilon$-SVR.

Figure 4: Trade-off between Accuracy and Efficiency for $p$-SG sketches with $\epsilon$-SVR on synthetic dataset.

