# OpenReview forum: "Fast Kernel Methods for Generic Lipschitz Losses via $p$-Sparsified Sketches"
_TMLR — Accepted by TMLR_

### Review · Reviewer_sd1L · 2023-06-30

**Summary Of Contributions:**

The broad context of this paper is that of designing and analyzing kernel-based machine learning methods, particularly for regression tasks, which are more computationally efficient than more naive implementations, while maintaining statistical guarantees (up to natural tradeoffs due to reduced precision).

The more refined context is that of "sketching." The basic idea is to replace the Gram matrix that appears in standard formulations of kernel-based ERM with a matrix of smaller dimension (by replacing gram matrix $K$ with $SKS^{\\top}$), here sketch matrix $S$ is $s \\times s$ instead of $K$ which is $n \times n$, where typically $s \\ll n$, and $n$ is the number of (iid) data points available for training. Arguably the simplest approach is to randomly sub-sample this data, but there are numerous other approaches. The first contribution of this work is a rather general analysis of ERM under sketching, with the critical technical starting point being the notion of "satisfiability", a property of the sketching matrix $S$. Roughly speaking, we want $S$ to preserve the "important" information in $K$ (preserving length of largest eigenvectors of $K$), while discarding the less important information. Their results take the form of excess risk bounds that hold for Lipschitz losses and bounded kernels, with ERM restricted to a ball around the true risk minimizer in RKHS, assuming that $S$ satisfies the "satisfiability" requirement. They handle both single output and multi-output regression tasks, but the overall flavor is the same.

The other main contribution is a method for generating sketch matrices (what they call $p$-sparsified sketches, of the Rademacher and Gaussian varieties), plus formal characterization of a "good event" (since the sketching is probabilistic) of high probability on which the $K$-satisfiability of the (random) $S$ generated by their method holds (Theorem 5). The notion of $p$-sparse sketches itself is not new here; the basic idea is simply to set a fixed probability (here, $1-p$) that each iid element of $S$ will be zero, otherwise it follows a typical (scaled) Gaussian/Rademacher distribution. According to the authors, the Rademacher variety has appeared previously (although with theoretical guarantees lacking), but the Gaussian variety is new here. By enforcing such sparsity, the authors show (Eqn 14) how the sketching can then be split into a fast sub-sampling sketch combined with a (small, post-sparsification) Gaussian sketch; this is only possible when full columns of $S$ are zero vectors, which is at least in principle possible here, whereas many existing sketching methods always have at least one non-zero element per column. They provide basic empirical evidence for the efficiency of their approach using both simulated and benchmark datasets.

**Audience:**

Yes

**Broader Impact Concerns:**

No relevant concerns.

**Claims And Evidence:**

Yes

**Requested Changes:**

Please see the comments above; none of the suggestions are critical to my recommendation, as I think the paper is already quite well-done as is, but I think would benefit the paper.

Besides these points, here are a few minor points that I think the authors should consider, in no particular order of importance.

- Why do the authors give the quadratic loss result in Thm 2? Bounding $\\mathcal{Y}$ and assuming a bounded kernel and finite diameter ball around the risk minimizer yields Lipschitzness for this function; is this special case for comparison with other works? Do there exist similar results (excess risk bounds) assuming the quadratic loss but without bounded $\\mathcal{Y}$? If so, I think this would be a natural point of comparison (such comparisons appear in more general regression contexts, outside of sketching/kernels).

- A point on nomenclature; using the current naming, a $0$-sparsified sketch is completely sparse, which is somewhat unappealing aesthetically.

- In the exposition following Eqn (14), the authors say that in their decomposition $S\_{\\mathrm{SG}}$ is a "sub-Gaussian sketch". I don't see where this term comes from, and I'm not sure if it is meaningful here. Is this term used just because the elements of $S\_{\\mathrm{SG}}$ are either Rademacher or Gaussian (assuming we are restricted to the confines of Defn 5), which are both sub-Gaussian? We could say the same thing about $S$ itself, so it seems weird to introduce the term here as if $S\_{\\mathrm{SG}}$ is special in this sense. Maybe I'm missing something.

- Assumption 5: I think it would be better to say "$S$ is $K$-satisfiable with probability 1" or something similar.

- $\\tilde{f}\_{s}$: this is a critical object in Thm 2, but it is buried in Defn 2; a link to this in Thm 2 would be nice.

- Defn 2: replace $<<$ with $\\ll$.

- Spelling/grammar: "sketchin" (p.2), "kernels methods" (p.2), ...

**Strengths And Weaknesses:**

Overall, I think this is a solid, well-written paper. To the best of my understanding, the theoretical results are not particularly ground-breaking, but having risk bounds for a wide variety of sketched ERM algorithms tied together by the satisfiability property is a nice result, even with the limitation to Lipschitz losses and bounded kernels; such restrictions can likely be (in principle) removed using some robustification techniques, although likely at great computational cost. As well, I cannot tell if the satisfiability guarantee for the authors $p$-sparse sketches is an easy extension of existing results for Gaussian sketches or not, but it is a natural extension and is nice to have.

I am not really familiar with the related literature, but I felt that the authors were quite meticulous in their comparison of technical results here with those established in the existing literature.

In terms of weaknesses, my personal feeling is that more weight could have been placed on empirical analysis of the gains we get due to the "decomposition trick," and how the statistical error changes across different datasets, all else equal. As $s$ gets larger, with $p$ fixed, it gets less and less likely that any column of $S$ will be completely filled with zeros, yet the computational boost (Fig 1(b)) doesn't change much. I found this a bit odd myself, but maybe I'm missing something. That said, the authors dedicate a lot of space to the scalar-->vector extension of their ERM analysis, a decision which I respect, but if the key point of interest in this paper is supposed to be the efficiency of $p$-sparse sketching, I think putting a bit more weight on related tests would be better aligned with the overall take-home message here.

If this were a review for an international conference, I think some reviewers would take issue that the ERM analysis is quite straightforward; it appears to be typical concentration plus porting the approximation error analysis of Yang et al 2017 under satisfiability. That said, this is TMLR, and as far as I can tell the results are valid, so the claims are solid. Plus, the multi-output extension has been formulated in a clean way, which I also think has value.

---

> ### Author Response · Authors · 2023-07-11
>
> > Overall, I think this is a solid, well-written paper.
>
> > I am not really familiar with the related literature, but I felt that the authors were quite meticulous in their comparison of technical results here with those established in the existing literature.
>
> Thank you for your positive comments about our work!
>
>
> > In terms of weaknesses, my personal feeling is that more weight could have been placed on empirical analysis of the gains we get due to the "decomposition trick," and how the statistical error changes across different datasets, all else equal. That said, the authors dedicate a lot of space to the scalar-->vector extension of their ERM analysis, a decision which I respect, but if the key point of interest in this paper is supposed to be the efficiency of $p$-sparse sketching, I think putting a bit more weight on related tests would be better aligned with the overall take-home message here.
>
> Thank you for this relevant remark. Actually, in this work, we wanted to show that, in many cases, thanks to sketching, we are able to accelerate kernel methods while maintaining very close statistical errors. In particular, the extension to the multi-output case constitutes a significant contribution since very few works in the literature approach the scalability of kernel methods in this case, along with a study of the excess risk bounds. This is why we propose many experiments in this setting with of course a focus on sparsified sketches and a comparison with other sketching distributions.
> Our main focus was to compare the sparsified sketches, first, with the Nyström approximation and the Gaussian sketching to highlight how sparsified sketches interpolate between these two with respect to $p$, and then with other sparse sketches and kernel approximation techniques, in terms of statistical errors and training times.
>
>
> > As $s$ gets larger, with $p$ fixed, it gets less and less likely that any column of $S$ will be completely filled with zeros, yet the computational boost (Fig 1(b)) doesn't change much. I found this a bit odd myself, but maybe I'm missing something.
>
> Note that, in the decomposition trick ($S = S_\mathrm{SG}\,S_\mathrm{SS}$ and $,S_\mathrm{SS} \in \mathbb{R}^{s^\prime \times n}$), the expected value of the intermediate size of the sub-sampling sketch obtained is $\mathbb{E}\left[s^\prime\right] = n (1 - \left(1-p\right)^s) \underset{p \rightarrow 0}{\sim} n s p$. Hence, for $s \in [40, 140]$, if $p = 5/n$, i.e. the lowest value of $p$ in Fig 1(b), $\mathbb{E}\left[s^\prime\right] \in [200, 700]$, whereas if $p = 50/n$, i.e. the highest value of $p$ in Fig 1(b), $\mathbb{E}\left[s^\prime\right] \in [2000, 7000]$. As a consequence, the complexity induced by the sketching matrix will vary much more for higher values of $p$, and the training time for $p = 50/n$ indeed increases much more with $s$ than $p = 5/n$'s training time.
>
>
> > Why do the authors give the quadratic loss result in Thm 2? Bounding $\mathcal{Y}$ and assuming a bounded kernel and finite diameter ball around the risk minimizer yields Lipschitzness for this function; is this special case for comparison with other works?
>
> We indeed give the quadratic loss result in Thm 2 to compare with other works. In particular, we extend the results obtained in Yang et al. (2017, Theorem 2) on $K$-satisfiable sketches with the empirical risk, i.e. $\frac{1}{n} \sum\_{i=1}^n (\tilde{f}\_s(x\_i) - f\_{\mathcal{H}\_k}(x\_i))^2$, which is actually equal to what we call the approximation error term in our decomposition of the expected risk, i.e. $\mathbb{E}\_n[\ell\_{\tilde{f}\_s}] - \mathbb{E}\_n[\ell\_{f\_{\mathcal{H}\_k}}]$. Thus, we consistently obtain that this term is proportional to $\lambda\_n + \delta\_n^2$, and we additionally have two terms proportional to $1/\sqrt{n}$ from the generalisation error terms, i.e. $\mathbb{E}[\ell\_{\tilde{f}\_s}] - \mathbb{E}\_n[\ell\_{\tilde{f}\_s}] + \mathbb{E}\_n[\ell\_{f\_{\mathcal{H}\_k}}] - \mathbb{E}[\ell\_{f\_{\mathcal{H}\_k}}]$ due to the fact that we focus on the expected risk, that we bound thanks to the Rademacher complexity theory, which is the reason why we need Lipschitzness of the loss functions. Besides, we can also compare to the results obtained in Li et al. (2021, Theorem 1) on Random Fourier Features, relying on very similar error decomposition and proof techniques.

---

> ### Author Response · Authors · 2023-07-11
>
> > Do there exist similar results (excess risk bounds) assuming the quadratic loss but without bounded $\mathcal{Y}$? If so, I think this would be a natural point of comparison (such comparisons appear in more general regression contexts, outside of sketching/kernels).
>
> Such results exist but they rely on very different proof techniques, that are specific to the quadratic loss setting. For instance, they do not use an error decomposition with generalisation error terms and do not use Rademacher complexity theory. See Caponnetto and De Vito (2007) for non-approximated kernel methods, Rudi et al. (2015) for kernel methods with Nystr\"{o}m approximation (no result exists for other sketching approximations) and Rudi and Rosasco (2017) for Random Fourier Features.
>
>
> > A point on nomenclature; using the current naming, a $0$-sparsified sketch is completely sparse, which is somewhat unappealing aesthetically.
>
> This is indeed a bit confusing. Note that the $p$ nomenclature derives from the parameter of a Bernoulli random variable which multiplies each entry of the subGaussian matrix, such that $p$ is the probability of the entry not-being null.
>
>
> > In the exposition following Eqn (14), the authors say that in their decomposition $S_\mathrm{SG}$ is a "sub-Gaussian sketch". I don't see where this term comes from, and I'm not sure if it is meaningful here. Is this term used just because the elements of $S_\mathrm{SG}$ are either Rademacher or Gaussian (assuming we are restricted to the confines of Defn 5), which are both sub-Gaussian? We could say the same thing about $S$ itself, so it seems weird to introduce the term here as if $S_\mathrm{SG}$ is special in this sense. Maybe I'm missing something.
>
> Actually, the components of $S_\mathrm{SG}$ are either Rademacher/Gaussian or zeros (depending on the value of the Bernouilli random variables), hence it is a sub-Gaussian sketching matrix since the product of a Rademacher/Gaussian r.v. with an independent Bernouilli r.v. is sub-Gaussian. This is indeed also the case of $S$ itself as you point out, but we wanted to refer to matrix $S_\mathrm{SG}$ in the decomposition.
>
>
> > Assumption 5: I think it would be better to say "$S$ is $K$-satisfiable with probability 1" or something similar.
>
> Actually, the $K$-satisfiability is a deterministic property in the sense that a deterministic matrix can be $K$-satisfiable.
> On the other side, a stochastic matrix may be $K$-stisfiable with high probability, as is the case for $p$-sparsified sketches.
> We thus state Asm. 5, Thms 2 and 4 in the general case, without loss of generality. For a random matrix, one works conditionally on the event that the matrix is $K$-satisfiable, see Corollaries 1 and 2 in Section C of the Appendix. Thank you for this remark, see the changes in blue in Definition 2 and Assumption 5 in the revised version.
>
>
> > $\tilde{f}_s$: this is a critical object in Thm 2, but it is buried in Defn 2; a link to this in Thm 2 would be nice.
>
> > Defn 2: replace $<<$ with $\ll$ .
>
> > Spelling/grammar: "sketchin" (p.2), "kernels methods" (p.2), ...
>
> Thank you, see Thm. 2, Def 2 and p.2 in the revised version.

---

### Review · Reviewer_JjKC · 2023-07-03

**Summary Of Contributions:**

In the context of kernel methods, the paper studies a) learning with generic Lipschitz losses and b) faster computation of Gram matrix produced by so called *sparsified* sketching techniques.
As random projections methods, sketching techniques reduce the dimension of the problem while preserving useful information. However computing the resulting kernel matrices is either still costly, either results in poor quality sketches.
By introducing *p-sparsified sketches*, the authors allow for faster numerical solving of the subsequent optimization problems, as well as a reduced memory footprint, while preserving useful information from the full dataset.

In addition, the contribution goes beyond the squared loss by handling generic Lipschitz losses, which enables it to encompass the framework of robust regression (Huber loss), quantile regression with the pinball loss or the popular $\epsilon$-insensitive losses.
In addition to standard scalar valued kernels, the results also apply to matrix-valued kernels, which play a role in multitask/multioutput learning, in the case of a *decomposable* kernel writing $K(\cdot, \cdot) = k(\cdot, \cdot) M$ for a fixed matrix $M$.



**Audience:**

Yes

**Claims And Evidence:**

Yes

**Requested Changes:**

Questions:
- P4 "The existence and uniqueness of $\delta_n^2$ is guaranteed for any unit ball: when what lives in the unit ball? And isn't existence clear since $\mu_i$ are fixed, and the RHS grows like delta^2 while the LHS becomes constant for $\delta \geq \mu_1$ and is strictly increasing before?
- K-satisfiability of S with respect to ... instead of $\delta^2_n$: where above is K-satisfiability proved with respect to $\delta_n^2$, is it not assumed with respect to any constant $c$ in assumption 5? Clarifying this in assumption 5 (instead of saying below "where $c$ is the constant of Ass. 5) may help.

Typos:
- P2, learning vector-valued... have been
- P2 sketchin
- P3, P4 is the notation arg inf instead of arg min standard?
- why using the notation $\mathcal{F}(\mathcal{X}, \mathbb{R})$ when above you have used $\mathcal{Y}^\mathcal{X}$?
- In Definition 2 use $\ll$ instead of $<<$
- In Def 3 why using $\tilde \gamma$ instead of $\hat \gamma$, by analogy  with $\hat \alpha$?
- $n^2$-identity matrix: is this the $n$ by $n$ identity matrix? If so why not using the introduced notation $I_n$?
- P3 the excess of risk
- P4 the estimators ... have bounded normS
- An MVK : A MVK
- P6: The 3rd term become
- P17 We first prove first
- P17 use of cite/citep/citet for Li et al, Bartlett etc.
- "let assume" is not correct

**Strengths And Weaknesses:**

Contributions:
- excess risk bounds for sketching of kernels with *generic Lipschitz loss* (Thm 2, Thm 4 for the multioutput case)
- theoretical validity of sparsified Gaussian and Rademacher sketching techniques is proved
- a technique to learn these kernels with an approximated feature map is proposed

---

> ### Author Response · Authors · 2023-07-11
>
> > Strengths And Weaknesses
>
> Thank you for pointing out the different contributions from our paper!
>
>
> > P4 "The existence and uniqueness of $\delta_n^2$ is guaranteed for any unit ball: when what lives in the unit ball? And isn't existence clear since $\mu_i$ are fixed, and the RHS grows like $\delta_n^2$ while the LHS becomes constant for $\delta_n \geq \mu_1$ and is strictly increasing before?
>
> Thank you for this comment, it is indeed confusing. Actually, this quantity $\psi(\delta_n) =
> (\frac{1}{n} \sum_{i=1}^n \min(\delta_n^2, \mu_i))^{1/2}$ has been first introduced in statistical learning theory, more specifically in the study of the local Rademacher complexity of kernel classes, i.e. the Rademacher complexity of function with low variance in the RKHS of a kernel, see Bartlett et al. (2005, Section 6.3) or Li et al. (2021, Section 2.3). In particular, they show that, by restricting the hypothesis space to the unit ball of the RKHS, i.e. by taking $\mathcal{F}=\{f \in \mathcal{H}: \left\|f\right\|\_{\mathcal{H}} \leq 1, \mathbb{E}\_x[f(x)^2]\leq \delta\_n^2\}$, the local Rademacher complexity is bounded by $\psi(\delta\_n)$, i.e. $\mathbb{E}\_{\sigma}\left[ \sup\_{f \in \mathcal{F}} \left| \frac{2}{n} \sum\_{i=1}^n \sigma\_i f(x\_i) \right| | x\_1, \ldots, x\_n \right] \leq \sqrt{2} \psi(\delta\_n)$, where $\sigma\_1, \ldots, \sigma\_n$ are independent Rademacher random variables,  see Bartlett et al. (2005, Lemma 6.6).
>
> However, you are right to point out that $x \mapsto (\frac{1}{n} \sum_{i=1}^n \min(x, \mu_i))^{1/2}$ is equal to the square root function for $x \leq \mu_n$, to a constant for $x \geq \mu_1$, and is strictly increasing in between, and then regardless of the values the $\mu_i$s, its curve crosses $x \mapsto x$'s curve once and then existence and uniqueness of $\delta_n^2$ is guaranteed. Thus, this is true for the RKHS of any kernel and there is no restriction to the unit ball to consider. This is corrected (see p.4 in the revised version), thank you for pointing out this confusion.
>
>
> > $K$-satisfiability of $S$ with respect to ... instead of $\delta_n^2$: where above is $K$-satisfiability proved with respect to $\delta_n^2$, is it not assumed with respect to any constant $c$ in assumption 5? Clarifying this in assumption 5 (instead of saying below "where $c$ is the constant of Ass. 5) may help.
>
> Actually, the RHS of the $K$-satisfiablity means that, for a given deterministic matrix $S$ or a given distribution of random matrices, one has to prove that $\left\|S U_{2} D_{2}^{1 / 2}\right\|_{\operatorname{op}} \leq c \delta_n$ for a $c > 0$ independent of $n$. Hence, this constant $c$ depends on $S$ or the distribution of $S$: the lower it is, the better $S$ or its distribution is. The goal is to prove the RHS of $K$-satisfiable with the tighter possible $c$. As a consequence, we indeed need to say that a sketching matrix is $K$-satisfiable for a given constant $c > 0$, as stated in Thm. 5 and in blue in Asm. 5 in the revised version of the paper. Thank you for your relevant remark.
>
>
> > P3, P4 is the notation arg inf instead of arg min standard?
>
> We keep the notation arg inf on P3 since there is no guarantee of the existence of the solution, but we use arg min on P4 since we assume the existence of $f_{\mathcal{H}_k}$. Thank you for this question.
>
>
> > In Def 3 why using $\tilde{\gamma}$ instead of $\hat{\gamma}$, by analogy with $\hat{\alpha}$?
>
> We prefer to have different notations $\hat{f}_n$ and $\tilde{f}_s$ to not confuse between non-sketched and sketched estimators, and then keep $\hat{ }$ for the coefficients of the non-sketched estimator and $\tilde{ }$ for the coefficients of the sketched estimator.
>
>
> > Typos
>
> The rest of the typos have been corrected, see blue notes in the revised version, thank you for pointing them out.

---

### Review · Reviewer_sWZq · 2023-07-04

**Summary Of Contributions:**

This paper studies a sketch-based approximation method of kernel regression. The following three items summarize the contributions of the authors.
1. They derive a risk bound of the sketched kernel regression.
2. They show that sketching by a sparse random matrix (sparse Gaussian or Rademacher) improves the time and space complexity while maintaining the theoretical guarantee.
3. They derive an algorithm for training.

**Audience:**

Yes

**Broader Impact Concerns:**

No concern.

**Claims And Evidence:**

Yes

**Requested Changes:**

Minor changes (not mandatory)
1. It would be nice to discuss how tight the bounds of Theorem 4 are. For example, you can plot the theoretical bounds in Figure 1 (a) in addition to the empirical errors, allowing us to see the gap.
2. You can summarize the conclusion of the optimal p~=0.7 in a theorem style so that readers easily digest it.

**Strengths And Weaknesses:**

Strengths
1. The paper is well-organized and easy to read.
2. The risk bound is derived in a theoretical way.
3. The performance is evaluated with both synthetic and real datasets, which provides empirical support for Contribution 2.

Weaknesses
1. The tightness of the derived bound is not discussed.
2. The discussion of optimal p (sparsity parameter) on page 8 could be summarized in a more explicit way.

---

> ### Author Response · Authors · 2023-07-11
>
> > The paper is well-organized and easy to read.
> > The risk bound is derived in a theoretical way.
> > The performance is evaluated with both synthetic and real datasets, which provides empirical support for Contribution 2.
>
> Thank you for your positive comments about our work!
>
>
> > It would be nice to discuss how tight the bounds of Theorem 4 are. For example, you can plot the theoretical bounds in Figure 1 (a) in addition to the empirical errors, allowing us to see the gap.
>
> Thank you for this remark, we added a small discussion of the rate obtained in the least square case in comparison with the literature, in particular,  Caponnetto and De Vito (2007) and Ciliberto et al. (2020)  (see p.6 of the revised version).
>
> Note that in Figure 1 (a) the empirical errors are plotted with respect to the sketching size $s$, and not the number of training samples $n$, such that adding the theoretical bound is not straightforward.
>
> The goal of these experiments is to study the $p$-sparsified sketches and to see that, by varying $p$, they interpolate between the Nyström approximation and the Gaussian sketching, and then show their relevance in terms of statistical error/computational training time trade-off. Besides, in the other experiments, we wanted to show the performance of our algorithm on real-world multi-output applications and show that sketching allows us to maintain good statistical performance in comparison to non-sketched kernel methods while being more efficient.
>
>
> > You can summarize the conclusion of the optimal p~=0.7 in a theorem style so that readers easily digest it.
>
> Thank you for this suggestion, we added it to the revised version (see p.8 in blue).

---

### Decision · Action_Editors · 2023-08-15

**Recommendation:** Accept as is

**Comment:**

The paper introduces the concept of p-sparsified sketches, which enable quicker numerical solutions for subsequent optimization problems. This approach also reduces memory usage while retaining valuable information from the complete dataset. The paper offers a comprehensive theoretical analysis of p-sparsified sketches. Experimental results further illustrate that the proposed method surpasses existing techniques in performance.

The paper is well-written and has an important contribution to kernel methods. All reviewers have recommended its acceptance. I also vote for acceptance.

**Audience:**

As a reviewer pointed out, the kernel method sketch technique is a narrower ML field. However, some experts are working on the topic; it is good to have in TMLR.

**Claims And Evidence:**

Yes.